# SPARK regulates AGC kinases central to the *Toxoplasma gondii* asexual cycle

Alice L Herneisen, Michelle L Peters, Tyler A Smith, Emily Shortt, Sebastian Lourido*

Whitehead Institute for Biomedical Research and Department of Biology, Massachusetts Institute of Technology, Cambridge, United States

## eLife assessment

This **fundamental** study identifies protein kinases in the parasitic protozoan, *Toxoplasma gondii* that are required for parasite invasion of host cells and differentiation to drug-resistant chronic stages. The use of advanced proteomic and functional approaches provides **compelling** evidence for the proposed signalling pathway, although additional analyses are needed to fully validate some findings. The work will be of broad interest to cell biologists and parasitologists with an interest in cell signalling and environmental sensing.

*For correspondence:
lourido@wi.mit.edu

**Abstract** Apicomplexan parasites balance proliferation, persistence, and spread in their metazoan hosts. AGC kinases, such as PKG, PKA, and the PDK1 ortholog SPARK, integrate environmental signals to toggle parasites between replicative and motile life stages. Recent studies have cataloged pathways downstream of apicomplexan PKG and PKA; however, less is known about the global integration of AGC kinase signaling cascades. Here, conditional genetics coupled to unbiased proteomics demonstrates that SPARK complexes with an elongin-like protein to regulate the stability of PKA and PKG in the model apicomplexan *Toxoplasma gondii*. Defects attributed to SPARK depletion develop after PKG and PKA are down-regulated. Parasites lacking SPARK differentiate into the chronic form of infection, which may arise from reduced activity of a coccidian-specific PKA ortholog. This work delineates the signaling topology of AGC kinases that together control transitions within the asexual cycle of this important family of parasites.

## Introduction

Apicomplexans parasitize a majority of warm-blooded species, including an estimated quarter of the human population (*Jones et al., 2014*). Central to parasite success is the ability to transition between different life cycle stages to balance proliferation and transmission (*Lourido, 2019*). The acute phase of infection by *Toxoplasma gondii*, a model apicomplexan, induces the pathology of disease. Parasites invade host cells, establish an intracellular niche that subverts innate immune responses, replicate, and exit the host via lysis to establish new sites of infection (*Blader et al., 2015*). These pathogens adapt to the inverted ion and nutrient gradients within and without the host cell in minutes. Such rapid changes in cellular state depend on signal transduction in *T. gondii* and related apicomplexans. Second messenger cascades within the parasite cytoplasm are especially well-suited to rapidly transduce and amplify signals from a changing environment. Accordingly, apicomplexans—and *T. gondii* in particular—have a complete repertoire of second messenger signaling components, from kinases to cyclases and phosphodiesterases (*Brown et al., 2020*; *Pace et al., 2020*).

The roles of second messenger kinases—collectively referred to as AGC kinases for the founding members protein kinases A, G, and C—in apicomplexan life cycles have been elaborated in

candidate-by-candidate approaches. cGMP-dependent protein kinase (or protein kinase G, PKG) is necessary for the secretion of parasite-specific adhesins and perforins, enabling efficient escape and invasion of host cells (*Brown et al., 2017*; *Brown et al., 2016*; *Wiersma et al., 2004*). In apicomplexan cells, PKG operates upstream of calcium release and thus potentiates an orthogonal second messenger signaling network. Consequently, PKG inhibitors have been pursued as anti-parasitic compounds (*Baker et al., 2017*; *Donald et al., 2006*; *Sidik et al., 2016a*). Another cyclic-nucleotide-dependent protein kinase, PKA, similarly promotes parasite spread during infection; in *Plasmodium* spp. merozoites, PKA function enables parasite invasion of host cells (*Flueck et al., 2019*; *Patel et al., 2019*; *Wilde et al., 2019*), and in *T. gondii* balanced PKA C1 function is required to ensure that parasites do not exit the host cell prematurely (*Jia et al., 2017*; *Uboldi et al., 2018*). *T. gondii* has an additional ortholog of the PKA catalytic subunit, PKA C3, which maintains parasites in the acute phase of the infection and reduces conversion to a slowly proliferating, chronic infection termed the bradyzoite stage (*Sugi et al., 2016*).

Several recent studies have sought to characterize the downstream targets of second messenger kinases in apicomplexans, including PKA, PKG, and calcium-dependent protein kinases (*Alam et al., 2015*; *Balestra et al., 2021*; *Brochet et al., 2014*; *Chan et al., 2023*; *Herneisen et al., 2022*; *Jia et al., 2017*; *Nofal et al., 2022*). The second messenger networks in these parasites are intricately interwoven, with documented crosstalk between parasite cAMP, cGMP, and calcium signaling cascades. Comparatively little work has addressed upstream signal integration in apicomplexans. In metazoans, growth factor kinases such as mammalian target of rapamycin (mTOR) and phosphoinositide-dependent protein kinase 1 (PDK1) prime numerous AGC kinases for further second messenger-based activation (*Laplante and Sabatini, 2012*; *Mora et al., 2004*).

We recently identified the *S*tore *P*otentiating/*A*ctivating *R*egulatory *K*inase (SPARK), an apicomplexan ortholog of metazoan PDK1, as a fitness-conferring candidate in pooled screens of the *T. gondii* kinome. Parasites depleted of SPARK for multiple replication cycles failed to enter and exit host cells upon stimulation with zaprinast, a chemical that elevates cGMP in parasites (*Smith et al., 2022*). The *Plasmodium falciparum* ortholog of SPARK has been implicated in invasion and proliferation through the regulation of the PKA pathway (*Hitz et al., 2021*). Although parasites with disrupted SPARK alleles were characterized phenotypically, evidence for the PDK1 activity of SPARK was indirect. Here, we map the topology of the signaling network regulated by SPARK in *T. gondii*. SPARK complexes with an elongin-like protein we name SPARKEL. Phenotypic and proteomic experiments show that SPARK dysregulation lessens PKA C1, PKG, and PKA C3 function. Phenotypes attributed to SPARK depletion—reduced invasion, host cell lysis, calcium signaling, and increased differentiation—can be explained by the attenuated activities of SPARK's client kinases. The proper functioning of SPARK thus ensures that *T. gondii* progresses through the acute stage of its infection cycle.

## Results

### SPARK complexes with an elongin-like protein, SPARKEL

To identify proteins interacting with SPARK, we immunopurified the mNG epitope of the previously described SPARK-mNG-AID strain (*Smith et al., 2022*) with anti-mNG nanobodies and quantified protein abundance using label-free quantitative proteomics (*Figure 1—figure supplement 1* and *Supplementary file 1*). The immunoprecipitation (IP) successfully enriched SPARK as well as a hypothetical protein, TGGT1_291010. IPs with lysates of parasites expressing SPARK-mNG lacking AID demonstrated that the interaction was not due to the presence of the degron; TGGT1_291010 was once again highly enriched (*Figure 1A*), along with a putative AGC kinase (TGGT1_205550). The AGC kinase is restricted to coccidians but is dispensable during acute *T. gondii* infection (*Sidik et al., 2016b*); we therefore focused our efforts on the hypothetical protein, which contributes to parasite fitness.

TGGT1_291010 is a 23 kDa protein with a C-terminal SKP1/BTB/POZ domain (*Figure 1B*, *Figure 1—figure supplement 1*). This domain shares homology with that seen in metazoan Elongin C. For this reason, we named the gene TGGT1_291010 'SPARK Elongin-Like protein' (SPARKEL, *Table 1*). SPARKEL homologs are found in the free-living alveolates *Vitrella brassicaformis* and *Chromera velia* but are absent in Aconoidasida (*Figure 1C*), suggesting that the gene was present in the ancestor of

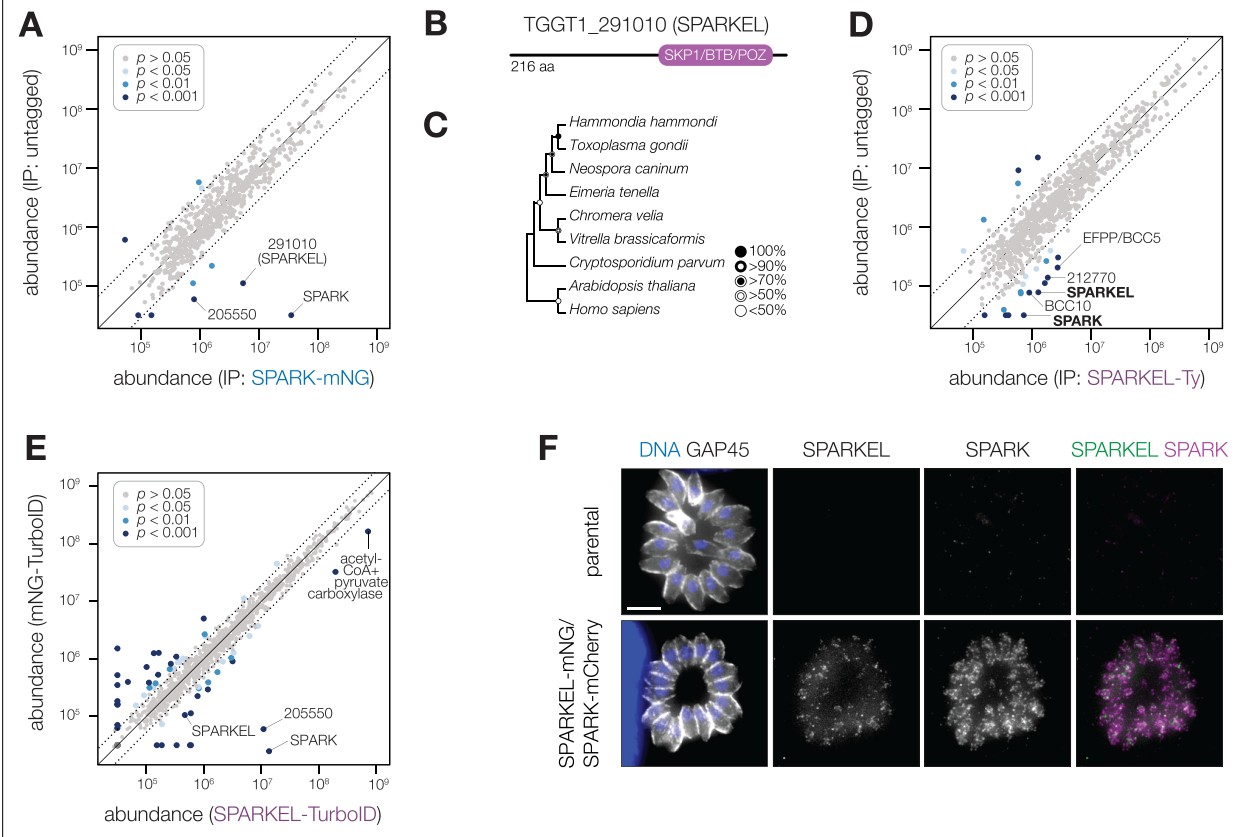

**Figure 1.** SPARK interacts with an Elongin C-like protein. (**A**) Protein abundances from immunopurified SPARK-mNG lysates or an untagged control strain. Dotted lines correspond to one modified z-score. Only proteins quantified by greater than one peptide are shown. Proteins identified in only one IP were assigned a pseudo-abundance of $10^{4.5}$. (**B**) TGGT1_291010 SPARK elongin-like protein (SPARKEL) gene model. (**C**) Neighbor-joining phylogenetic tree of SKP1/BTB/POZ domains of TGGT1_291010 orthologs in apicomplexan species along with human and *arabidopsis* proteins as outgroups. Bootstrap values determined from 1000 replicates. (**D**) Protein abundances from immunopurified SPARKEL-Ty lysates or an untagged control strain. Dotted lines correspond to one modified z-score. Only proteins quantified by greater than one peptide are shown. Proteins identified in only one IP were assigned a pseudo-abundance of $10^{4.5}$. (**E**) Protein abundances following biotinylation and streptavidin enrichment of samples derived from parasites expressing mNG- or SPARKEL-TurboID fusion constructs. A pseudocount of $10^{4.5}$ was assigned to proteins identified in only one sample. Point colors correspond to significance thresholds. Dotted lines correspond to one median absolute deviation. (**F**) Immunofluorescence microscopy of intracellular parasites co-expressing SPARKEL-mNG and SPARK-mCherry. Parasites and nuclei were stained with GAP45 and Hoechst 33342, respectively. Scale bar, 5 µm.

The online version of this article includes the following figure supplement(s) for figure 1:

**Figure supplement 1.** Additional data supporting the interaction between SPARK and SPARKEL.

apicomplexans but was selectively lost in some lineages. The function of proteins with Elongin C-like domains has not been widely investigated in unicellular eukaryotes.

Immunoprecipitation and mass spectrometry of SPARKEL enriched SPARK (*Figure 1D* and *Supplementary file 1*) and several putative components of daughter cell replication, including TGGT1_212770 (*Dos Santos Pacheco et al., 2021*), BCC10, and EFPP/BCC5. To confirm that SPARKEL and SPARK interact in situ, we introduced a C-terminal TurboID fusion at the SPARKEL endogenous locus (*Branon et al., 2018*). These transgenic parasites were treated with biotin while intracellular, and biotinylated proteins were enriched from lysates via streptavidin affinity purification. As a control, we conducted a parallel experiment with transgenic parasites expressing TurboID fused to a cytosolic mNG fluorophore. SPARK and the AGC kinase TGGT1_205550 were highly enriched in the SPARKEL proximity labeling experiment (*Figure 1E* and *Supplementary file 2*). A proximity labeling experiment with parasites expressing TurboID at the N terminus of SPARKEL similarly enriched SPARK and TGGT1_205550 (*Figure 1—figure supplement 1* and *Supplementary file 2*). Thus, the interactions between SPARK, SPARKEL, and TGGT1_205550 were consistently observed across a variety of approaches.

**Table 1.** Proteins discussed in the text.

| Gene ID | Description | Context in text | Modifications | Reference | Dataset |
|---------|-------------|-----------------|---------------|-----------|---------|
| TGGT1_268210 | AGC kinase | SPARK | | *Smith et al., 2022* | |
| TGGT1_291010 | hypothetical protein | SPARKEL | | | SPARK IP-MS |
| TGGT1_205550 | AGC kinase | AGC kinase | | | SPARK TurboID, SPARKEL TurboID |
| TGGT1_310220 | hypothetical protein | BCC10 | | *Engelberg et al., 2022* | SPARKEL IP-MS |
| TGGT1_269460 | Ser/Thr phosphatase family protein | EFPP/BCC5 | S550 | *Engelberg et al., 2022; Liang et al., 2021; Roumégous et al., 2022* | SPARKEL IP-MS, SPARK depletion timecourse phosphoproteome |
| TGGT1_311360 | protein kinase G AGC kinase family member PKG | PKG | | *Brown et al., 2017* | 24 hr SPARK depletion proteome, SPARK depletion timecourse phosphoproteome |
| TGGT1_226030 | AGC kinase | PKA C1 | | *Jia et al., 2017; Uboldi et al., 2018* | 24 hr SPARK depletion proteome, SPARK depletion timecourse phosphoproteome |
| TGGT1_242070 | cAMP-dependent protein kinase regulatory subunit | PKA R | S27 | *Jia et al., 2017; Uboldi et al., 2018* | 24 hr SPARK depletion proteome, SPARK depletion timecourse phosphoproteome, PKA C3 depletion proteome |
| TGGT1_270240 | MAG1 protein | MAG1 | S238 | *Parmley et al., 1994* | 24 hr SPARK depletion proteome, PKA C3 depletion phosphoproteome |
| TGGT1_314250 | bradyzoite rhoptry protein | BRP1 | | *Schwarz et al., 2005* | 24 hr SPARK depletion proteome, 24 hr PKA C3 depletion proteome |
| TGGT1_208740 | putative microneme protein | | | *Waldman et al., 2020* | 24 hr SPARK depletion proteome, 24 hr PKA C3 depletion proteome |
| TGGT1_264660 | SAG-related sequence SRS44 | CST1 | | *Tomita et al., 2013* | 24 hr SPARK depletion proteome, PKA C3 depletion transcriptome |
| TGGT1_312330 | hypothetical protein | CST10 | | *Tu et al., 2020; Waldman et al., 2020* | 24 hr SPARK depletion proteome, PKA C3 depletion transcriptome |
| TGGT1_260190 | microneme protein MIC13 | MIC13 | | *Fritz et al., 2012* | 24 hr SPARK depletion proteome, PKA C3 depletion transcriptome |
| TGGT1_311100 | zinc finger (CCCH type) motif-containing protein | BFD2 | S183 | *Licon et al., 2023* | 24 hr SPARK depletion proteome, 24 hr PKA C3 depletion proteome, PKA C3 depletion phosphoproteome |
| TGGT1_266010 | phosphatidylinositol 3- and 4-kinase | PI3/4 K | S1439 | | SPARK depletion timecourse phosphoproteome |
| TGGT1_202550 | NLI interacting factor family phosphatase | CTD3/BCC6 | T384 and T386 | *Engelberg et al., 2022* | SPARK depletion timecourse phosphoproteome |
| TGGT1_224240 | protein phosphatase 2 C domain-containing protein | PPM1 | S557 | *Yang and Arrizabalaga, 2017* | SPARK depletion timecourse phosphoproteome |
| TGGT1_268770 | dual specificity phosphatase, catalytic domain-containing protein | phosphosite down-regulated with SPARK depletion | S11 | | SPARK depletion timecourse phosphoproteome |
| TGGT1_219682 | putative pyruvate dehydrogenase kinase | | S633 | *Ferrarini et al., 2021* | SPARK depletion timecourse phosphoproteome |
| TGGT1_225960 | STE kinase | | S2905 | | SPARK depletion timecourse phosphoproteome |
| TGGT1_305860 | calcium-dependent protein kinase 3 | CDPK3 | T40 | | SPARK depletion timecourse phosphoproteome |
| TGGT1_267580 | cyclin2-related protein | | S491 | | SPARK depletion timecourse phosphoproteome |
| TGGT1_204280 | cell-cycle-associated protein kinase DYRK | | T642, S645, and S649 | *Smith et al., 2022* | SPARK depletion timecourse phosphoproteome |
| TGGT1_239885 | hypothetical protein | | S933 and S936 | *Smith et al., 2022* | SPARK depletion timecourse phosphoproteome |
| TGGT1_267100 | protein phosphatase 2 C domain-containing protein | PPM2B | S/T778-788 | *Yang et al., 2019* | SPARK depletion timecourse phosphoproteome |
| TGGT1_311310 | protein phosphatase 2B catalytic subunit, calcineurin family phosphatse superfamily protein | CnA | S104 | *Paul et al., 2015* | SPARK depletion timecourse phosphoproteome |
| TGGT1_277895 | ubiquitin carboxyl-terminal hydrolase | UBP1 | S44, S52, S1691, and S1695 | *Koreny et al., 2023; Wan et al., 2023* | SPARK depletion timecourse phosphoproteome |
| TGGT1_294360 | putative ubiquitin specific protease 39 isoform 2 | USP39 | S20 | | SPARK depletion timecourse phosphoproteome |
| TGGT1_226050 | hypothetical protein | RBR E3 ligase | S319 and S322 | | SPARK depletion timecourse phosphoproteome |
| TGGT1_239410 | hypothetical protein | putative CCR4-NOT complex subunit | S80 | | SPARK depletion timecourse phosphoproteome |

*Table 1 continued on next page*

*Table 1 continued*

| Gene ID | Description | Context in text | Modifications | Reference | Dataset |
|---------|-------------|-----------------|---------------|-----------|---------|
| TGGT1_295658 | zinc finger in N-recognin protein | UBR box E3 ligase | S3335 | | SPARK depletion timecourse phosphoproteome |
| TGGT1_295710 | HECT-domain (ubiquitin-transferase) domain-containing protein | HECT E3 | S2166, S4313 | | SPARK depletion timecourse phosphoproteome, PKA R depletion timecourse phosphoproteome |
| TGGT1_216130 | putative ubiquitin conjugating enzyme E2 | putative E2 enzyme | S193 | | SPARK depletion timecourse phosphoproteome, PKA R depletion timecourse phosphoproteome |
| TGGT1_293670 | transcription elongation factor A TFIIS | TFIIS | S356 | | SPARK depletion timecourse phosphoproteome |
| TGGT1_226810 | histone lysine methyltransferase SET1 | SET1 | S1780 and S1781 | | PKA C3 and SPARK depletion timecourse phosphoproteome |
| TGGT1_218070 | hypothetical protein | NOC3p | S1343 | | PKA C3 and SPARK depletion timecourse phosphoproteome |
| TGGT1_280800 | SWI2/SNF2 SRCAP/Ino80 | TgSRCAP | S635 and S638 | *Sullivan et al., 2003* | SPARK depletion timecourse phosphoproteome |
| TGGT1_306660 | RNA pseudouridine synthase superfamily protein | PUS1 | S506 | *Anderson et al., 2009* | SPARK depletion timecourse phosphoproteome |
| TGGT1_312370 | RNA pseudouridine synthase superfamily protein | PUS protein | S951 | | SPARK depletion timecourse phosphoproteome |
| TGGT1_231970 | pre-mRNA processing splicing factor PRP8 | PRP8 | T820 | | SPARK depletion timecourse phosphoproteome |
| TGGT1_310820 | putative SLU7 splicing factor | SLU7 | S[516-519] | | SPARK depletion timecourse phosphoproteome |
| TGGT1_267600 | FHA domain-containing protein | a FHA protein | S406 and S413 | | SPARK depletion timecourse phosphoproteome |
| TGGT1_216670 | FUSE-binding protein 2/KH type splicing regulatory protein | KH protein | S8 | | PKA R, PKA C3, and SPARK depletion phosphoproteome |
| TGGT1_241170 | hypothetical protein | KH protein | S362, S368, and S665 | | SPARK depletion timecourse phosphoproteome |
| TGGT1_235930 | domain K- type RNA binding proteins family protein | KH protein | S58 and S59 | *Farhat et al., 2021* | SPARK depletion timecourse phosphoproteome |
| TGGT1_227450 | hydrolase, NUDIX family protein | DCP2 homolog | T1982 and S1985 | | SPARK depletion timecourse phosphoproteome |
| TGGT1_260600 | Pumilio-family RNA binding repeat-containing protein | PUF1 | S225 | *Joyce et al., 2013; Liu et al., 2014* | SPARK depletion timecourse phosphoproteome |
| TGGT1_246040 | MIF4G domain-containing protein | MIF4G domain protein | S341 and S383 | | SPARK depletion timecourse phosphoproteome |
| TGGT1_249610 | hypothetical protein | CBP80 | S851 and S853 | *Gissot et al., 2013* | PKA C3 and SPARK depletion timecourse phosphoproteome |
| TGGT1_317720 | putative eukaryotic translation initiation factor 3 subunit 7 | eIF3 | S587 | | SPARK depletion timecourse phosphoproteome |
| TGGT1_231480 | putative GCN1 | GCN1 | S1100 | | SPARK depletion timecourse phosphoproteome |
| TGGT1_251630 | slc30a2 protein | TgZnT | S448 | *Chasen et al., 2019* | PKA C3 and SPARK depletion timecourse phosphoproteome |
| TGGT1_288540 | nucleoside transporter protein | nucleoside transporter | | | SPARK depletion timecourse phosphoproteome |
| TGGT1_260310 | ATP-binding cassette transporter ABC.B1 | ABC transporter | | | SPARK depletion timecourse phosphoproteome |
| TGGT1_318710 | ATP-binding cassette sub-family F member 1 | ABC transporter | | | SPARK depletion timecourse phosphoproteome |
| TGGT1_280660 | HECT-domain (ubiquitin-transferase) domain-containing protein | putative HECT-domain E3 ubiquitin ligase | S5275 and S5279 | | SPARK TurboID, SPARK depletion timecourse phosphoproteome |
| TGGT1_286470 | AGC kinase | PKA C3 | | *Sugi et al., 2016* | SPARK TurboID |
| TGGT1_218240 | hypothetical protein | IMC25 | S1231 | *Wang et al., 2016* | PKA R and SPARK depletion phosphoproteome |
| TGGT1_220900 | hypothetical protein | AC13 | S101, S220, S659 | *Back et al., 2023* | PKA R, PKG, PKA C3, and SPARK depletion phosphoproteome |
| TGGT1_225690 | hypothetical protein | CIP1 | S850 and S851 | *Long et al., 2017* | PKA R and SPARK depletion phosphoproteome |
| TGGT1_257300 | hypothetical protein | CIP2 | S26 | *Long et al., 2017* | PKA R and SPARK depletion phosphoproteome |
| TGGT1_308860 | hypothetical protein | AC3 | S216 | *Chen et al., 2015* | PKA R and SPARK depletion phosphoproteome |
| TGGT1_315150 | putative eukaryotic initiation factor-4E | eIF4E | S1231 | | PKA R and SPARK depletion phosphoproteome |
| TGGT1_235930 | domain K- type RNA binding proteins family protein | KH protein | S58 and S59 | *Farhat et al., 2021* | PKA R and SPARK depletion phosphoproteome |

*Table 1 continued on next page*

*Table 1 continued*

| Gene ID | Description | Context in text | Modifications | Reference | Dataset |
|---|---|---|---|---|---|
| TGGT1_251640 | ubiquitin-conjugating enzyme subfamily protein | E2 protein | S23 | | PKA R and SPARK depletion phosphoproteome |
| TGGT1_253700 | transporter, major facilitator family protein | MFS transporter | S144, S1287, S1288 | | PKG and SPARK depletion phosphoproteome |
| TGGT1_312100 | plasma membrane-type Ca(2+)-ATPase A1 PMCAA1 | TgA1 | S444 and S445 | | PKG and SPARK depletion phosphoproteome |
| TGGT1_314780 | myosin G | MyoG | T1300 | *Frénal et al., 2017* | PKG and SPARK depletion phosphoproteome |
| TGGT1_295360 | hypothetical protein | IMC18 | S159, S161, and S165 | *Chen et al., 2015* | PKG and SPARK depletion phosphoproteome |
| TGGT1_239400 | hypothetical protein | IMC28 | S1081 and S1082 | *Chen et al., 2017* | PKG and SPARK depletion phosphoproteome |
| TGGT1_225560 | hypothetical protein | IMC41 | S440 | *Back et al., 2023* | PKG and SPARK depletion phosphoproteome |
| TGGT1_313480 | hypothetical protein | AAP3 | S178 | *Engelberg et al., 2020* | PKG and SPARK depletion phosphoproteome |
| TGGT1_235380 | hypothetical protein | AC5/TLAP3 | S463 | *Chen et al., 2015; Liu et al., 2013* | PKG and SPARK depletion phosphoproteome |
| TGGT1_263520 | microtubule associated protein SPM1 | SPM1 | S54 and S57 | *Tran et al., 2012* | PKG and SPARK depletion phosphoproteome |
| TGGT1_211710 | TB2/DP1, HVA22 family protein | TB2/DP1, HVA22 family proteins | S155 and S177 | | PKG and SPARK depletion phosphoproteome |
| TGGT1_257040 | TB2/DP1, HVA22 family protein | TB2/DP1, HVA22 family proteins | S32 | | PKG and SPARK depletion phosphoproteome |
| TGGT1_229490 | tetratricopeptide repeat-containing protein | TPR protein TGGT1_229490 | S456, S570 and S575 | | PKG and SPARK depletion phosphoproteome |
| TGGT1_246600 | ABC1 family protein | ER ABC transporter | T1278 | | PKG and SPARK depletion phosphoproteome |
| TGGT1_232190 | Sec7 domain-containing protein | Sec7 protein | S180 | | PKG and SPARK depletion phosphoproteome |
| TGGT1_224150 | hypothetical protein | TgCOG6 | T533 and T673 | *Marsilia et al., 2023* | PKG and SPARK depletion phosphoproteome |
| TGGT1_203910 | TBC domain-containing protein | TgTBC10 | T254 and S256 | *Quan et al., 2023* | PKG and SPARK depletion phosphoproteome |
| TGGT1_250870 | DHHC zinc finger domain-containing protein | TgDHHC1 | S345 and S348 | *Frénal et al., 2013* | PKG and SPARK depletion phosphoproteome |
| TGGT1_212820 | ubiquitin family protein | ubiquitin family protein with a C-terminal extension | S368 | | PKG and SPARK depletion phosphoproteome |
| TGGT1_290970 | 8-amino-7-oxononanoate synthase | TgSPT2 | | *Nyonda et al., 2022* | PKA C3 depletion transcriptome |
| TGGT1_259020 | bradyzoite antigen BAG1 | BAG1 | | *Behnke et al., 2008* | PKA C3 depletion transcriptome |
| TGGT1_253440 | cell-cycle-associated protein kinase SRPK, putative | SRPK | | *Talevich et al., 2011* | PKA C3 IP-MS |
| TGGT1_268960 | putative 5'-AMP-activated protein kinase subunit beta-1 family protein | AMPK subunit beta | | *Yang et al., 2022* | PKA C3 IP-MS |
| TGGT1_209985 | cAMP-dependent protein kinase | secreted cAMP-dependent protein kinases | | | PKA C3 depletion proteome |
| TGGT1_356400 | cAMP-dependent protein kinase | secreted cAMP-dependent protein kinases | | | PKA C3 depletion proteome |
| TGGT1_277790 | hypothetical protein | DEP domain protein | S586-600 | | PKA C3 depletion phosphoproteome |
| TGGT1_318150 | transporter, major facilitator family protein | TgApiAT3-1 | S54 | | PKA C3 and SPARK depletion phosphoproteomes |
| TGGT1_259260 | membrane protein FtsH1 | FtsH1 | S553 | | PKA C3 depletion phosphoproteome |
| TGGT1_227280 | dense granule protein GRA3 | GRA3 | S120, T145, S197 | | PKA C3 depletion phosphoproteome |
| TGGT1_312420 | hypothetical protein | GRA38 | S457 | *Nadipuram et al., 2016* | PKA C3 depletion phosphoproteome |
| TGGT1_204340 | hypothetical protein | CST8 | T601 | *Tu et al., 2020* | PKA C3 depletion phosphoproteome |
| TGGT1_230180 | hypothetical protein | GRA24 | T144 | *Braun et al., 2013* | PKA C3 depletion phosphoproteome |
| TGGT1_269180 | MIF4G domain-containing protein | eIF4G1 | S1312 | *Holmes et al., 2023* | PKA C3 depletion phosphoproteome |
| TGGT1_218300 | zinc finger (CCCH type) motif-containing protein | zinc finger protein | S663 | | PKA C3 depletion phosphoproteome |
| TGGT1_209500 | hypothetical protein | DNA repair protein | S619 | | PKA C3 depletion phosphoproteome |

*Table 1 continued*

| Gene ID | Description | Context in text | Modifications | Reference | Dataset |
|---------|-------------|-----------------|---------------|-----------|---------|
| TGGT1_310450 | putative myosin heavy chain | IAP2 | T1269 | | PKA C3 depletion phosphoproteome |
| TGGT1_273050 | hypothetical protein | BCC8 | | *Liang et al., 2021* | PKA C3 depletion phosphoproteome |
| TGGT1_287240 | hypothetical protein | condensin 2 | | | PKA C3 depletion phosphoproteome |
| TGGT1_278660 | putative P-type ATPase4 | ATP4 | S83 and T166 | | PKA C3 depletion phosphoproteome |
| TGGT1_233130 | nucleoside transporter protein | nucleoside transporter | S6 | | PKA C3 depletion phosphoproteome |
| TGGT1_292020 | GCC2 and GCC3 domain-containing protein | CRMPb | S7300, S7303, S7363, and S7365 | *Singer et al., 2023; Sparvoli et al., 2022* | PKA C3 depletion phosphoproteome |
| TGGT1_221180 | hypothetical protein | microneme protein | S788 | | PKA C3 depletion phosphoproteome |
| TGGT1_304490 | hypothetical protein | microneme protein | S10 | | PKA C3 depletion phosphoproteome |
| TGGT1_309590 | rhoptry protein ROP1 | ROP1 | T248 | | PKA C3 depletion phosphoproteome |
| TGGT1_211290 | rhoptry protein ROP15 | ROP13 | S312 | | PKA C3 depletion phosphoproteome |
| TGGT1_258580 | rhoptry protein ROP17 | ROP17 | T51 | | PKA C3 depletion phosphoproteome |
| TGGT1_291960 | rhoptry kinase family protein ROP40 (incomplete catalytic triad) | ROP40 | S98 | | PKA C3 depletion phosphoproteome |
| TGGT1_308810 | rhoptry neck protein RON9 | RON9 | S190, S239, S327, and S375 | | PKA C3 depletion phosphoproteome |
| TGGT1_310780 | dense granule protein GRA4 | GRA4 | S248 | | PKA C3 depletion phosphoproteome |
| TGGT1_275440 | dense granule protein GRA6 | GRA6 | S133 and S134 | | PKA C3 depletion phosphoproteome |
| TGGT1_203310 | dense granule protein GRA7 | GRA7 | S62, S72, S77, S135, S153 | | PKA C3 depletion phosphoproteome |
| TGGT1_254720 | dense granule protein GRA8 | GRA8 | S198, T202, T262 | | PKA C3 depletion phosphoproteome |
| TGGT1_220240 | hypothetical protein | GRA31 | S420 | *Nadipuram et al., 2020; Young et al., 2020* | PKA C3 depletion phosphoproteome |
| TGGT1_217680 | hypothetical protein | GRA57 | S806 | *Krishnamurthy et al., 2023; Nadipuram et al., 2020; Young et al., 2020* | PKA C3 depletion phosphoproteome |
| TGGT1_215360 | hypothetical protein | GRA62 | S319 | *Cygan et al., 2021* | PKA C3 depletion phosphoproteome |
| TGGT1_249990 | hypothetical protein | GRA70 | S436 | *Krishnamurthy et al., 2023; Lockyer et al., 2023* | PKA C3 depletion phosphoproteome |
| TGGT1_289540 | hypothetical protein | SFP1 | S897 | *Young et al., 2020* | PKA C3 depletion phosphoproteome |
| TGGT1_251740 | AP2 domain transcription factor AP2XII-9 | AP2XII-9 | S1697 | | PKA C3 depletion phosphoproteome |
| TGGT1_273870 | SWI2/SNF2 ISWI-like (AT hook) | ISWI protein | T765 | | PKA C3 depletion phosphoproteome |
| TGGT1_260240 | hypothetical protein | CAF1 | S272 | | PKA C3 depletion phosphoproteome |
| TGGT1_253750 | PLU-1 family protein | PLU-1 | S3874 and S3877 | *Wang et al., 2014* | PKA C3 depletion phosphoproteome |
| TGGT1_300330 | hypothetical protein | GCFC | S27 | | PKA C3 depletion phosphoproteome |
| TGGT1_223880 | zinc finger, C3HC4 type (RING finger) domain-containing protein | zinc finger protein | S3701 | | PKA C3 depletion phosphoproteome |
| TGGT1_288380 | heat shock protein HSP90 | HSP90 | S600 | | PKA C3 depletion phosphoproteome |
| TGGT1_321520 | hypothetical protein | p23 | S109 | | PKA C3 depletion phosphoproteome |
| TGGT1_225450 | hypothetical protein | CSN3 | S509 and S517 | | PKA C3 depletion phosphoproteome |
| TGGT1_242890 | hypothetical protein | PSME4 | S3127 | | PKA C3 depletion phosphoproteome |
| TGGT1_218960 | AP2 domain transcription factor AP2XII-1 | AP2XII-1 | T1697, S1702, T1667, T1703 | *Antunes et al., 2024* | PKA C3 depletion phosphoproteome |
| TGGT1_262150 | kelch repeat and K+channel tetramerisation domain containing protein | Kelch13 | S139 | *Harding et al., 2020; Koreny et al., 2023; Wan et al., 2023* | PKA C3 depletion phosphoproteome |
| TGGT1_254940 | MIF4G domain-containing protein | eIF4G2 | S985 and S989 | *Holmes et al., 2023* | PKA C3 and SPARK depletion phosphoproteomes |
| TGGT1_320020 | transporter, major facilitator family protein | TgApiAT2 | S316 and T318 | | PKA C3 and SPARK depletion phosphoproteomes |

*Table 1 continued on next page*

*Table 1 continued*

| Gene ID | Description | Context in text | Modifications | Reference | Dataset |
|---|---|---|---|---|---|
| TGGT1_233000 | KOW motif domain-containing protein | Spt5 | S1011 and S1014 | | PKA C3 and SPARK depletion phosphoproteomes |
| TGGT1_279320 | hypothetical protein | nucleotidyltransferase | S4010 | | PKA C3 and SPARK depletion phosphoproteomes |
| TGGT1_313180 | putative cell-cycle-associated protein kinase PRP4 | PRP4 | S316 | *Swale et al., 2022* | |
| TGGT1_214140 | hypothetical protein | associates with AP2IX4/MORC complex | S205 and S212 | | |
| TGGT1_309250 | hypothetical protein | associates with GCN5b complex | PMID: 31381949 and 24391497 | | |

Endogenous tagging of SPARK and SPARKEL in the same parasite strain revealed punctate cytosolic staining for each protein (*Figure 1F*); however, the low expression level of SPARKEL precluded robust colocalization. Nevertheless, strong mass spectrometry evidence for the physical interaction between SPARK and SPARKEL motivated further analysis of their functional relationship.

## SPARK and SPARKEL depletion phenocopy at multiple steps in the lytic cycle

To determine the regulatory interaction between SPARK and SPARKEL, we first generated a SPARKEL conditional knockdown allele by inserting V5, HaloTag, mini auxin-inducible degron (mAID), and Ty epitopes at the endogenous SPARKEL C terminus. In this strain, we tagged SPARK with a C-terminal V5-mCherry-HA. Parasites treated with auxin (indole-3-acetic acid; IAA) to deplete SPARKEL failed to form plaques in host cell monolayers (*Figure 2A*). As indicated by immunoblot, IAA treatment led to a reduction in detectable SPARKEL signal within 1 hr, as well as co-depletion of SPARK on a similar time scale (*Figure 2B*). We next generated a strain expressing SPARK-V5-mAID-HA. Similarly, SPARK levels were reduced within 1 hr of IAA addition (*Figure 2—figure supplement 1*), as reported for a similar strain previously (*Smith et al., 2022*). In this genetic background, we tagged SPARKEL endogenously with a C-terminal V5-mNG-Ty. IAA treatment reduced SPARK levels and led to a reduction in SPARKEL abundance (*Figure 2C* and *Figure 2—figure supplement 1*). Together, these results suggest that conditional knockdown via the AID system leads to co-depletion of SPARK and SPARKEL. However, SPARK and SPARKEL abundances are low and approach the limit of detection. We could only detect each protein by the V5 epitope. Although our experiments included single-tagged controls, we cannot formally eliminate the possibility that SPARK-AID yields degradation products that run at the expected molecular weight of SPARKEL. More sensitive methods, such as targeted mass spectrometry, may be required to measure the absolute abundance and stoichiometries of SPARK and SPARKEL.

To assess the impact of complex knockdown on parasite replication and transitions between intracellular and extracellular environments, we treated SPARKEL-AID and SPARK-AID parasites with IAA, and performed replication, invasion, and egress assays. SPARKEL-AID parasites exhibited normal replication kinetics after 24 hr of IAA treatment (*Figure 2D*). Invasion and egress efficiency of the AID-tagged strains were reduced after parasites were treated with IAA for 24 hr (*Figure 2E and F*), as previously reported for SPARK (*Smith et al., 2022*). Intracellular calcium measurements of GCaMP-expressing parasites revealed a similar trend, with SPARKEL- and SPARK-depleted cells exhibiting reduced calcium mobilization after 24 hr of IAA treatment when stimulated with zaprinast, a compound that stimulates calcium release from parasite intracellular stores (*Brown et al., 2016*; *Sidik et al., 2016a*; *Figure 2G–H*, and *Figure 2—figure supplement 1*).

## SPARK and SPARKEL depletion phenotypes develop over time

SPARK was previously identified in a genetic screen that distinguished acute and delayed death phenotypes in a pooled population of parasites (*Smith et al., 2022*). As parasites with conditional SPARK alleles were assessed only in the context of delayed death phenotypes—that is, after an entire lytic cycle with IAA treatment—we characterized the ability of SPARK- and SPARKEL-AID parasites to invade and egress after 3 hr of IAA, which is shorter than the *T. gondii* cell cycle. In marked contrast

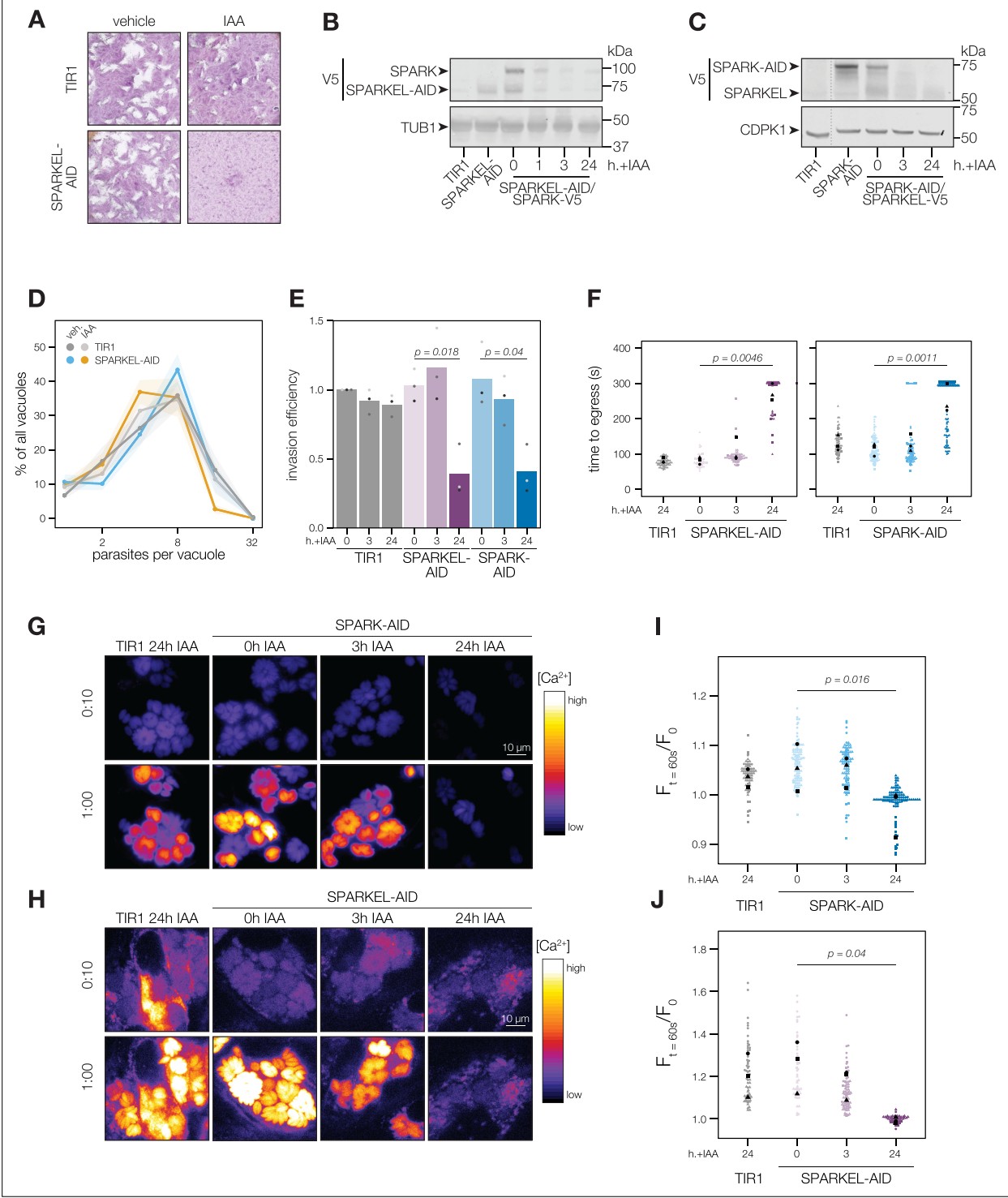

**Figure 2.** SPARKEL depletion phenocopies the loss of SPARK. (**A**) Plaque assays of 500 TIR1 or SPARKEL-AID parasites inoculated onto host cell monolayers and allowed to undergo repeated cycles of invasion, replication, and lysis for 7 days in media with or without 500 µM IAA. (**B**) Immunoblot of parasites expressing SPARKEL-V5-AID and SPARK-V5 after the indicated h with IAA. TUB1 serves as a loading control. (**C**) Immunoblot of parasites expressing SPARK-V5-AID and SPARKEL-V5 after the indicated h with IAA. CDPK1 serves as loading control. (**D**) The number of parasites per vacuole measured for SPARKEL-AID and the parental strain after 24 hr of 500 µM IAA treatment. Mean counts (n=8) are expressed as a percentage of all vacuoles counted. SEM is shown as shaded area. No comparisons yielded significant p-values using ANOVA and Tukey's test. (**E**) Invasion assays SPARKEL-AID, SPARK-AID or TIR1 parental strains treated with IAA or vehicle for 3 or 24 hr. Parasites were incubated on host cells for 20 min prior to differential staining of intracellular and extracellular parasites. Parasite numbers were normalized to host cell nuclei for each field. Different shapes

*Figure 2 continued on next page*

*Figure 2 continued*

correspond to means of n=3 biological replicates. For clarity, only significant comparisons (Welch's one-sided *t*-test) are shown. (**F**) The time to egress of individual intracellular vacuoles following zaprinast treatment. Points correspond to different vacuoles; shapes to different biological replicates (n = 3). Black shapes are the mean for each replicate. p-Values were calculated from a one-tailed *t*-test. (**G, H**) Selected frames from live video microscopy of zaprinast-treated SPARK-AID and SPARKEL-AID parasites, respectively, expressing the calcium indicator GCaMP6f, and the corresponding parental strain, 25 hr after infection and with the indicated IAA treatment period. See also *Figure 2—video 1* and *Figure 2—video 2*. (**I, J**) Relative GCaMP fluorescence of SPARK-AID or SPARKEL-AID vacuoles, respectively, 60 s following zaprinast treatment. Points correspond to different vacuoles; shapes to different biological replicates (n = 3). Black shapes are the mean for each replicate. p-Values were calculated from a one-tailed *t*-test.

The online version of this article includes the following video, source data, and figure supplement(s) for figure 2:

**Source data 1.** This file contains source data that was used to generate the blot in *Figure 2B*.

**Source data 2.** This file contains source data that was used to generate the blot in *Figure 2B*.

**Source data 3.** This file contains source data that was used to generate the blot in *Figure 2C*.

**Source data 4.** This file contains source data that was used to generate the blot in *Figure 2C*.

**Figure supplement 1.** Extended data related to *Figure 2*.

**Figure supplement 1—source data 1.** This file contains source data that was used to generate the blot in *Figure 2—figure supplement 1B*.

**Figure supplement 1—source data 2.** This file contains source data that was used to generate the blot in *Figure 2—figure supplement 1B*.

**Figure 2—video 1.** Representative image series of SPARKEL-AID or TIR1-expresing parental parasites expressing the genetically encoded calcium indicator GcaMP following stimulation with 500 µmM zaprinast after the indicated period of IAA treatment.

https://elifesciences.org/articles/93877/figures#fig2video1

**Figure 2—video 2.** Representative image series of SPARK-AID or TIR1-expresing parental parasites expressing the genetically encoded calcium indicator GcaMP following stimulation with 500 µmM zaprinast after the indicated period of IAA treatment.

https://elifesciences.org/articles/93877/figures#fig2video2

to the 24-hr-treatment regime, SPARKEL- and SPARK-AID parasites exhibited no deficits in invasion, egress, or calcium mobilization after 3 hr of depletion (*Figure 2E–J*). Thus, in all assays performed, mutants lacking SPARK and SPARKEL are phenotypically identical, likely due to reciprocal co-depletion, but such phenotypes develop over time.

## SPARK and SPARKEL depletion leads to AGC kinase down-regulation and up-regulation of bradyzoite-stage proteins

Having verified the interaction between SPARK and SPARKEL, we next investigated pathways regulated by the complex. Initially, we performed a quantitative proteomics experiment following 24 hr of SPARK depletion—the previously described depletion window (*Smith et al., 2022*). Extended SPARK depletion led to down-regulation of PKG and both the catalytic and regulatory subunits of PKA (*Figure 3A* and *Table 1*). We also observed up-regulation of bradyzoite-specific genes upon SPARK depletion (*Figure 3A* and *Table 1*), including MAG1, BRP1, TGME49_208740, CST1, CST10, MIC13, and BFD2. We were unable to measure SPARK or SPARKEL abundances in this proteome.

To determine the critical window of SPARK and SPARKEL function, we performed quantitative proteomics with tandem mass tag multiplexing that included samples with 0, 3, 8, and 24 hr of SPARK or SPARKEL depletion harvested 32 hr post-infection. The experiments included internal TIR1 parental strain controls to account for possible basal downregulation arising from the degron (*Figure 3B*). We detected 3,333 and 3,880 proteins with quantification values in the SPARK and SPARKEL depletion time courses, respectively (*Supplementary file 3*). PKA C1, PKA R, and PKG protein abundances only began to drop after 8 hr of IAA treatment (*Figure 3C*). Globally, these kinase subunits were the most down-regulated proteins following 24 hr of either SPARK or SPARKEL depletion (*Figure 3—figure supplement 1*). Bradyzoite proteins up-regulated upon SPARK depletion primarily increased between 8 and 24 hr of IAA treatment (*Figure 3D*); the same proteins increased in abundance with 24 hr of SPARKEL depletion (*Figure 3D*). We were unable to measure SPARK or SPARKEL abundances in either proteome. Nevertheless, depletion of the SPARK-SPARKEL complex consistently leads to up-regulation of proteins associated with the bradyzoite stage of development, as well as down-regulation of PKG and the PKA C1 complex.

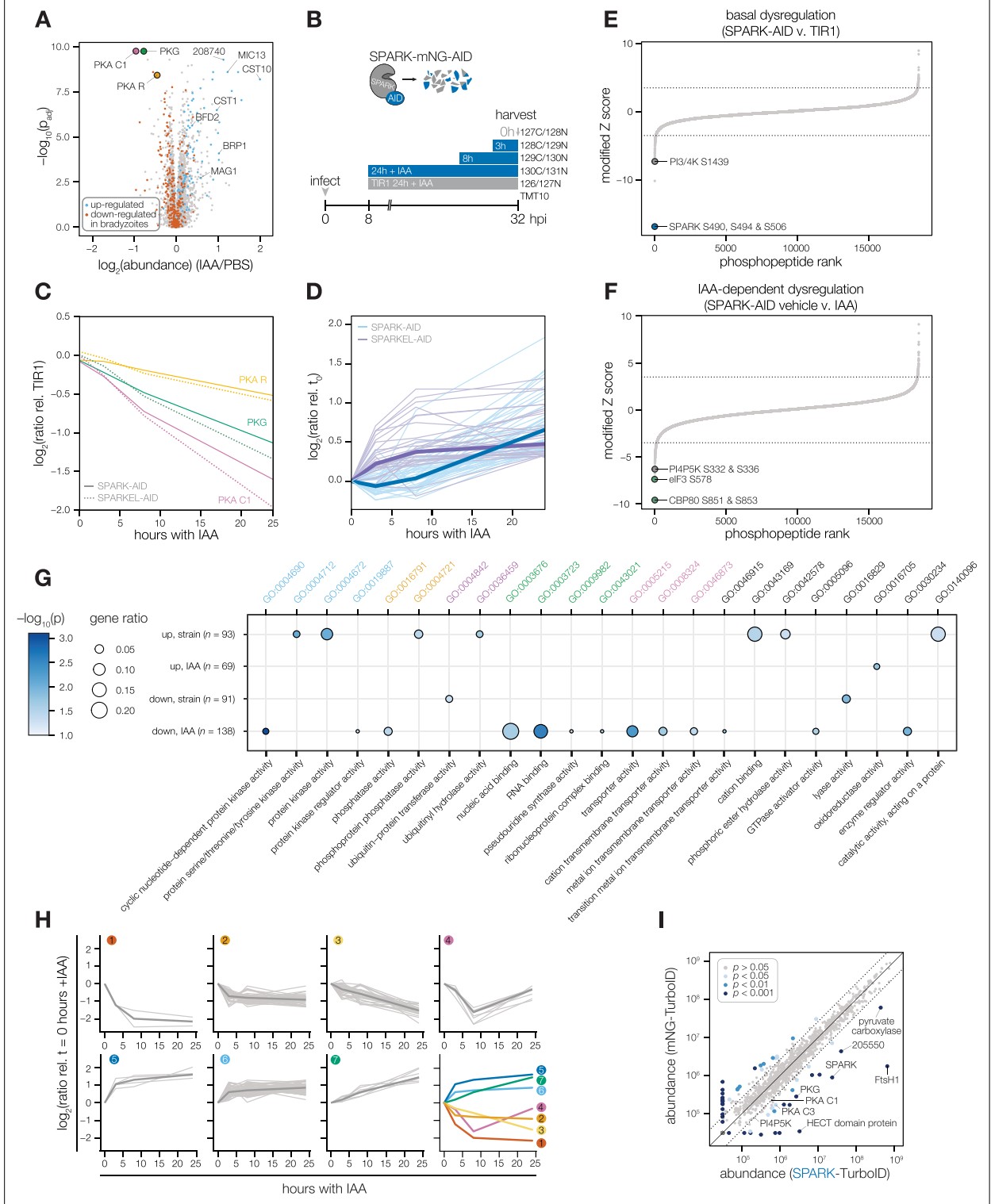

**Figure 3.** Depletion of SPARK or SPARKEL leads to downregulation of AGC kinases and upregulation of chronic-stage markers. (**A**) Volcano plot displaying the protein abundance ratios of SPARK-AID parasites treated with IAA or vehicle for 24 hr and adjusted p-values for n = 2 biological replicates. Proteins identified as up- or down-regulated in parasites overexpressing the driver of differentiation (BFD1) (**Waldman et al., 2020**) are shown in blue and vermilion, respectively. In total, 4474 proteins were quantified, 3847 of which registered more than one peptide. (**B**) Schematic of the phosphoproteomics experiment following SPARK depletion. Intracellular parasites expressing SPARK-AID were treated with 500 µM IAA for 24, 8, 3, or 0 hr and were harvested at 32 hpi simultaneously with the TIR1 parental strain as a control. Samples were multiplexed with tandem mass tags (TMT). The same experimental design was used for SPARKEL-AID proteomics. Each experiment included two biological replicates. (**C**) Average protein abundances

*Figure 3 continued on next page*

*Figure 3 continued*

of PKG, PKA-R, and PKA-C1 relative to the TIR1 parental strain after the indicated period of SPARK (thick lines) or SPARKEL (dotted lines) depletion. (**D**) Average protein abundances of up-regulated bradyzoite genes relative to the TIR1 parental strain after the indicated period of SPARK (blue lines) or SPARKEL (purple lines) depletion. Up-regulated bradyzoite proteins were defined as those significantly increased in parasites overexpressing BFD1 (*Waldman et al., 2020*) and two modified Z-scores above the median in the SPARK depletion proteome. Rank-ordered plots of (**E**) Phosphopeptide basal dysregulation score (peptide abundance in the SPARK-AID strain without IAA treatment relative to the TIR1 parental strain) or (**F**) IAA-dependent score (summed peptide ratios relative to the SPARK-AID $t_0$ peptide abundance). Dotted lines correspond to 3.5 modified Z scores. Colored points correspond to phosphosites discussed in the main text. (**G**) Gene ontology (GO) enrichment of phosphoproteins exhibiting SPARK-dependent regulation. Gene ratio is the proportion of proteins with the indicated GO term divided by the total number of proteins. Significance was determined with a hypergeometric test; only GO terms with p<0.05 are shown. Redundant GO terms were removed. Categories discussed in the main text are highlighted with colored text. (**H**) Gaussian mixture modeling of SPARK-dependent peptides identified by more than one PSM revealed seven kinetically resolved clusters. Individual peptides or the median ratios in each cluster are depicted by light and dark gray lines, respectively. Clusters are numbered according to their discussion in the main text. (**I**) Protein abundances following biotinylation and streptavidin enrichment of samples derived from parasites expressing mNG- or SPARK-TurboID fusion constructs. A pseudocount of $10^{4.5}$ was assigned to proteins identified in only one sample. Point colors correspond to significance thresholds for n = 2 biological replicates. Dotted lines correspond to one median absolute deviation.

The online version of this article includes the following figure supplement(s) for figure 3:

**Figure supplement 1.** Extended analysis of the SPARK-AID depletion phosphoproteome.

## A depletion phosphoproteome implicates SPARK in the signaling, gene regulation, and metabolic states of the lytic cycle

To determine SPARK-dependent phosphoregulation, we enriched and analyzed phosphopeptides from the depletion proteome. We identified 18,518 phosphopeptides with quantification values, of which 8867 were measured with more than one peptide-spectrum match (PSM). Principal component analysis (PCA) separated the SPARK-AID samples treated for 24 hr with IAA from all other samples, while closely clustering biological replicates (*Figure 3—figure supplement 1*). Despite the low abundance of SPARK and SPARKEL, individual phosphopeptides were detected for each protein. Likely due to basal degradation, the abundance of the SPARK phosphopeptide was ten-fold lower in SPARK-AID samples compared to the TIR1 control such that further depletion could not be measured (*Figure 3—figure supplement 1*). The SPARKEL phosphopeptide abundance was also reduced relative to the TIR1 control sample; however, it decreased further between 3 and 8 hr of IAA treatment (*Figure 3—figure supplement 1*). Because no other SPARKEL phosphopeptides were detected, it is not possible to determine whether this decrease is phosphosite-specific or due to a general decrease in SPARKEL protein abundance.

SPARK-regulated phosphopeptides may manifest in a strain-dependent or IAA-dependent manner. To account for basal downregulation, we ranked phosphopeptides by their $\log_2$-ratios of altered abundances in the SPARK-AID strain relative to the TIR1 parental strain (*Figure 3E*). When a cutoff of 3.5 modified Z-scores was used, 91 phosphopeptides (44 with >1 PSMs) and 69 phosphopeptides (19 with >1 PSMs) were down- or up-regulated, respectively. The aforementioned phosphosite belonging to SPARK was the most strongly down-regulated phosphopeptide by this metric. One phosphopeptide belonging to a phosphatidylinositol 3- and 4-kinase (PI3,4K) was also strongly down-regulated (*Table 1*); however, the protein abundance of this enzyme was not quantified. To identify phosphopeptides most altered by IAA treatment, and hence SPARK depletion, we summed and ranked phosphopeptides by $\log_2$-ratios of abundances in the SPARK-AID strain relative to the untreated samples (*Figure 3F*). By this metric, 138 phosphopeptides (51 with >1 PSMs) and 93 phosphopeptides (36 with >1 PSMs) were down- or up-regulated by more than 3.5 modified Z-scores, respectively. Phosphoproteins belonging to this category are discussed below.

To uncover the molecular pathways regulated by SPARK, we performed enrichment analysis of the phosphopeptides exhibiting the highest degree of altered abundance upon SPARK depletion (*Figure 3G*). Changes in SPARK function likely alter post-translational modification (PTM) cascades both directly and indirectly. SPARK depletion reduced the abundance of phosphopeptides belonging to cyclic nucleotide-regulated kinases, which was in part driven by a reduction in kinase abundance. SPARK depletion also coincided with a reduction in phosphosites on several phosphatases (*Table 1*), including BCC6, PPM1, and an uncharacterized dual-specificity phosphatase. By contrast, phosphorylation of other kinases and phosphatases increased upon SPARK depletion, which may arise as an indirect consequence of SPARK function. Upregulated phosphosites were found on pyruvate

dehydrogenase and STE kinase, CDPK3, cyclin2-related protein, and the kinases TGGT1_204280 and TGGT1_239885; as well as the protein phosphatases PPM2B, calcineurin, and EFPP. Phosphosites belonging to proteins involved in ubiquitin transfer and hydrolysis also increased in abundance upon SPARK depletion. Examples of this class included UBP1, USP39, an RBR E3 ligase, CCR4-NOT subunit, and UBR box E3 ligase. To a lesser degree, phosphosites on ubiquitin ligases were also down-regulated following SPARK depletion—for example, a HECT E3 and a putative E2 enzyme. Thus, SPARK function has broad implications for PTM cascades.

Additional functional categories enriched for SPARK-dependent regulation included nucleic acid–binding proteins (*Figure 3G* and *Table 1*). A number of down-regulated phosphosites belong to proteins involved in DNA accessibility and transcription—for example, TFIIS, SET1, NOC3p, and *Tg*SRCAP. RNA modification enzymes also exhibited phosphoregulation, such as two PUS proteins, PRP8, SLU7, a FHA protein, three KH proteins, and a DCP2 homolog. Other regulated phosphoproteins may function in mRNA cap or untranslated region (UTR) binding, such as *Tg*PUF1, a MIF4G protein, CBP80, eIF3, and GCN1. Some of these proteins have been linked to expression changes during stage conversion, with PUF1 and SRCAP levels having been noted to increase during stress (*Joyce et al., 2013*; *Liu et al., 2014*; *Sullivan et al., 2003*). Moreover, CBP80, eIF3, and the KH protein TGGT1_216670 co-purified with the Alba complex (*Gissot et al., 2013*), which has been implicated in bradyzoite differentiation. Regulation of these proteins thus may be linked to the elevated expression of bradyzoite-stage proteins upon SPARK depletion.

Several putative transporters also exhibited decreased phosphopeptide abundance upon SPARK knockdown (*Figure 3G*). Some of these phosphoproteins channel inorganic cations, such as *Tg*ZnT (*Chasen et al., 2019*). Others likely transport small organic molecules—for example, a nucleoside and two ABC transporters. In aggregate, these analyses implicate SPARK in numerous cellular processes, including signaling, gene regulation, and metabolic exchange.

## Clustering of phosphopeptide kinetics identifies seven response signatures

We sought to resolve the SPARK phosphoproteome into kinetically distinct clusters. Gaussian mixture-model algorithms heuristically clustered phosphopeptides identified by more than one PSM into seven classes on the basis of their response signatures following SPARK depletion (*Figure 3H*). Two phosphopeptides in Cluster 1 were distinguished by their rapid and sustained depletion and corresponded to CBP80 and eIF3, both of which function in translation. The second cluster of phosphopeptides was largely down-regulated by 3 hr of IAA treatment and lacked dynamics following this window. PKA and PKG phosphopeptides belonged to Cluster 3, which decreased only after 8 hr of IAA treatment. The final cluster of depleted phosphopeptides, Cluster 4, only exhibits down-regulation at 8 hr of IAA treatment.

Up-regulated phosphopeptides mirrored the dynamics of the down-regulated clusters. Cluster 5 phosphopeptides were elevated within 3 hr of IAA treatment and continued to increase thereafter. Cluster 6 was up-regulated by 3 hr of IAA treatment and was stable thereafter. Cluster 7 increased gradually and predominantly after 8 hr of IAA treatment. Upregulated phosphosites may arise from a number of mechanisms following SPARK depletion. For example, SPARK activity may inhibit downstream kinases or activate downstream phosphatases, sterically block access of serines/threonines to other enzymes, or give rise to crosstalk between PTMs.

Because non-phosphopeptide and phosphopeptide abundances were quantified in different mass spectrometry experiments, it is challenging to compare the rates of phosphopeptide and parent protein abundance changes, especially when phosphorylation status and protein stability are interconnected. In general, both PKA C1, PKA R, and PKG protein and phosphosite abundances decreased following SPARK depletion (*Figure 3—figure supplement 1*), as discussed below. We also observed down-regulation of phosphopeptide and protein abundances of a MIF4G domain protein. Approaches that take into account subcellular proximity were required to distinguish direct substrates from pathways downstream of SPARK function.

## Proximity labeling identifies putative SPARK targets in situ

To gain insight into proximal SPARK interactors in intracellular parasites, we tagged the endogenous C terminus of SPARK with a TurboID domain and carried out proximity labeling experiments, as previously

described for SPARKEL. We observed overlap in the phosphoproteomic and proximity labeling experiments. PKG was enriched (*Figure 3I* and *Supplementary file 3*), while PKA C1 narrowly missed our cutoffs. A putative HECT E3 ubiquitin ligase (TGGT1_280660) with two phosphosites down-regulated in the SPARK-dependent phosphoproteome was a candidate interactor. We also detected enrichment of PKA C3. PKA C3 is a coccidian-specific ortholog of the apicomplexan PKA catalytic subunit *Sugi et al., 2016*; however, no PKA C3 phosphopeptides were quantified in our depletion phosphoproteome experiments, likely due to the low abundance of the protein. SPARKEL was not quantified in the proximity labeling experiment, despite prior detection of the reciprocal interaction (*Figure 1E*). These integrated proteomics approaches hint that the phenotypes associated with SPARK depletion may arise from the altered activity of AGC kinases regulated by SPARK.

## SPARK functionally interacts with PKA and PKG

The SPARK depletion time course phosphoproteome showed a reduction in the abundance of PKA C1 T190 and T341, which are located in the activation loop and C-terminal tail, respectively (*Figure 4A*). Several phosphosites residing in the N terminus of PKA R (e.g. S17, S27, and S94) also decreased following SPARK depletion (*Figure 4B*). Transcript levels for PKA C1, and two other AGC kinases tested, remained unchanged following 3 or 24 hr of SPARK depletion (*Figure 4—figure supplement 1*). To confirm the depletion kinetics of PKA, we tagged the kinase subunits with fluorophores in the SPARK-AID parasite strain (*Figure 4C*). Prolonged SPARK depletion (24 hr) reduced PKA C1 and PKA R immunofluorescence signals in intracellular parasites (*Figure 4D*). To monitor kinase levels kinetically, we performed flow cytometry at several time points following the addition of IAA (*Figure 4E*). PKA C1 and PKA R fluorophore intensity decreased predominantly after 8 hr of IAA treatment, consistent with proteomic results. In parallel experiments, we tagged PKA C1 and PKA R in the SPARKEL-AID strain, revealing a comparable loss of the two markers following SPARKEL depletion (*Figure 4F*).

We considered the possibility that the SPARK depletion phosphoproteome represents a convolution of phosphoproteomes arising from broader AGC kinase down-regulation. We systematically tested this possibility by generating phosphoproteomes of parasites depleted of PKA R and PKG (*Supplementary file 4*). As association with PKA R inhibits PKA C1 activity (*Jia et al., 2017*), we reasoned that phosphosites up-regulated after a short window of PKA R depletion represent candidate PKA C1 targets. Similarly, phosphosites down-regulated after PKG depletion represent candidate PKG substrates (*Brown et al., 2017*). Phosphopeptides above two modified Z-scores were considered candidate targets of each kinase.

Compared to the bulk phosphoproteome, the PKA C1-dependent phosphopeptides significantly decreased in abundance between 8 and 24 hr of SPARK depletion (*Figure 4G*). When we applied the same analysis to a similarly acquired PP1-dependent phosphoproteome (*Herneisen et al., 2022*), we found no significant relationship to the SPARK phosphoproteome (*Figure 4—figure supplement 1*), suggesting that the association with PKA C1 is specific. Several of the overlapping SPARK and PKA C1–dependent phosphoproteins localized to the parasite periphery and apex (*Figure 4H*), for example, IMC25, AC13, CIP1, CIP2, and AC3. Others are putatively involved in RNA-binding functions, such as eIF4E and two KH proteins, including a component of the METTL3/METTL14 core (*Farhat et al., 2021*). We also observed regulation of phosphoproteins involved in ubiquitin transfer, including the aforementioned E2 and HECT E3 proteins (*Table 1*). Regulation of these proteins may account for both the kinetic- and replicative-phase phenotypes attributed to perturbed PKA C1 activity (*Jia et al., 2017*; *Uboldi et al., 2018*).

We carried out related experiments to validate that SPARK regulates PKG function. The SPARK depletion time course phosphoproteome showed a reduction in the abundance of several phosphosites residing in the N terminus of PKG as well as T838, which corresponds to the activation loop (*Figure 5A*). By contrast, S105 did not greatly decrease, and S40 abundance increased slightly. Attempts to tag PKG with a fluorophore in a functional SPARK-AID background were unsuccessful. Global genetic screens have indicated that parasites with hypomorphic PKG—as can arise through endogenous tagging—are rapidly outcompeted by wildtype parasites (*Fang et al., 2018*; *Smith et al., 2022*). Therefore, we instead relied on chemical-genetic interactions to validate the relationship between PKG and SPARK. We reasoned that parasites with down-regulated PKG activity might be sensitized to the specific PKG inhibitor Compound 1 (*Donald et al., 2002*). Previous studies had shown that SPARK is dispensable for A23187-induced egress *Smith et al., 2022*; however, PKG function is

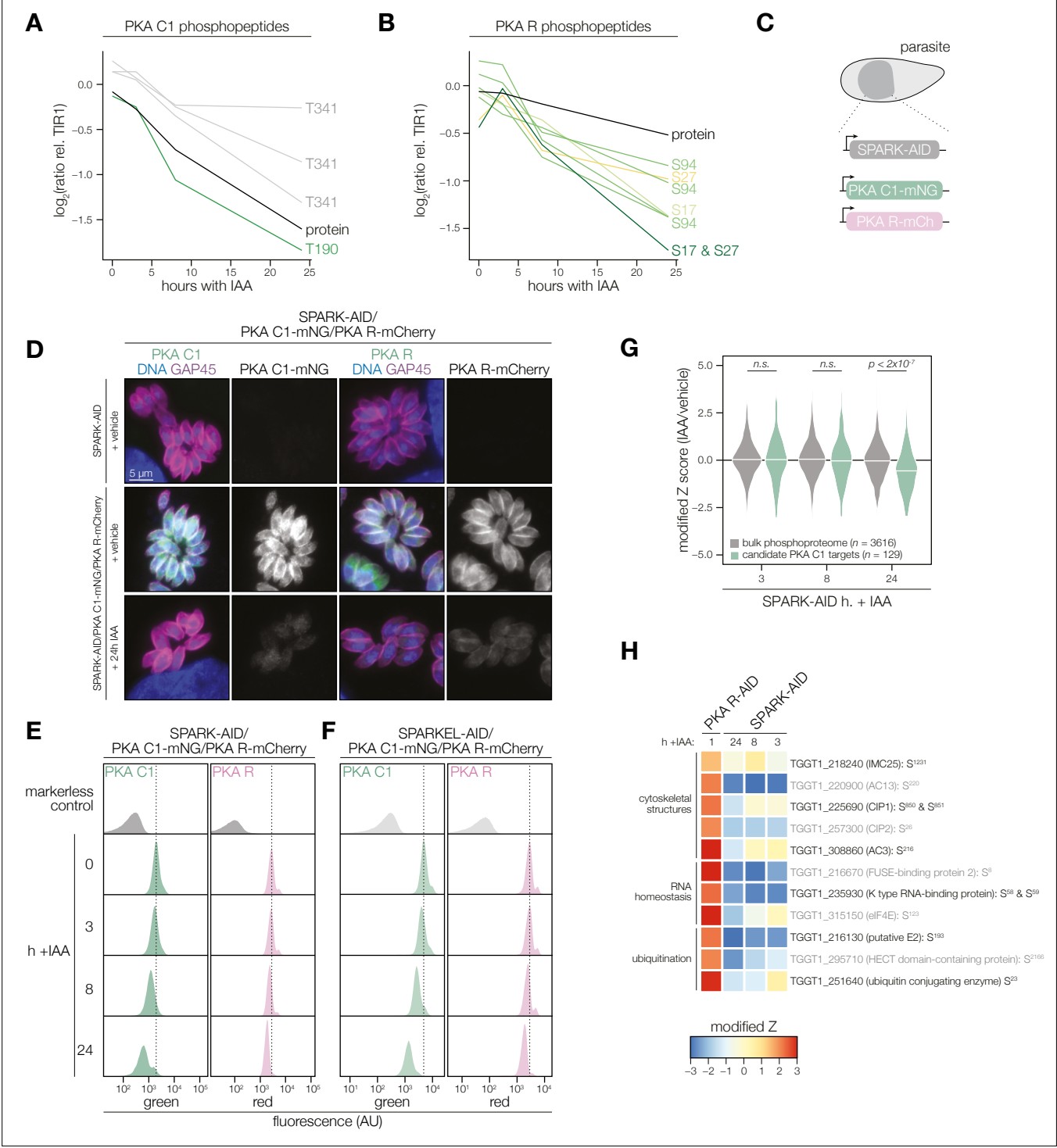

**Figure 4.** PKA C1 levels are down-regulated upon SPARK depletion. (**A, B**) Average protein and phosphopeptide abundances of PKA C1 (**A**) and PKA R (**B**) following SPARK depletion. (**C**) Schematic of the genetic strategy used to monitor PKA C1 and PKA R abundances following SPARK (or SPARKEL) down-regulation with IAA. (**D**) Immunofluorescence microscopy of parasites expressing SPARK-AID, PKA C1-mNG, and PKA R-mCherry after 0 or 24 hr of IAA treatment to degrade SPARK. Parasites and nuclei were stained with GAP45 and Hoechst 33342, respectively. GAP45 staining and mNG or mCherry staining were normalized to vehicle-treated tagged samples. (**E, F**) Flow cytometry analysis of parasites expressing PKA C1-mNG, PKA R-mCherry, and SPARK-AID or SPARKEL-AID, respectively, after the indicated period of IAA treatment. The dotted line centers the mode of the vehicle-treated sample. Traces were normalized by unit area. (**G**) Violin plots displaying the distribution of phosphopeptide abundance values following SPARK depletion. The distributions of candidate PKA C1 targets, as defined in the text and methods, are shown in green. The distributions and p-values (KS

*Figure 4 continued on next page*

*Figure 4 continued*

test) were derived from the overlapping subset of phosphopeptides identified in each dataset. (**H**) Heat map of the abundance ratios of candidate PKA C1 targets following SPARK depletion. PKA R depletion results in up-regulation of PKA C1, and candidate PKA C1 targets therefore have positive abundance ratios following PKA R down-regulation.

The online version of this article includes the following figure supplement(s) for figure 4:

**Figure supplement 1.** Transcriptional and phosphoproteomic effects of SPARK depletion.

required for ionophore-induced egress (*Brown et al., 2017*; *Lourido et al., 2012*). We performed egress assays using parasites depleted of SPARK for 24 hr and treated with different concentrations of Compound 1. As anticipated, SPARK depletion sensitized parasites to inhibition with Compound 1, as assessed by A23187-induced egress (*Figure 5B–C*), suggesting that PKG activity is decreased in the absence of SPARK.

As with PKA, we quantified the behavior of putative PKG substrates in the SPARK-AID phosphoproteome. Compared to the bulk phosphoproteome, the PKG-dependent phosphopeptides significantly decreased in abundance between 8 and 24 h of SPARK depletion (*Figure 5D*). Several overlapping SPARK- and PKG-dependent phosphoproteins localized to the parasite periphery, including an MFS transporter, *Tg*A1, MyoG, IMC18, IMC28, and IMC41. Other PKG/SPARK-dependent sites were found in the apical proteins AAP3, AC5/TLAP3, AC13, and SPM1; while several other proteins localized to the ER, including two TB2/DP1, HVA22 family proteins, a TPR protein, and an ABC transporter (*Figure 5E*). Others function in the secretory or protein trafficking pathways, such as Sec7, COG6, *Tg*TBC10, and *Tg*DHHC1. We also observed down-regulation of phosphosites belonging to proteins involved in ubiquitin transfer, such as an E2 enzyme and a ubiquitin family protein with a C-terminal

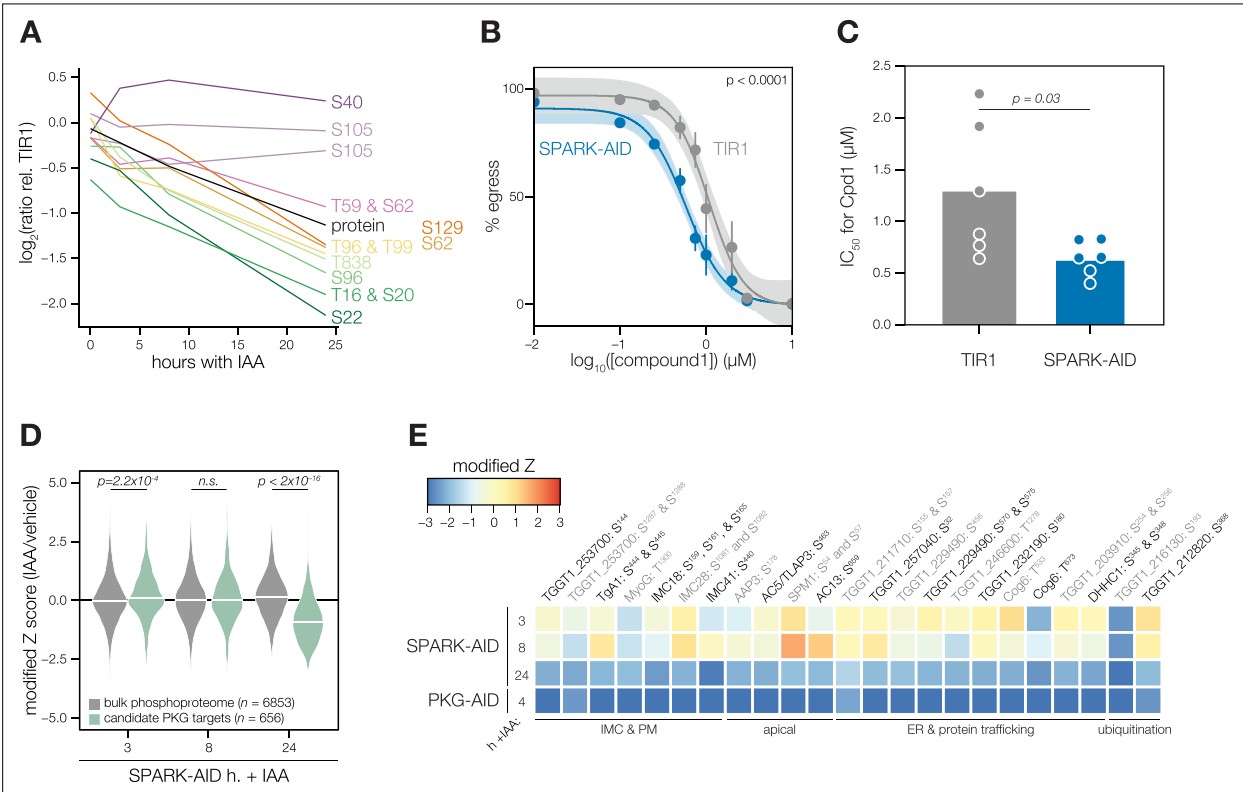

**Figure 5.** Characterization of PKG function during SPARK depletion. (**A**) Protein and phosphopeptide abundances of PKG following SPARK depletion. (**B**) A23187-stimulated egress assays performed at different concentrations of compound 1 after TIR1 and SPARK-AID parasites had been treated with IAA for 24 hr. Curves were fit to the average values of six replicates and were compared with an extra sum of squares F test. (**C**) The IC50 values of each strain for compound 1; each point represents a biological replicate (n = 6). Significance was assessed with a two-tailed t-test. (**D**) Violin plots displaying the distribution of phosphopeptide abundance values following SPARK depletion. The distributions of candidate PKG targets are shown in green. The distributions and p-values (KS test) were derived from the overlapping subset of phosphopeptides identified in each dataset. (**E**) Heat map of the abundance ratios of candidate PKG targets following SPARK depletion.

extension. The precise pathways regulated by *Tg*PKG remain to be defined; however, recent studies triggering PKG activity in *T. gondii* (*Chan et al., 2023*; *Herneisen et al., 2022*) or related *Plasmodium* spp. (*Alam et al., 2015*; *Balestra et al., 2021*; *Brochet et al., 2014*) suggest that PKG substrates relevant for egress reside at the parasite periphery and ER, as observed for the overlapping PKG and SPARK targets here.

## Depletion of SPARK, SPARKEL, or PKA C3 promotes chronic differentiation

Several pieces of evidence suggested that SPARK and SPARKEL depletion may prompt differentiation from the tachyzoite to bradyzoite state through dysregulation of another AGC kinase, PKA C3. First, down-regulation of SPARK and SPARKEL coincided with up-regulation of bradyzoite markers (*Figure 3A and D*). Second, SPARK proximity labeling resulted in enrichment of PKA C3 (*Figure 3I*)—a negative regulator of the bradyzoite stage (*Sugi et al., 2016*) and, to date, the only AGC kinase characterized with a function in differentiation (*Augusto et al., 2021*). Prolonged SPARK and SPARKEL depletion indeed elevated differentiation under normal culture conditions, as measured by staining of the cyst walls with dolichos binding lectin, a characteristic of chronic-stage bradyzoite vacuoles (*Figure 6A–B*). N-terminal tagging of SPARKEL with AID similarly elevated differentiation upon IAA treatment (*Figure 6—figure supplement 1*). Alkaline media in combination with serum starvation and low carbon dioxide is commonly used to induce bradyzoite transformation in cell culture. SPARK and SPARKEL down-regulation enhanced differentiation under such conditions (*Figure 6A–B*).

To dissect the hypothesized PKA C3–dependent arm of SPARK regulation, we altered the PKA C3 genomic locus with a V5-mAID-mNG-Ty tagging payload. Addition of IAA to the culture medium resulted in down-regulation of PKA C3 levels to below the detection limit within one hour (*Figure 6C–D*). PKA C3 exhibited low expression levels in cytoplasmic puncta (*Figure 6C*), as previously reported (*Sugi et al., 2016*). We confirmed that prolonged PKA C3 depletion led to elevated differentiation rates under normal and alkaline-stress culture conditions (*Figure 6A–B*) and observed that the effect was similar to SPARK or SPARKEL knock-down. Given that SPARK also regulates PKG and PKA C1, we considered the possibility that differentiation may be linked to a block in motility. However, the inability to egress is not sufficient for differentiation, as conditional knockdown of PKG or CDPK1 (*Brown et al., 2017*; *Chan et al., 2023*; *Shortt et al., 2023*) did not enhance the frequency of differentiation markers (*Figure 6A–B*). We did not assess differentiation in PKA C1 knockdown parasites, as these mutants spontaneously egress from host cells (*Jia et al., 2017*; *Uboldi et al., 2018*).

The RH strain in which we generated mutants is reported to be refractory to differentiation into the bradyzoite form (*Dubey et al., 1999*), likely due to defects downstream of the pathways that lead to the initial synthesis of the cyst wall. We generated SPARKEL- and PKA C3-AID strains in the ME49 background (*Figure 6—figure supplement 1*), which readily undergoes differentiation in tissue culture and mice, and confirmed that depletion of either prompts spontaneous differentiation in tissue culture (*Figure 6E–F*). We did not assess alkaline-induced differentiation, as this stress treatment results in near-complete differentiation in the ME49 strain (*Waldman et al., 2020*), which complicates measurement of enhanced differentiation.

To assess whether excess differentiation caused by SPARK and PKA C3 depletion is dependent on a previously characterized transcriptional regulator of differentiation, BFD1 (*Waldman et al., 2020*), we replaced the *BFD1* CDS with a sortable dTomato cassette in the SPARK- and PKA C3-AID strains (*Figure 6—figure supplement 1*). The resulting SPARK- and PKA C3-AID/Δ*bfd1* mutants failed to undergo differentiation as measured by cyst wall staining (*Figure 6G–H*), suggesting that differentiation caused by depletion of these kinases depends on the BFD1 circuit. We additionally profiled transcriptomic remodeling in PKA C3-AID parasites that had been depleted of the kinase for 24 hr and compared these changes to the chronic-stage transcriptome of parasites overexpressing BFD1 (*Waldman et al., 2020*). We observed up-regulation of several chronic-stage proteins upon PKA C3 depletion (*Figure 6I*), including BAG1, CST1, CST10, MIC13, and *Tg*SPT2 (*Licon et al., 2023*). PKA C3 mRNA abundance was down-regulated, suggesting that tagging may alter steady-state transcript levels. Together, these results suggest that the bradyzoite stage conversion induced by knockdown of PKA C3 and SPARK proceeds through the canonical transcriptional pathway.

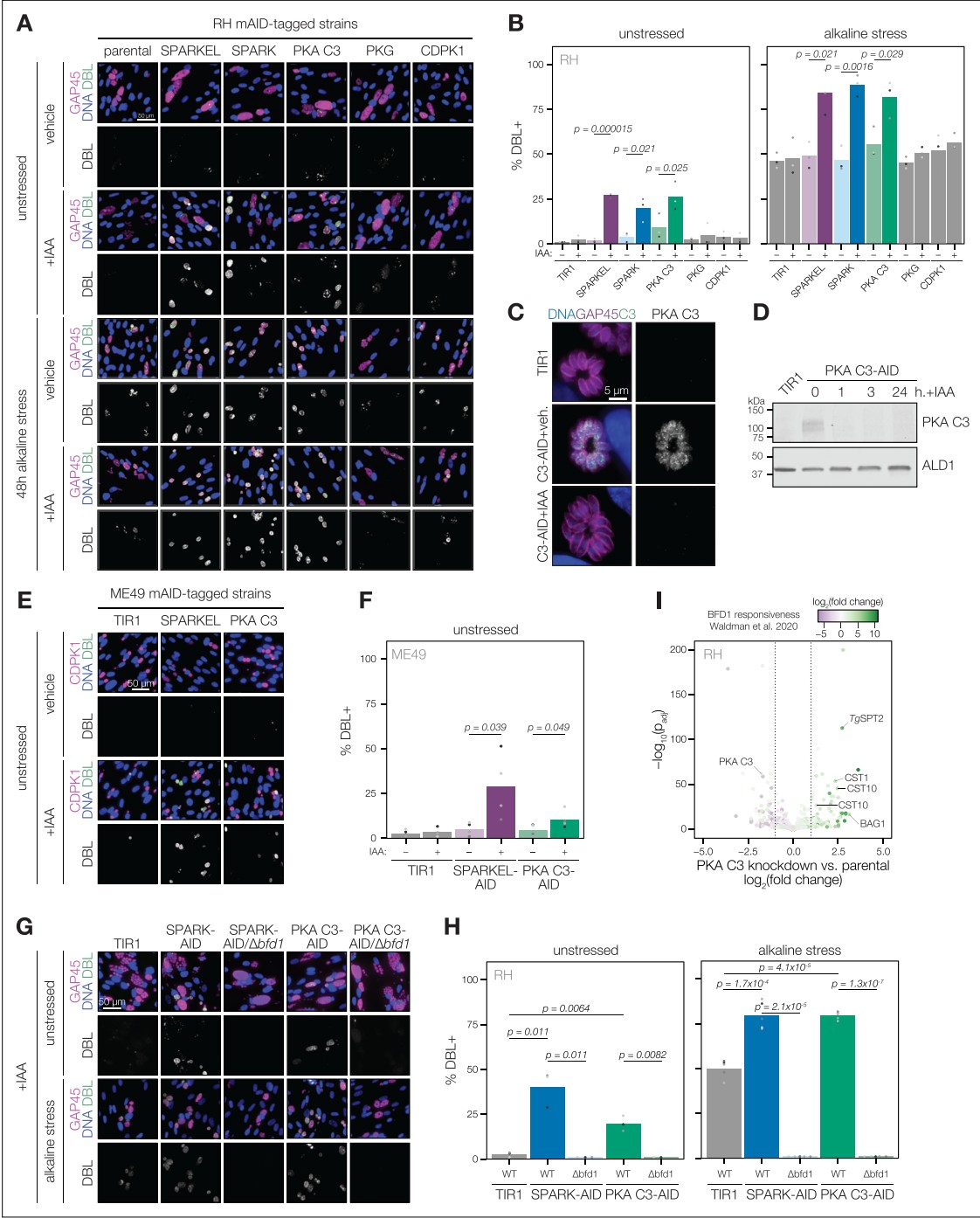

**Figure 6.** SPARK, SPARKEL, and PKA C3 are negative regulators of differentiation. (**A**) Immunofluorescence differentiation assays following knockdown of the indicated AID strains for 48 hr under standard growth conditions (unstressed) or alkaline stress. GAP45 was used to stain parasite vacuoles. Differentiated vacuoles were stained with biotinylated DBA/streptavidin-APC. Nuclei were stained with Hoechst. (**B**) Quantification of the number of DBL +vacuoles expressed as a percentage of the total stained vacuoles is shown for parasites grown under unstressed or stressed conditions. One-sided *t*-test, n=3 biological replicates. (**C**) Fixed, intracellular PKA C3-mNG-AID parasites visualized by immunofluorescence microscopy using the mNG epitope after 1 hr of vehicle or IAA treatment. The mNG signal was internally normalized to the TIR1 parental strain. (**D**) Immunoblot of PKA C3-AID parasites following the addition of vehicle or 500 μM IAA for 1, 3, or 24 hr. TIR1 was included as an untagged control. PKA C3-AID was detected with V5, and ALD1 was probed as a loading control. (**E**) Immunofluorescence differentiation assays following knockdown of the indicated ME49/AID-tagged strains under unstressed conditions for 48 hr. Staining was performed as described above, except CDPK1 was used as a parasite vacuole marker. (**F**) Quantification of the number of DBL +vacuoles expressed as a percentage of the total stained vacuoles in (**E**). Two-sided *t*-test, n=5 biological replicates. (**G**) Immunofluorescence differentiation assays of parasite strains with or without BFD1, depleted of PKA C3 or SPARK with 500 μM IAA and

*Figure 6 continued on next page*

*Figure 6 continued*

grown under unstressed conditions or alkaline stress for 48 hr. (**H**) The percentage of DBL +vacuoles corresponding to (**G**). One-sided *t*-test. Three biological replicates were quantified under unstressed conditions; five replicates were performed under alkaline stress (**I**) Effects of 24 hr of PKA C3 knockdown on the transcriptome relative to the untagged strain. Dotted lines correspond to an absolute log$_2$ change of 1. Genes significantly affected by BFD1 overexpression p-value <0.001 as previously defined (*Waldman et al., 2020*) are colored according to log$_2$ change in the chronic-stage transcriptome. Highlighted points are discussed in the text. P-values were determined from a Wald test implemented in DESeq2.

The online version of this article includes the following source data and figure supplement(s) for figure 6:

**Source data 1.** This file contains source data that was used to generate the blot in *Figure 6D*.

**Source data 2.** This file contains source data that was used to generate the blot in *Figure 6D*.

**Figure supplement 1.** Extended data related to *Figure 6*.

## PKA C3 interacts with the SPARK complex

Several AGC kinases, including PKA C3, share the activation loop motif TLC/VGTxxY, which displayed SPARK-dependent phosphorylation in PKA C1 and PKG. PKA C3 was not detected in our SPARK depletion proteomics experiments (*Figure 3*). To determine whether SPARK and PKA C3 interact, we immunopurified PKA C3 lysates, using the mNG epitope as a handle. The immunoprecipitated PKA C3 was highly enriched for SPARK (*Figure 7A*), and to a lesser extent, other protein kinases including TGGT1_205550, SRPK, and AMPK subunit beta. The latter has recently been characterized as a key metabolic regulator that enhances differentiation when knocked out (*Yang et al., 2022*). To a lesser extent, SPARKEL was also significantly co-purified with PKA C3 (*Supplementary file 1*). Our results are consistent with other reports that PKA C3 does not interact with the canonical PKA regulatory subunit in *T. gondii* (*Jia et al., 2017*; *Uboldi et al., 2018*) and, in conjunction with TurboID data presented in *Figure 3I*, provide support for a strong physical interaction between SPARK and PKA C3.

## SPARK down-regulation does not reduce PKA C3 protein levels

To determine whether SPARK activity affects PKA C3 abundance, as for PKA C1 and PKG, we tagged the endogenous PKA C3 C terminus with an mNG fluorophore in the SPARK-AID background. After 24 hr of IAA treatment, PKA C3 levels were unaltered or slightly elevated, as measured by immunoblot (*Figure 7B–C*). Immunofluorescence and flow cytometry measurements of mNG intensity after 0, 3, 8, and 24 hr of IAA treatment revealed a trend towards increasing PKA C3 abundance, although this difference was not significant (*Figure 7D–F*). Thus, unlike PKA C1, PKA C3 does not exhibit down-regulation following SPARK depletion. We considered the possibility that SPARK function could instead alter PKA C3 activity.

## The PKA C3 phosphoproteome identifies candidates regulating the transition to the bradyzoite state

To pursue the functional consequences of PKA C3 inhibition, we performed a depletion phosphoproteomics time course experiment with the PKA C3-AID strain following 1, 3, 8, or 24 hr of IAA treatment (*Supplementary file 3*). PKA C3-dependent phosphopeptides were identified as for the SPARK depletion proteome (*Figure 7G*). The resulting proteome identified 4792 proteins and 12,721 phosphopeptides, of which 9576 were quantified with more than one PSM. PCA separated the 24 hr IAA samples from all others (*Figure 7—figure supplement 1*). Similar to the SPARK depletion proteome, several of the up-regulated proteins were reported to be transcriptional signatures of the bradyzoite stage, including BRP1, TGGT1_208740, two secreted cAMP-dependent protein kinases, and BFD2 (*Figure 7—figure supplement 1*).

To identify phosphopeptides most altered by IAA treatment, we summed and ranked phosphopeptides by log$_2$-ratios of abundances in the PKA C3-AID strain relative to the untreated (0 hr) samples (*Figure 7G*). By this metric, 129 phosphopeptides (94 with >1 PSMs) and 49 phosphopeptides (31 with >1 PSMs) were down- or up-regulated by more than 3.5 in the modified Z-scores, respectively. Gaussian mixture-model algorithms heuristically clustered phosphopeptides identified by more than one PSM into three classes (*Figure 7H*). Cluster 1 contained 12 phosphopeptides that rapidly decreased within 1–3 hr of PKA C3 depletion and continued to decrease thereafter. This cluster includes CBP80, a DEP domain protein, *Tg*ZnT, *Tg*ApiAT3-1, AC13, and FtsH1 (*Table 1*).

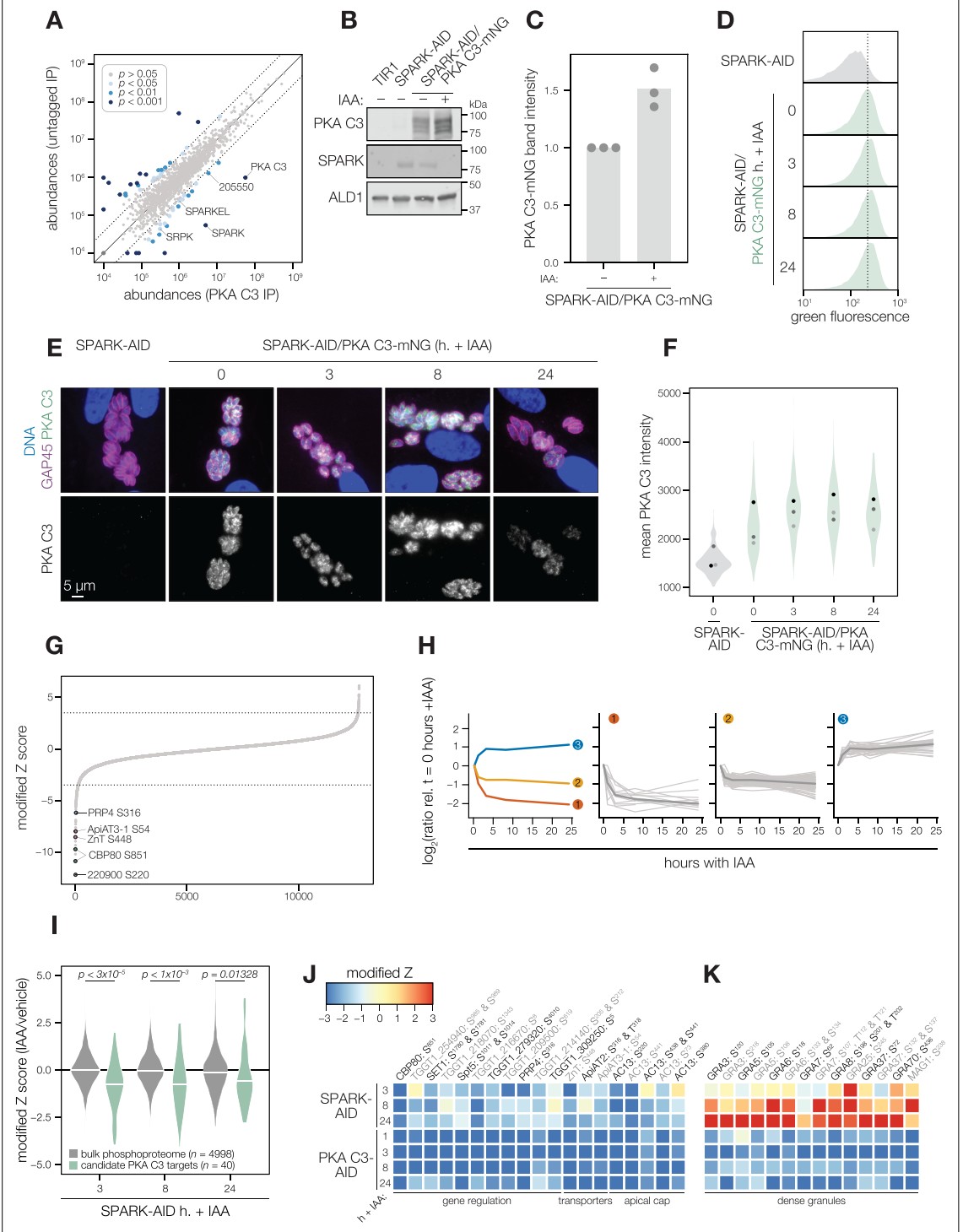

**Figure 7.** SPARK and PKA C3 physically and genetically interact. (**A**) Protein abundances from immunopurified PKA C3-mNG-AID lysates or an untagged control strain from n = 2 biological replicates. Dotted lines correspond to one modified z-score. Only proteins quantified by greater than one peptide are shown. Proteins identified in only one IP were assigned a pseudo-abundance of $10^{4.5}$. Point colors correspond to significance thresholds. (**B**) Immunoblot of parasites expressing SPARK-V5-AID-HA and PKA C3-mNG after 24 hr of IAA treatment. ALD1 serves as a loading control. Band intensity normalized to the dual-tagged strain is shown in (**C**) from three replicates. (**D**) Flow cytometry analysis of SPARK-AID parasites expressing PKA C3-mNG treated with IAA for the indicated number of h. Traces are representative of two biological replicates. The dotted line centers the mode of the vehicle-treated sample. Traces were normalized by unit area. (**E, F**) Fixed, intracellular SPARK-AID/PKA C3-mNG parasites visualized by immunofluorescence microscopy using the mNG epitope after the indicated period of IAA treatment (**E**). The mNG signal was internally normalized to the SPARK-AID parental strain. Quantification of the PKA C3 signal intensity of three replicates is shown in (**F**). (**G**) Phosphopeptide IAA-dependent score (summed

*Figure 7 continued on next page*

*Figure 7 continued*

peptide ratios relative to the PKA C3-AID $t_0$ peptide abundance). Dotted lines correspond to 3.5 modified Z scores. Colored points correspond to phosphosites discussed in the main text. (**H**) Gaussian mixture modeling of PKA C3-dependent peptides identified by more than one PSM revealed three kinetically resolved clusters. Individual peptides or the median ratios in each cluster are depicted by light and dark gray lines, respectively. Clusters are numbered according to their discussion in the main text. (**I**) Violin plots displaying the distribution of phosphopeptide abundance values following SPARK depletion. The distributions of candidate PKA C3 targets are shown in green. The distributions and p-values (KS test) were derived from the overlapping subset of phosphopeptides identified in each dataset. (**J**) Heat map of the abundance ratios of candidate PKA C3 targets following SPARK depletion. (**K**) Heat map of the abundance ratios of dense granule proteins following SPARK depletion, as discussed in the text.

The online version of this article includes the following source data and figure supplement(s) for figure 7:

**Source data 1.** This file contains source data that was used to generate the blot in *Figure 7D*.

**Source data 2.** This file contains source data that was used to generate the blot in *Figure 7D*.

**Source data 3.** This file contains source data that was used to generate the blot in *Figure 7D*.

**Figure supplement 1.** Extended analysis of the PKA C3 depletion proteome.

Cluster 2 comprised 82 phosphopeptides that also decreased within 1–3 hr of IAA treatment, after which their abundances remained stable, overall decreasing to a lesser extent than cluster 1 (*Figure 7H*). Cluster 2 included genes putatively involved in gene regulation, including eIF4G1, NOC3p, a zinc finger protein, a DNA repair protein, SET1, and a nucleotidyltransferase (*Table 1*). Several phosphosites belonging to proteins involved in daughter cell biogenesis were also down-regulated, including AC13, IAP2, BCC8, and condensin 2. We also observed down-regulated phosphosites for transporters in this cluster—for example, TgApiAT2, ATP4, and a nucleoside transporter. The early events in bradyzoite differentiation have not been extensively characterized but involve translational regulation of BFD1 enhanced by a BFD1/BFD2 feed-forward loop (*Licon et al., 2023*; *Waldman et al., 2020*). Down-regulation of PKA C3 may enhance conditions that activate this loop, for example due to global down-regulation of cap-dependent translation (CBP80, eIF4G1) or alteration of the replicative cycle (*Radke et al., 2003*).

Cluster 2 also contains numerous proteins trafficking through secretory organelles, including the micronemes, rhoptries, and dense granules. Modified phosphosites belonging to micronemal proteins included CRMPb, TGGT1_221180, and TGGT1_304490 (*Table 1*). The rhoptry proteins ROP1, ROP13, ROP17, ROP40, and RON9 were also differentially regulated. Modified dense granule/parasitophorous vacuole proteins included GRA3, GRA4, GRA6, GRA7, GRA8, GRA31, GRA57, GRA62, GRA70, SFP1, and MAG1. Many of these proteins are lumenal within their respective organelles, and regulation by PKA C3 is therefore likely indirect. Nonetheless, each of the secretory organelles contain stage-specific subproteomes with implications for metabolism and protein complex assembly in bradyzoites (*Sinai et al., 2020*); the regulation observed here may reflect such early stages of proteomic remodeling.

The 31 phosphopeptides belonging to cluster 3 increased in abundance following PKA C3 depletion and therefore are likely indirect targets of the kinase. This cluster contained phosphoproteins involved in gene regulation (*Table 1*), such as AP2XII-9, an ISWI protein, CAF1, PLU-1, GCFC, and two zinc finger proteins: TGGT1_223880 and BFD2. We also identified several candidates involved in proteostasis, including HSP90, p23, CSN3, PSME4. Regulation of such proteins may be related to the transition to the bradyzoite state, albeit not as a direct consequence of PKA activity.

The protein phosphatase PP2A has recently been characterized as a regulator of bradyzoite differentiation (*Wang et al., 2022*). We therefore analyzed proteins that exhibited both PKA C3- and PP2A-dependent phosphoregulation. Several candidates in this category were identified as differentially important in interferon-stimulated cells, including TGGT1_209500, GRA57, and GRA70. Numerous phosphosites belonging to AC13 were differentially regulated in both phosphoproteomes. Some proteins involved in endo- and exocytosis also exhibited dynamic phosphoregulation, including CRMPb and Kelch13. PKA R S[27] was up-regulated upon PKA C3 depletion and was also up-regulated in *Δpp2a* mutants. Finally, nucleic-acid-binding proteins exhibited differential phosphoregulation, including BFD2, a KH protein, eIF4G2, and AP2XII-1. Given the stage conversion phenotypes of parasites lacking PKA C3 or PP2A, the phosphoregulation of overlapping targets may be functionally important in the context of bradyzoite differentiation.

## The SPARK phosphoproteome shares signatures with the PKA C3–depletion phosphoproteome

We systematically compared the abundances of candidate PKA C3 targets (as defined above) to overlapping phosphosites in the SPARK phosphoproteome. Compared to the bulk phosphoproteome, the PKA C3–dependent phosphopeptides significantly decreased in abundance within three h of SPARK depletion (*Figure 7I*). Several of the phosphoproteins found to be altered by both PKA C3 and SPARK depletion belonged to proteins involved in gene regulation (*Figure 7J* and *Table 1*), including CBP80, eIF4G2, SET1, NOC3p, Spt5, a KH protein, a nucleotidyltransferase, and PRP4. Some of these proteins associate with large transcription factor assemblies, including the AP2IX4/MORC and GCN5b complexes. Transporters also overlapped between the two phosphoproteomes, including ZnT, TgApiAT2, and TgApiAT3-1. Several phosphosites belonging to apical cap proteins were also shared between the phosphoproteomes, including numerous AC13 phosphopeptides.

The SPARK and PKA C3 phosphoproteomes notably diverged with regards to regulation of dense granule proteins, which was observed as a subpopulation of PKA C3–dependent phosphopeptides that increased in abundance upon SPARK depletion (*Figure 7I and K*). This effect was specific to the indicated phosphopeptides, as corresponding protein levels of the dense granule proteins were not changed (*Supplementary file 4*). The differentially regulated dense granule proteins include many of those belonging to cluster 2 of the PKA C3 depletion proteome (*Figure 7H and K*). The phosphoproteome of SPARK shares signatures of reduced PKA C3 activity; however, due to concomitant downregulation of other AGC kinases, the phosphoproteomes of SPARK and PKA C3 depletion diverge in ways that remain to be explored.

## Discussion

SPARK is an ortholog of PDK1, which is considered a key regulator of AGC kinases (*Mora et al., 2004*; *Pearce et al., 2010*). These kinases are united by general structural features required for activity and integrity, such as the activation segment (also known as the activation loop or T-loop), and the C-terminal hydrophobic and turn motifs, which are common loci of regulation that can position the kinase into an active conformation. In mammals, AGC kinases depend on PDK1 in various ways. For example, phosphorylation of the Akt (PKB) activation loop by PDK1 leads to partial activation; additional phosphorylation by the target of rapamycin (mTOR) complex produces maximal activity (*Pearce et al., 2010*). Some protein kinase C (PKC) isoforms undergo sequential phosphorylation by upstream enzymes—including PDK1—that prime PKC for activity and prevent kinase degradation (*Newton, 2010*). PKA catalytic isoforms require T-loop phosphorylation for activity; however, whether this modification is installed autocatalytically or by PDK1 depends on the cellular context (*Cheng et al., 1998*; *Moore et al., 2002*; *Williams et al., 2000*). Consequently, PDK1 dysregulation provokes diverse cellular consequences, including enhanced growth, migration, and cell survival—processes associated with cancer in animals (*Gagliardi et al., 2018*; *Mora et al., 2004*; *Raimondi and Falasca, 2011*). In this study, we characterize the diverse consequences of the PDK1 ortholog dysregulation in the ubiquitous and divergent eukaryotic parasite of animals, *T. gondii*.

A prior study identified SPARK as a regulator of parasite invasion and egress following 24 hr of kinase depletion (*Smith et al., 2022*). Unexpectedly, we observed that three h of SPARK depletion were insufficient to impact *T. gondii* motility or calcium-dependent signaling, indicating that the phenotypes associated with SPARK depletion develop over time. A similar time delay was observed for the development of phenotypes following depletion of the SPARK-interacting protein SPARKEL. Quantitative proteomics revealed that PKA and PKG abundances began to decrease after more than 3 hr of SPARK depletion. Proximity labeling experiments also suggested that SPARK, PKA, and PKG are spatially associated within the parasite cell. We propose a model in which SPARK is required for the activation and stabilization of PKG and PKA.

From our own studies, multiple mass spectrometric and cell biological experiments support the relationship between SPARK and AGC kinases. Furthermore, PDK1 mutations in the related apicomplexan *Plasmodium falciparum* were observed to suppress toxicity related to overexpression of *Pf*PKA (*Hitz et al., 2021*). Both *Tg*PKG and *Tg*PKA C1 possess features of AGC kinases involved in PDK1 interaction, such as the PDK1-interacting fragment FXXF. PKA C1 terminates in the sequence FTSW, whereas PKG has two non-terminal FXXF motifs, FGDF and FLYF. Although these motifs are not strictly

C-terminal, as in animals, their position resembles that observed in some plant AGC kinases, up to 50 amino acids upstream of the C terminus (*Rademacher and Offringa, 2012*). The T-loop phosphorylation sites of each kinase—PKA C1 T190 and PKG T838—were detected in the global phosphoproteome and were down-regulated following SPARK knockdown. The requirement of T-loop threonine phosphorylation for PKG activity is unclear, as recombinant *Plasmodium* spp. PKG was still active when this residue was mutated (*El Bakkouri et al., 2019*). Conformational changes due to cGMP binding—in particular, rearrangement of the N-terminal autoinhibitory segment otherwise adjacent to the activation loop—appeared particularly important for PKG activity. Upon SPARK depletion, multiple phosphorylation sites in the N-terminal and cyclic nucleotide–binding domains of PKG (T16, S20, S22, T59, S62, T96, T99, and S129) decrease in abundance, whereas others remain stable or increase (S40 and S105). Thus, phosphorylation may function in the regulation of these domains. Heterologous expression and activity assays of PKA C1 and PKG may resolve the importance of activation loop and cyclic nucleotide–domain phosphorylation for kinase function; however, such biochemical assays are not currently available for the *T. gondii* kinases. The cause for reduction of PKA C1 and PKG levels also requires further study; in principle, active degradation or dilution mechanisms due to new protein synthesis and cell division are possible. In support of the latter, down-regulation of both kinases was only observed after 8 hr of SPARK depletion, a period longer than the *T. gondii* cell cycle. SPARK could also indirectly stabilize PKA C1 and PKG abundances, for example by phosphorylating and inhibiting a ubiquitin ligase. One candidate could be the HECT domain protein TGGT1_280660, which exhibited SPARK-dependent phosphoregulation and was enriched in SPARK proximity labeling experiments. Examination of the SPARK-dependent ubiquitinome or pulse-chase phosphoproteomics experiments could yield further insight into the mechanisms of PKA C1 and PKG down-regulation following SPARK depletion.

Independent of biochemical mechanisms, genetic evidence supports the hypothesis that SPARK phenotypes associated with kinetic transitions in the lytic cycle—such as motility, egress, and invasion—arise from dysregulated PKA C1 and PKG activities. In *T. gondii*, PKA negatively regulates egress and positively regulates the transition between invasion and the establishment of a replicative niche (*Jia et al., 2017*; *Uboldi et al., 2018*); by contrast, in *P. falciparum*, PKA mediates parasite invasion (*Flueck et al., 2019*; *Patel et al., 2019*; *Wilde et al., 2019*). PKG is more broadly required for apicomplexan motility, invasion, and egress and functions by mobilizing parasite intracellular calcium stores. Genetic pathway analysis of *T. gondii* PKA C1 and PKG inhibition supports a model in which PKG lies downstream of PKA (*Jia et al., 2017*). Increasingly, proteomic, genetic, and physiological studies implicate negative feedback mechanisms between cAMP and cGMP signaling pathways in apicomplexans (*Alam et al., 2015*; *Bisio et al., 2019*; *Brochet et al., 2014*; *Moss et al., 2022*; *Nofal et al., 2022*). In *T. gondii*, SPARK inhibition releases the brakes that PKA C1 holds on egress and jams the engine of PKG activity, resulting in parasite stasis. The PKA signaling circuit differs in *Plasmodium* spp., where a sole PKA C subunit is indispensable for invasion but not egress (*Patel et al., 2019*; *Wilde et al., 2019*). However, SPARK inhibition would lead to a similarly inhibitory effect on parasite proliferation, as has been suggested recently in studies employing altered *P. falciparum* PDK1 and PKA alleles (*Hitz et al., 2021*).

*T. gondii* has two additional paralogs of the PKA catalytic subunit, relative to *Plasmodium* spp., PKA C2 and PKA C3. PKA C2 expression levels are low outside of the parasite sexual stages (*Ramakrishnan et al., 2019*; *Sugi et al., 2016*). PKA C3 disruption was previously reported to increase conversion to the bradyzoite stage (*Sugi et al., 2016*). Our results using a PKA C3 conditional knockdown support this finding. Furthermore, PKA C3 and SPARK associate, as revealed through approaches that probe reciprocal interactions. PKA C3 possesses a canonical PDK1-interacting fragment (FDNF) and shares an activation loop motif with PKA C1. Although PKA C3 peptide levels were below the detection limit in our depletion phosphoproteome experiments, we propose that SPARK depletion leads to down-regulation of PKA C3 activity separately from a reduction in PKA C3 protein abundance. SPARK and PKA C3 depletion phenocopy each other with respect to in vitro differentiation. Candidates for the molecular targets responsible for differentiation may reside in the overlapping phosphosites in the SPARK and PKA C3 depletion phosphoproteomes.

The regulation of SPARK remains an open inquiry. Mammalian PDK1 is constitutively active due to auto-phosphorylation in trans. The PH domain triggers mammalian PDK1 homodimerization upon binding of phosphatidylinositol (3,4,5)-trisphosphate (*Levina et al., 2022*; *Masters et al., 2010*;

*Pearce et al., 2010*). SPARK lacks the PH domain *Smith et al., 2022*; therefore, apicomplexans have likely evolved alternative mechanisms for regulating PDK1. The complex of proteins interacting with SPARK—the elongin-like protein SPARKEL and the putative AGC kinase TGME49_205550—may represent a means of modulating SPARK activity. Elongin C forms a component of multiprotein complexes. E3 ubiquitin ligases are well-known examples; however, elongin C can affect steady-state protein levels through mechanisms distinct from ubiquitination (*Hyman et al., 2002*). In metazoans, PDK1 has been reported to associate with E3 ligases and exhibit monoubiquitination (*Jiang et al., 2021*; *Uras et al., 2012*). However, the SPARK and SPARKEL IPs and proximity experiments failed to identify obvious components of stable ubiquitin ligase complexes. Given the co-depletion of the complex—as has been reported for other tightly associated proteins studied with the AID system (*Tosetti et al., 2020*)—further biochemical studies are required to discern the regulatory interactions between SPARK and SPARKEL. Unlike most apicomplexans, *Plasmodium* spp. appear to have lost SPARKEL, possibly requiring alternative mechanisms to control SPARK activity.

Our unbiased and integrative proteomic approaches uncovered ancient features of signaling networks de novo and revealed how such relationships can be repurposed for parasitism. Apicomplexans are well-adapted to transition between life stages in their hosts. Synthesizing genetic, proteomic, and phenotypic data, we propose that SPARK fundamentally integrates cyclic nucleotide signaling networks in apicomplexan parasites. In *T. gondii*, proper SPARK function preserves tachyzoite physiology by activating PKG, PKA C1, and PKA C3, thus promoting cycling within the acute stages of the infection over differentiation into the chronic stage wherein AGC kinase activity is lower (*Fu et al., 2021*). In principle, SPARK may serve as a central node for regulation of the asexual stages.

# Materials and methods

## Reagents and cell culture

*T. gondii* parasites of the type I RH strain Δ*ku80*/Δ*hxgprt* genetic background ATCC PRA-319, (*Huynh and Carruthers, 2009*) or the type II ME49 strain Δ*ku80*/Δ*hxgprt* background (*Waldman et al., 2020*) were grown in human foreskin fibroblasts (HFFs, ATCC SCRC-1041; male) maintained in DMEM (GIBCO) supplemented with 3% calf serum and 10 μg/mL gentamicin (Thermo Fisher Scientific). When noted, DMEM was supplemented with 10% inactivated fetal calf serum and 10 μg/mL gentamicin. HFFs and parasites were tested routinely for mycoplasma using the ATCC Universal Mycoplasma Detection Kit (30–1012 K).

## Parasite transfection and strain construction

Oligonucleotide sequences are listed in *Supplementary file 5*.

### Genetic background of established parasite strains

This study used the following strains that have been characterized in other publications: RHΔku80Δhxgprt/TIR1 (*Brown et al., 2017*), ME49Δku80Δhxgprt/TIR1 (*Licon et al., 2023*) RHΔku80Δhxgprt/TIR1/pTUB1-GCaMP6f (*Smith et al., 2022*), RHΔku80Δhxgprt/TIR1/pMIC2-MIC2-Gluc-myc-P2A-GCaMP6f (*Herneisen et al., 2022*), PKG-mAID-HA/RHΔku80/TIR1 (*Brown et al., 2017*), CDPK1-mNG-mAID-Ty/RHΔku80Δhxgprt/TIR1 (*Shortt et al., 2023*; *Smith et al., 2022*), SPARK-mNG-mAID-Ty/RHΔku80Δhxgprt/TIR1 (*Smith et al., 2022*), SPARK-mCherry-mAID-HA/RHΔku80Δhxgprt/TIR1/pTUB1-GCaMP6f (*Smith et al., 2022*), and pMIC2-mNG-TurboID-Ty/RHΔku80Δhxgprt/TIR1 (*Chan et al., 2023*). The strains described below were derived from these parental lines.

### SPARK-V5-mNG-Ty/RHΔku80Δhxgprt

The HiT vector cutting unit gBlock described for SPARK in *Smith et al., 2022* was cloned into the pALH193 HiT empty vector, which is based on the pGL015 cloning vector (GenBank: OM640005) but possesses the XTEN-V5-mNG-Ty-3′CDPK3/DHFR tagging unit. The vector was linearized with BsaI and co-transfected with the pSS014 Cas9 expression plasmid (GenBank: OM640002) into RHΔku80Δhxgprt parasites. Clones were selected with 1 μM pyrimethamine and isolated via limiting dilution. Clones were verified by immunofluorescence microscopy and immunoprecipitation and mass spectrometry.

## SPARKEL-V5-mNG-Ty/SPARK-V5-mCherry-HA/RHΔku80Δhxgprt/TIR1

The HiT vector cutting unit gBlock for SPARKEL (P1) was cloned into the pALH193 HiT empty vector. The vector was linearized with BsaI and co-transfected with the pSS014 Cas9 expression plasmid into RHΔku80Δhxgprt/TIR1 parasites. Clones were selected with 1 µM pyrimethamine and isolated via limiting dilution to generate the *SPARKEL-V5-mNG-Ty/RHΔku80Δhxgprt/TIR1* strain. The HiT vector cutting unit gBlock described for SPARK in *Smith et al., 2022* was cloned into the pALH142 HiT empty vector, which is based on the pALH052 cloning vector (GenBank: OM863784) but possesses the XTEN-V5-mCherry-HA-3′CDPK3/HXGPRT tagging unit. The BsaI-linearized template was co-transfected with pSS014 into the *SPARKEL-V5-mNG-Ty/RHΔku80Δhxgprt/TIR1* strain. Clones were selected with 25 µg ml⁻¹ mycophenolic acid and 50 µg ml⁻¹ xanthine and isolated via limiting dilution. Clones were verified by immunofluorescence microscopy.

SPARKEL-V5-HaloTag-mAID-Ty/RHΔku80Δhxgprt/TIR, 1 SPARKEL-V5-HaloTag-mAIDTy/ ME49Δku80Δhxgprt/TIR1, SPARKEL-V5-HaloTag-mAID-Ty/RHΔku80Δhxgprt/TIR1/pMIC2-MIC2-Gluc-myc-P2A-GCaMP6f and SPARKEL-V5-HaloTag-mAID-Ty/SPARK-V5-mCherry-HA/RHΔku80Δhxgprt/ TIR1

The HiT vector cutting unit gBlock for SPARKEL was cloned into the pALH076 HiT empty vector, which is based on the pGL015 cloning vector (GenBank: OM640005) but possesses the XTEN-V5-HaloTag-mAID-Ty-3′CDPK3/DHFR tagging unit. The vector was linearized with BsaI and co-transfected with the pSS014 Cas9 expression plasmid into RHΔku80Δhxgprt/TIR1, RHΔku80Δhxgprt/TIR1/ pMIC2-MIC2-Gluc-myc-P2A-GCaMP6f, or ME49Δku80Δhxgprt/TIR1 parasites. Clones were selected with 1 µM pyrimethamine and isolated via limiting dilution to generate the *SPARKEL-V5-HaloTag-mAID-Ty/RHΔku80Δhxgprt/TIR1* strain. The linearized SPARK-V5-mCherry-HA HiT vector described above was co-transfected with pSS014 into the *SPARKEL-V5-HaloTag-mAID-Ty/RHΔku80Δhxgprt/ TIR1* strain. Clones were selected with 25 µg ml⁻¹ mycophenolic acid and 50 µg ml⁻¹ xanthine and isolated via limiting dilution. Clones were verified by PCR amplification and sequencing of the junction between the 3′ end of SPARKEL (5′-GGGAGGCCACAACGGCGC-3′) and 5′ end of the protein tag (5′-gggggtcggtcatgttacgt-3′), immunoblotting, and immunoprecipitation and mass spectrometry.

## SPARK-V5-mAID-HA and PKA R-V5-mAID-HA/RHΔku80Δhxgprt/TIR1

HiT cutting units previously described for SPARK (*Smith et al., 2022*) and PKA R (P2) were cloned into HiT empty vector pALH086 (XTEN-V5-mAID-HA-3′CDPK3/HXGPRT; GenBank ON312869). In the case of the PKA R vector, the CDPK3 3′UTR was exchanged for the sequence 1092 base pairs downstream of the PKA R stop codon. The vectors were linearized and co-transfected with the Cas9 plasmid pSS014 into the *RHΔku80Δhxgprt/TIR1* strain. Clones were selected with 25 µg ml⁻¹ mycophenolic acid and 50 µg ml⁻¹ xanthine and isolated via limiting dilution. In-frame tagging of PKA R was notably rare. Clones were verified by PCR amplification and sequencing of the junction between the 5′ end of the protein tag (5′-gggggtcggtcatgttacgt-3′) and 3′ end of SPARK (5′-GACGCAAAACTGGCAAGACG -3′) or the 5′UTR of the DHFR selection cassette (5′-tcgacaacgaatgacacaca-3′) and 3′ end of PKA R (5′-GTGCGCATCTTGGAAGACATGGATC-3′).

## SPARKEL-V5-mNG-Ty/SPARK-V5-mAID-HA/RHΔku80Δhxgprt/TIR1

The HiT vector cutting unit gBlock for SPARKEL (P1) was cloned into the pALH193 HiT empty vector. The vector was linearized with BsaI and co-transfected with the pSS014 Cas9 expression plasmid into SPARK-V5-mAID-HA/RHΔku80Δhxgprt/TIR1 parasites. Clones were selected with 1 µM pyrimethamine and isolated via limiting dilution to generate the SPARKEL-V5-mNG-Ty/SPARK-V5-mAID-HA/ RHΔku80Δhxgprt/TIR1 strain. Clones were verified by PCR amplification and sequencing of the junction between the 3′ end of SPARKEL (5′-GGGAGGCCACAACGGCGC-3′) and 5′ end of the protein tag (5′-gggggtcggtcatgttacgt-3′).

## 3HA-mAID-SPARKEL/RHΔku80Δhxgprt/TIR1

An N-terminal HiT cutting unit designed for SPARKEL (P3) was cloned into HiT empty vector pALH460, which is based on the pTUB1-DD HiT vector described in *Licon et al., 2023* but with the 5′DHFR-DHFR-T2A-3HA-mAID-XTEN tagging unit. The vector was linearized and co-transfected with the Cas9 plasmid pSS014 into the *RHΔku80Δhxgprt/TIR1* strain. Clones were selected with 1 µm pyrimethamine

and isolated via limiting dilution. Clones were verified for in-frame tagging via immunofluorescence microscopy.

## PKA C1-mNG-3myc/PKA R-mCherry-Ty/RHΔku80Δhxgprt/TIR1 and SPARKEL-V5-HaloTag-mAID-Ty/PKA C1-mNG-3myc/PKA R-mCherry-Ty

The HiT cutting unit for PKA C1 (P4) was cloned into the HiT empty vector pALH4052, which is based on the pGL015 cloning vector (GenBank: OM640005) but possesses the XTEN-mNG-3myc-3′CDPK3 tagging unit. The HiT cutting unit for PKA R (P3) was cloned into the HiT empty vector pALH4082, which is based on the pGL015 cloning vector (GenBank: OM640005) but possesses the XTEN-mCherry-Ty-3′CDPK3 tagging unit. The vectors were linearized and co-transfected with the Cas9 plasmid pSS014 into the *RHΔku80Δhxgprt/TIR1* strain. Clones were sorted on the basis of mNG and mCherry fluorescence and were isolated via limiting dilution. Clones were verified by live microscopy and flow cytometry. The linearized *SPARKEL-V5-HaloTag-mAID-Ty* HiT vector described above was transfected into the *PKA C1-mNG-3myc/PKA R-mCherry-Ty/RHΔku80Δhxgprt/TIR1* strain with pSS014. Clones were selected with 1 µM pyrimethamine and isolated via limiting dilution.

## SPARK-V5-mAID-HA/PKA C1-mNG-3myc/PKA R-V5-mCherry-2A-DHFR

The PKA C1-mNG-3myc HiT vector described above was co-transfected with pSS014 into the SPARK-V5-mAID-HA/RHΔku80/TIR1 strain. Clones were sorted on the basis of mNG fluorescence and were isolated via limiting dilution. Clones were verified by live microscopy. The HiT cutting unit described for PKA R above was cloned into the HiT empty vector pALH5292, which is based on the pGL015 cloning vector (GenBank: OM640005) but possesses the XTEN-V5-mCherry-P2A-DHFR tagging unit. This vector was linearized with BsaI and was transfected with pSS014 into the SPARK-V5-mAID-HA/PKA C1-mNG-3myc/RHΔku80/TIR1 strain. Clones were selected with 1 µM pyrimethamine and isolated via limiting dilution. Clones were verified by live microscopy.

## PKA C3-V5-mNG-mAID-Ty/RHΔku80Δhxgprt/TIR1 and PKA C3-V5-mNG-mAID-Ty/ME49Δku80Δhxgprt/TIR1

The HiT vector cutting unit gBlock for PKA C3 (P5) was cloned into the pGL015 HiT empty vector (XTEN-V5-mNG-mAID-Ty-3′CDPK3/DHFR; GenBank OM640005). The vector was linearized with BsaI and co-transfected with the pSS014 Cas9 expression plasmid into RHΔku80Δhxgprt/TIR1 or ME49Δku80Δhxgprt/TIR1 parasites. Clones were selected with 1 µM pyrimethamine and isolated via limiting dilution. Clones were verified by PCR amplification and sequencing of the junction between the 3′ end of PKA C3 (5′-CGGGGTCATGGGCTACCTG-3′) and 5′ end of the protein tag (5′-gggggtcggtcatgttacgt-3′), immunofluorescence microscopy, immunoblotting, and immunoprecipitation and mass spectrometry.

## PKA C3-V5-mNG-mAID-Ty/Δbfd1::dTomato/RHΔku80Δhxgprt/TIR1 and SPARK-V5-mAID-HA/Δbfd1::dTomato/RHΔku80/TIR1

The sequences 5′GGGGGGGCTTGATGTAACAGA-3′ and 5′-cacacacttaagtacgggga-3′, which target upstream of the BFD1 transcription start site (*Markus et al., 2020*; *Waldman et al., 2020*) and downstream of its last exon, respectively, were cloned as small guide RNAs into the Cas9 plasmid pSS013 pU6-Universal (*Sidik et al., 2016b*). A repair template sequence with 40 bases of homology upstream to each of these guides was amplified from a pTUB1-dTomato-3′DHFR plasmid using the oligos 5′-caacctgcaggggtcacctctacagtgtttcgcaccatccgtgtcatgtagcctgccaga-3′ and 5′- cgactctacacgggggaggaaggacgtcaacagaccctcttCATGCATGTCCCGCGTTCGT-3′. This repair template was co-transfected with the two sgRNA plasmids described above into *PKA C3-V5-mNG-mAID-Ty/RHΔku80Δhxgprt/TIR1* and *SPARK-V5-mAID-HA/RHΔku80Δhxgprt/TIR1*. Clones were sorted on the basis of dTomato fluorescence and were isolated via limiting dilution. Clones were verified by live microscopy and PCR amplification of the junctions spanning upstream of the BFD1 transcription start site and in the first exon of BFD1 or the 3′ sequence of dTomato. The oligos used for this amplification are included in *Supplementary file 5*.

### PKA C3-mNG-P2A-DHFR/SPARK-V5-mAID-HA/RHΔku80/TIR1

The HiT vector cutting unit gBlock for PKA C3 described above was cloned into the pALH459 HiT empty vector, which is based on the pGL015 cloning vector (GenBank: OM640005) but possesses the XTEN-mNG-P2A-DHFR tagging unit. The vector was linearized with BsaI and co-transfected with the pSS014 Cas9 expression plasmid into SPARK-V5-mAID-HA/RHΔku80/TIR1 parasites (described above). Clones were selected with 1 µM pyrimethamine and isolated via limiting dilution. Clones were verified by PCR amplification and sequencing of the junction between the 3′ end of PKA C3 (5′-CGGGGTCATGGGCTACCTG-3′) and 3′ end of the DHFR selection cassette (5′-gggcagcttctgtatttccg -3′), immunofluorescence microscopy, immunoblotting, and flow cytometry.

### SPARK-TurboID-Ty and SPARKEL-TurboID-Ty/RHΔku80Δhxgprt/TIR1

The HiT vector cutting unit gBlocks for SPARK and SPARKEL described above were cloned into the pALH173 HiT empty vector with the XTEN-TurboID-Ty-3′CDPK3/DHFR tagging payload (*Chan et al., 2023*). The vectors were linearized with BsaI and co-transfected with the pSS014 Cas9 expression plasmid into *RHΔku80Δhxgprt/TIR1* parasites. Clones were selected with 1 µM pyrimethamine and isolated via limiting dilution. Clones were verified by PCR amplification and sequencing of the junction between the 3′ end of SPARK and SPARKEL (using oligos described above) and the 5′ end of the protein tag (5′-gggggtcggtcatgttacgt-3′) as well as mass spectrometry.

### 3HA-TurboID-SPARKEL/RHΔku80Δhxgprt/TIR1

An N-terminal HiT cutting unit designed for SPARKEL (P3) was cloned into HiT empty vector pALH461, which is based on the pTUB1-DD HiT vector described in *Licon et al., 2023* but with the 5′DHFR-DHFR-T2A-3HA-TurboID-XTEN tagging unit. The vector was linearized and co-transfected with the Cas9 plasmid pSS014 into the *RHΔku80Δhxgprt/TIR1* strain. Parasites selected with 1 µm pyrimethamine. Integration of the construct was verified by immunofluorescence microscopy.

## Immunoprecipitation and mass spectrometry

### SPARK-mNG-mAID, SPARK-mNG, and PKA C3-mNG IPs

Approximately $2 \times 10^8$ extracellular SPARK-mNG/SPARK-mNG-mAID or *Δku80Δhxgprt*/ TIR1 tachyzoites were concentrated by spinning and resuspended in a volume of 400 µl lysis buffer (5 mM NaCl, 142 mM KCl, 1 mM $MgCl_2$, 5.6 mM glucose, 25 mM HEPES pH 7.2 with 0.8% IGEPAL CA-630, 1 X Halt Protease Inhibitors, and 250 U/ml benzonase). After 20 min on ice, the parasite lysates were combined with mNeonGreen-Trap Magnetic Agarose (ChromoTek) and were incubated with rotation at 4 °C for 1 hr. The beads were washed three times with lysis buffer. Bound proteins were eluted with 5% SDS in 50 mM TEAB pH 8.5 for 10 min at 70 °C. The eluates were reduced with 5 mM TCEP for 20 min at 50 °C. Alkylation was performed with 20 mM MMTS at room temperature for 10 min. The samples were then precipitated, washed, and digested with 1 µg trypsin/LysC using the S-trap mini columns (Protifi) according to the manufacturer's protocol. Eluted peptides were lyophilized and stored at –80 °C until analysis. PKA C3-mNG samples were harvested and processed in the same way, except the corresponding control strain was mNG-TurboID-Ty/*Δku80Δhxgprt*.

### SPARKEL-Ty IPs

Approximately $2 \times 10^8$ extracellular SPARKEL-mNG-Ty or *Δku80Δhxgprt* tachyzoites were concentrated by spinning and resuspended in a volume of 400 µl lysis buffer (10 mM Tris-HCl pH 7.5, 140 mM NaCl, 1% IGEPAL CA-630, 0.1% sodium deoxycholate, 0.1% SDS, Halt protease inhibitors, and Pierce universal nuclease). After 20 min on ice, the parasite lysates were combined with anti-Ty (BB2) conjugated Protein G beads and were incubated with rotation at 4 °C for 1 hr (*Huet et al., 2018*). The beads were washed three times with lysis buffer. Bound proteins were eluted with 5% SDS in 50 mM TEAB pH 8.5 for 10 min at 70 °C. The eluates were reduced with 5 mM TCEP for 20 min at 50 °C. Alkylation was performed with 20 mM MMTS at room temperature for 10 min. The samples were then precipitated, washed, and digested with 1 µg trypsin/LysC using the S-trap mini columns (Protifi) according to the manufacturer's protocol. Eluted peptides were lyophilized and stored at –80 °C until analysis.

## Mass spectrometry data acquisition and analysis

The lyophilized peptides were resuspended in 10–20 µl of 0.1% formic acid for MS analysis and were analyzed on an Exploris 480 Orbitrap mass spectrometer equipped with a FAIMS Pro source (*Bekker-Jensen et al., 2020*) connected to an EASY-nLC chromatography system using 0.1% formic acid in water as Buffer A and 0.1% formic acid in 80% acetonitrile as Buffer B. SPARK-mNG and PKA C3-mNG IP samples were separated at 300 nl/min on a 60 min gradient of 1–2% B for 1 min, 2–25% B for 41 min, 25–40% B for 6 min, and 40–100% B for 12 min. SPARKEL-Ty IP samples were separated at 300 nl/min on a 90 min gradient of 1–6% B for 1 min, 6–21% B for 41 min, 21–36% B for 21 min, 36–50% B for 10 min, and 50–100% B for 14 min. Both methods ended with a seesaw gradient of 2% B for 6 min and 98% B for 6 min.

The orbitrap and FAIMS were operated in positive ion mode with a positive ion voltage of 1800 V; with an ion transfer tube temperature of 270 °C; using standard FAIMS resolution and compensation voltages of –50 and –65 V. Peptides were filtered for a charge state of 2–6, and a dynamic exclusion of 20 s was applied. Full scan spectra were acquired in profile mode with 1 microscan at a resolution of 60,000, with a scan range of 350–1400 m/z, fill time of 25 ms, normalized AGC target of 300%, intensity threshold of $5 \times 10^3$. The top 15 MS2 spectra were acquired. MS2 spectra were generated with a HCD collision energy of 30 at a resolution of 15,000 with a first mass at 110 m/z, an isolation window of 1.3 m/z, AGC target of 200%, and auto injection time.

Raw mass spectrometry files were processed with the Proteome Discoverer 2.4 software (Thermo). Spectra were searched against the ToxoDB release49 GT1 protein database using Sequest HT (Thermo Fisher Scientific). The search included the following post-translational modifications: dynamic oxidation (+15.995 Da; M), dynamic acetylation (+42.011 Da; N-Terminus), dynamic phosphorylation (+79.966 Da; S, T, Y), and static methylthio (+45.988 Da; C) with a mass tolerance of 10 ppm for precursor ions and 0.2 Da for fragment ions. Enzymatic cleavage was set to trypsin, with 2 allowed missed cleavages per peptide. False discovery was assessed using Percolator with a concatenated target/decoy strategy using a strict FDR of 0.01, relaxed FDR of 0.05, and maximum Delta CN of 0.05. Protein abundance values were calculated using default workflows. Only unique peptide quantification values were used. Protein abundances were calculated from summation of peptide abundances. Sample ratios were calculated based on protein abundances. Hypothesis testing was performed by t-test on the background population of proteins and peptides. Protein abundances and statistical tests are reported in *Supplementary file 1*. The mass spectrometry proteomics data have been deposited to the ProteomeXchange Consortium via the PRIDE partner repository (*Perez-Riverol et al., 2022*) with the following dataset identifiers: SPARK-mNG-mAID IPs, PXD039922; SPARK-mNG IPs, PXD039919; SPARKEL-Ty IPs, PXD039979; PKA C3 IPs, PXD039896.

## Proximity labeling of intracellular TurboID strains and mass spectrometry

### Parasite treatment and harvest

Parasites freshly egressed from host cell monolayers and expressing SPARK-TurboID-Ty, SPARKEL-TurboID-Ty, or mNG-TurboID-Ty were infected onto a 15 cm dish confluent with HFFs (approximately $1–4 \times 10^7$ parasites per dish). The parasites invaded and replicated in host cells for approximately 32 hr, at which point the media was aspirated and replaced with 10 ml of Ringer's solution (155 mM NaCl, 2 mM CaCl$_2$, 3 mM KCl, 1 mM MgCl$_2$, 3 mM NaH$_2$PO$_4$, 10 mM HEPES, 10 mM glucose) with 1% IFS and 500 µM biotin. The dishes were incubated at 37 °C and ambient CO$_2$ for 30 min to allow equilibration of biotin across the host and parasite plasma membranes. The monolayers were then scraped, and parasites were mechanically released by passage through a 27-gauge syringe into a conical tube. The parasite suspension was diluted with phosphate-buffered saline (PBS) and was spun at 1,500 x *g* for 5 min at ambient temperature. The supernatant was decanted. The parasite pellet was resuspended in 1 ml PBS and transferred to a 1.5 ml protein low-bind tube. The tubes were spun at 4 °C and 21,000 x *g* for 1 min, after which the supernatant was discarded and the parasite pellet was resuspended in 1 ml PBS. This spin and wash cycle was repeated twice more, for three washes total to remove excess biotin and serum. After the final wash, the parasite pellet was resuspended in 500 µl lysis buffer (140 mM NaCl, 1% IGEPAL CA-630, 0.1% sodium deoxycholate, 0.1% SDS, 10 mM Tris-HCl, pH 7.5 supplemented with Pierce protease inhibitors and benzonase), was snap-frozen in liquid nitrogen, and was stored at –80 °C until sample processing.

## Biotinylated protein enrichment and MS

The frozen lysates were thawed and combined with 30 µl of Pierce magnetic streptavidin beads. The samples were rotated for 1 hr at room temperature. The beads were magnetically separated and were washed with 1 ml lysis buffer, 1 ml 1 M KCl, 0.1 M NaCO$_3$, 2 M urea/10 mM Tris pH 8.0, and 1 ml lysis buffer. Elution of bound proteins was performed with two incubations of 10 min at 37 or 70 °C in 5% SDS, 50 mM TEAB pH 8.5 with 2.5 mM biotin. The eluates were immediately reduced, alkylated, and digested as described in the immunoprecipitation methods. TurboID samples were analyzed by MS as were IP samples. The mass spectrometry proteomics data have been deposited to the ProteomeXchange Consortium via the PRIDE partner repository (*Perez-Riverol et al., 2022*) with the following dataset identifiers: SPARK-TurboID, PXD039983; SPARKEL-TurboID, PXD039985; TurboID-SPARKEL, PXD039986.

## Immunofluorescence microscopy

Where indicated, parasites were infected onto confluent HFFs and were allowed to grow for 24 hr. Coverslips or coverglass were fixed in 4% formaldehyde in PBS unless otherwise noted. Following three washes in PBS, the fixed cells were permeabilized with 0.25% TritonX-100 for 10 min at room temperature followed by three washes with PBS. The coverslips were incubated in blocking solution (2% BSA in PBS) for 10 min at room temperature, followed by a 60 min incubation in primary antibody solution. After three washes with PBS, the coverslips were incubated in blocking solution at room temperature for 5 min, followed by a 60 min incubation in secondary antibody solution. The coverslips were washed three times in PBS and once in water. Coverslips were mounted with Prolong Diamond and were cured for 30 min at 37 °C. Imaging was performed with the Nikon Ti Eclipse and NIS Elements software package.

## Immunoblotting

### SPARK/SPARKEL co-depletion immunoblotting

HFFs were infected with parasites of the TIR1, SPARKEL-HaloTag-mAID-Ty/TIR1, or SPARK-V5-mCherry-HA/SPARKEL-V5-HaloTag-mAID-Ty/TIR1 strains. Approximately 16 hr later, IAA was added to a final concentration of 500 µM to initiate the 24 hr depletion period. The following day, IAA was added to the 3- and 1 hr treatment samples. The host cell monolayers were scraped and harvested with spinning for 7 min at 1500 x *g*. The parasite pellet was resuspended in 1 ml of HHE and was spun again for 10 min at 2000 x *g* and 4 °C. The pellet was resuspended to $2 \times 10^7$ parasite equivalents per 10 µl in lysis buffer (5 mM NaCl, 142 mM KCl, 1 mM MgCl$_2$, 5.6 mM glucose, 25 mM HEPES pH 7.2 with 0.8% IGEPAL CA-630, 1 X Halt Protease Inhibitors, 1 X PMSF, and 250 U/ml benzonase). The lysates were diluted with 5 x laemmli buffer (10% SDS, 50% glycerol, 300 mM Tris HCl pH 6.8, 0.05% bromophenol blue) and were boiled at 95 °C for 5 min. Samples were run with $1 \times 10^7$ parasite equivalents per lane on a precast 4–15% SDS-PAGE gel. The gel was incubated in 20% ethanol for 10 min and subsequently transfer buffer (25 mM TrisHCl, 192 mM glycine, 20% methanol) for 10 min. Samples were transferred onto a nitrocellulose membrane at 4 °C for 1 hr at 100 V. The membrane was blocked for 1 hr in 5% milk in TBS-T and incubated overnight in primary antibody solution (mouse anti-V5 at 1:1000 dilution or mouse anti-TUB1 at 1:5000 dilution) in 5% milk/TBS-T at 4 °C. The membrane was incubated in secondary antibody solution (1:10,000 Goat anti-Mouse IgG IRDye 800 or 680) in 5% milk/TBS-T for 1 hr at room temperature and was visualized by LI-COR Odyssey CLx. Three 5-min washes were performed between incubations in antibodies. The expected molecular weight of SPARKEL-V5-HaloTag-mAID-Ty is 66 kDa, from the 42.7 kDa tagging payload and 23.3 kDa protein sequence. The expected molecular weight of SPARK-V5-mCherry-HA is 89.7 kDa, from the 31.9 kDa tagging payload and 57.8 kDa protein sequence. The expected molecular weight of SPARK-V5-mAID-HA is 71.3 kDa, from the 13.5 kDa tagging payload and 57.8 kDa protein sequence. The expected molecular weight of SPARKEL-V5-mNG-Ty is 55.2 kDa, from the 31.9 kDa tagging payload and 23.3 kDa protein sequence.

### PKA C3-AID immunoblotting

HFFs were infected with parasites of the TIR1 and PKA C3-V5-mNG-mAID-Ty/TIR1 strains. Approximately 16 hr later, IAA was added to a final concentration of 500 µM to initiate the 24 hr depletion period. The following day, IAA was added to the 3- and 1 hr treatment samples. The host cell monolayers were scraped and harvested with spinning for 7 min at 1500 x *g*. The parasite pellet was

resuspended in 1 ml of HHE and was spun again for 10 min at 2000 x $g$ and 4 °C. The pellet was resuspended to $1\times10^7$ parasite equivalents per 10 µl in lysis buffer (5 mM NaCl, 142 mM KCl, 1 mM MgCl$_2$, 5.6 mM glucose, 25 mM HEPES pH 7.2 with 0.8% IGEPAL CA-630, 1 X Halt Protease Inhibitors, 1 X PMSF, and 250 U/ml benzonase). The lysates were diluted with 5 x laemmli buffer (10% SDS, 50% glycerol, 300 mM Tris HCl pH 6.8, 0.05% bromophenol blue) and were boiled at 95 °C for 5 min. Samples were run with $2\times10^7$ parasite equivalents per lane on a precast 4–15% SDS-PAGE gel. The gel was incubated in 20% ethanol for 10 min and subsequently transfer buffer (25 mM TrisHCl, 192 mM glycine, 20% methanol) for 10 min. Samples were transferred onto a nitrocellulose membrane at 4 °C for 1 hr at 100 V. The membrane was blocked for 1 hr in 5% milk in TBS-T and incubated overnight in primary antibody solution (mouse anti-V5 at 1:1000 dilution or mouse anti-TUB1 at 1:5000 dilution) in 5% milk/TBS-T at 4 °C. The membrane was incubated in secondary antibody solution (1:10,000 Goat anti-Mouse IgG IRDye 800 or 680) in 5% milk/PBS for 1 hr at room temperature and was visualized by LI-COR Odyssey CLx. Three 5 min washes with TBS-T or PBS were performed between incubations in primary and secondary antibodies, respectively. The expected molecular weight of PKA C3-V5-mNG-mAID-Ty is 97.4 kDa, from the 40.2 kDa tagging payload (*Smith et al., 2022*) and 57.2 kDa protein sequence.

## Plaque assays

500 parasites of the indicated strain were inoculated into 12-well plates of HFFs maintained in D10 and allowed to grow undisturbed for 7 days. IAA or vehicle (PBS) was added to a final concentration of 500 µM. Plates were washed with PBS and fixed for 10 min at room temperature with 70% ethanol. Staining was performed for 5 min at room temperature with crystal violet solution, followed by two washes with PBS, one wash with water, and drying.

## Replication assays

Parasites recently egressed from host cells were allowed to invade coverslips confluent with HFFs for 1 hr, after which the media was aspirated and replaced with media containing 500 µM IAA or vehicle. After 24 hr, coverslips were fixed with 4% formaldehyde for 10 min at room temperature. The samples were prepared for immunofluorescence microscopy as described above, using anti-GAP45 as a parasite counterstain. The number of parasites per vacuole was enumerated from three fields of view acquired using a 40 x objective. Significance was assessed using ANOVA and Tukey's test in R (4.0.4).

## Flow cytometry

Parasites were infected into ix-well dishes. Eight h later, the media was replaced. IAA was added to wells at the appropriate time periods prior to sample harvest. The monolayers were scraped, and parasites were mechanically released from host cells by passage through a 27-gauge syringe. Parasites were then passed through a 5 µm filter. The samples were concentrated for 5 min via spinning at 1000 x $g$. The media was replaced with 1 ml PBS. The parasite solutions were analyzed by flow cytometry with a Miltenyi MACSQuant VYB. Histograms are representative of two biological replicates.

## Invasion assays

Parasites expressing SPARKEL-AID and SPARK-AID (*Smith et al., 2022*) were grown in HFFs for 40 hr with 24, 3, or 0 hr of treatment with 500 µM IAA. Extracellular parasites or parasites mechanically released from the host cells were spun at 1500 x $g$ for 5 min and were resuspended in invasion media (HEPES-buffered DMEM without phenol red) supplemented with 1% IFS. The parasite suspension was normalized to $1 \times 10^6$ cells/ml, and $2 \times 10^5$ parasites were added to three wells of a 96-well clear-bottom plate with confluent HFFs. The plate was spun at 290 x $g$ for 5 min to distribute parasites over the host cells. Parasites were allowed to invade for 20 min at 37 °C and 5% CO$_2$. The plate was washed once with PBS to remove unattached parasites and was fixed with 4% formaldehyde in PBS for 15 min. Extracellular parasites were stained with mouse anti-SAG1 for 30 min at room temperature. Following permeabilization with 0.25% triton-X100 for 10 min, all parasites were stained with guinea-pig anti-CDPK1 for 30 min at room temperature. The wells were incubated with a secondary antibody solution containing Hoechst for 30 min. The plate was imaged at ×20 magnification using a Nikon Ti Eclipse epifluorescence scope. The number of parasites invaded was calculated by normalizing the number of intracellular, invaded parasites to host cell nuclei. Assays were performed in biological triplicate. Mean

invasion efficiencies for each biological replicate were calculated from technical triplicates. Plotting and statistical tests were performed with ggplot2 and the compare_means() function with a one-sided t-test.

## Zaprinast egress assays and GCaMP6f measurements

Parasites expressing SPARKEL-AID and pMIC2-MIC2-Gluc-P2A-GCaMP6f at a defined genomic locus (*Herneisen et al., 2022*) were grown in HFFs in glass-bottom dishes (35 mm, 1.5, Ibidi) for 25 hr with 24, 3, or 0 hr of treatment with 500 µM IAA. The dishes were washed once with one volume of Ringer's buffer (155 mM NaCl, 2 mM $CaCl_2$, 3 mM KCl, 1 mM $MgCl_2$, 3 mM $NaH_2PO_4$, 10 mM HEPES, 10 mM glucose), and the media was replaced with 500 µl of Ringer's solution. Imaging was recorded every 2 s for 300 using an Eclipse Ti microscope (Nikon) with an enclosure maintained at 37 °C. After approximately 10 s of imaging, 1 ml of 750 µM zaprinast was added to the dish, for a final compound concentration of 500 µM. The same procedure was used to measure calcium mobilization in parasites expressing SPARK-AID and GCaMP6f (*Smith et al., 2022*). GCaMP6f intensity was measured in regions of interest around vacuoles containing two or more parasites using FIJI. Assays were performed on different days in biological triplicate. Vacuoles were manually scored for time to egress. Plotting and statistical tests were performed with ggplot2 and the scale_compare_means() function with a one-sided t-test.

## Compound 1 treatment egress assays

HFF monolayers in a clear-bottomed 96-well plate were infected with $1 \times 10^5$ parasites of the TIR1 or SPARK-AID strains in fluorobrite media supplemented with 10% calf serum. IAA or PBS were added to a final concentration of 500 µM 24 hr before the start of the assay. The media on the wells was aspirated, and wells were washed once with PBS to remove extracellular parasites. A fluorobrite solution containing a 3 x concentration of compound 1 was added to each well, and the plate was incubated at 37 °C and 5% $CO_2$ for 10 min. Ringer's solution with 2 µM A23187 was added to each well, resulting in dilution of compound 1 to a 1 x concentration. The plate was incubated at 37 °C and 5% $CO_2$ for 10 min, after which 16% formaldehyde was added to each well, resulting in a final concentration of 4% formaldehyde in each well. Fixation occurred for 10 min. The plate was washed with PBS three times. Wells were permeabilized with 0.25% Triton X-100 followed by three washes with PBS. Parasites were stained with guinea pig anti-CDPK1 (*Waldman et al., 2020*) for 1 hr at room temperature. After three washes with PBS, the wells were incubated in a secondary antibody solution (Alexa Fluor 594 goat anti-guinea pig) for 30 min. The wells were washed three times with PBS and were imaged using a BioTek Cytation 3; four fields were acquired at ×20 magnification per well. Percent egress was calculated as the ratio of liberated parasites to intact vacuoles. Curves were fit to the percent egress data in Prism using a sigmoidal dose-response model and difference between the TIR1 and SPARK-AID response curves was assessed with an extra sum of squares F-test. $IC_{50}$ values from individual replicates were assessed for difference using a paired t-test in Prism.

## Differentiation assays

### Plate-based differentiation assays

HFFs in black glass-bottom 96-well plates were infected with three dilutions ranging from $5 \times 10^3$–$2 \times 10^4$ tachyzoites. The plates were spun for 5 min at 290 x *g*. After a four-hour invasion period at 37 °C and 5% $CO_2$, the media was exchanged for DMEM supplemented with 10% FBS for spontaneous differentiation assays or RPMI with 1% FBS for alkaline stress differentiation assays. Spontaneous and alkaline stress differentiation assay samples were incubated at 37 °C/5%$CO_2$ or 37 °C/0% $CO_2$ for 48 hr, respectively, before fixation in 100% cold methanol for 2 min. The assays were incubated for 10 min at room temperature in blocking solution (1% bovine serum albumin in PBS). The wells were then incubated in a primary antibody blocking solution containing rabbit anti-GAP45 or biotinylated dolichos (Vector labs B-1035) for 30 min at room temperature. Secondary staining was performed with streptavidin-APC and goat anti-rabbit Alexa fluor 488 in blocking solution for 30 min at room temperature. Three PBS washes were performed between antibody incubations. Images were acquired with a widefield Nikon Ti epifluorescence scope. The percentage of DBL +vacuoles was determined by manual quantification of vacuoles with DBL staining as a proportion of vacuoles with GAP45 staining. Images included in figure panels were false-colored in FIJI.

## Coverglass-based differentiation assays

HFFs confluent on coverglass were infected with $1\times10^4$–$5\times10^4$ tachyzoites. After a four-hour invasion period at 37 °C and 5% $CO_2$, the media was exchanged for DMEM supplemented with 10% FBS for spontaneous differentiation assays or RPMI with 1% FBS for alkaline stress differentiation assays. Spontaneous and alkaline stress differentiation assay samples were incubated at 37 °C/5%$CO_2$ or 37 °C/0% $CO_2$ for 48 hr, respectively, before fixation in 4% formaldehyde in PBS for 10 min. The assays were incubated for 10 min at room temperature in blocking solution (1% bovine serum albumin in PBS). The wells were then incubated in a primary antibody blocking solution containing rabbit anti-GAP45 or guinea pig anti-CDPK1 or biotinylated dolichos (Vector labs B-1035) for 60 min at room temperature. Secondary staining was performed with streptavidin-APC and goat anti-rabbit Alexa fluor 488 or goat anti-guinea pig Alexa fluor 488 in blocking solution for 60 minat room temperature. Three PBS washes were performed between antibody incubations. Images were acquired with a widefield Nikon Ti epifluorescence scope with 20 x objective, with three images per coverslip. The percentage of DBL +vacuoles was determined by manual quantification of vacuoles with DBL staining as a proportion of vacuoles with GAP45 staining. Images included in figure panels were false-colored in FIJI.

## SPARK 24 hr depletion proteome

### Parasite treatment and harvest

HFF monolayers were infected with parasites expressing SPARK-mNG-mAID. Eight h later, uninvaded parasites were washed away. The replacement media contained 500 µM IAA or PBS vehicle. After 24 hr, the host cell monolayers were scraped, and parasites were mechanically released by passage through syringes with 27-gauge needles. The suspension was passed through 5 µm filters and collected into conical tubes, which were spun for 7 min at 1000 x *g*. The supernatant was decanted, and the parasite pellet was resuspended in 1 ml Fluorobrite DMEM lacking serum. The suspension was spun for 1000 x *g* for 5 min. The supernatant was aspirated, and an additional wash in Fluorobrite DMEM followed by a spin at 1000 x *g* for 5 min was performed. The parasite pellet was resupended in 250 µl, and four technical replicates of 50 µl were combined with 50 µl of 2 x lysis buffer (10% SDS, 4 mM $MgCl_2$, 100 mM TEAB pH 7.55 with 2 X Halt Protease and Phosphatase Inhibitors and 500 U/ml benzonase). The experiment was performed in biological duplicate.

### Proteomics sample preparation and TMT labeling

Samples were prepared as previously described (*Herneisen et al., 2022*). Lysates were reduced 5 mM TCEP at 55 °C for 15 min. Alkylation was performed with 20 mM MMTS for 10 min at room temperature. After the lysates were acidified to a final concentration of 2.5% v/v phosphoric acid, a 6 X volume of S-trap binding buffer (90% methanol, 100 mM TEAB pH 7.55) was added. The solution was loaded onto S-trap mini columns (Protifi) and spun at 4000 x *g* until all of the solution had been passed through the column. The columns were washed four times with 150 µl S-trap binding buffer, followed by a 30 s spin at 4000 x *g*. Proteins were digested overnight in 20 µl of 50 mM TEAB pH 8.5 containing 2 µg of trypsin/LysC mix (Thermo Fisher Scientific) at 37 °C in a humidified incubator. Peptides were eluted in three 40 µl washes with 50 mM TEAB, 0.2% formic acid, and 50% acetonitrile/0.2% formic acid. The eluted peptides were snap-frozen and lyophilized.

The dried peptides were resuspended in 50 µl 100 mM TEAB 8.5. The peptide concentrations of 1/50 dilutions of the samples were quantified using the Pierce Fluorometric Peptide Assay according to manufacturer's instructions. Sample abundances were normalized to 50 µg peptides in 50 µl 100 mM TEAB pH 8.5. Each sample was combined with TMTpro reagents at a 5:1 label:peptide weight/weight ratio. Labeling reactions proceeded for 1 hr at room temperature with shaking at 350 rpm. Unreacted TMT reagent was quenched with 0.2% hydroxylamine. The samples were pooled, acidified to 3% with formic acid, and were loaded onto an EasyPep Maxi Sample Prep column. The samples were washed and eluted according to the manufacturer's instructions. The eluted peptides were snap-frozen and lyophilized until dry. Samples were fractionated with the Pierce High pH Reversed-Phase Peptide Fractionation Kit according to the manufacturer's instructions for TMT-labeled peptides.

## LC-MS data acquisition

The fractions were lyophilized and resuspended in 10–20 µl of 0.1% formic acid for MS analysis and were analyzed on an Exploris 480 Orbitrap mass spectrometer equipped with a FAIMS Pro source (*Bekker-Jensen et al., 2020*) connected to an EASY-nLC chromatography system. Peptides were separated at 300 nl/min on a gradient of 5–20% B for 110 min, 20–28% B for 10 min, 28–95% B for 10 min, 95% B for 10 min, and a seesaw gradient of 95–2% B for 2 min, 2% B for 2 min, 2–98% B for 2 min, 98% B for 2 min, 98–2% B for 2 min, and 2% B for 2 min. The orbitrap and FAIMS were operated in positive ion mode with a positive ion voltage of 1800 V; with an ion transfer tube temperature of 270 °C; using FAIMS user defined mode with compensation voltages of –50 and –65 V, an inner electrode temperature of 100 °C, an outer electrode temperature of 85 °C, with 4.6 ml/min carrier gas and a default charge state of 2. Full scan spectra were acquired in profile mode at a resolution of 60,000, with a scan range of 400–1400 m/z, automatically determined maximum fill time, 300% AGC target, intensity threshold of $5 \times 10^4$, 2–5 charge state, and dynamic exclusion of 30 sconds with a cycle time of 1.5 s between master scans. MS2 spectra were generated with a HCD collision energy of 32 at a resolution of 45,000 with a first mass at 110 m/z, an isolation window of 0.7 m/z, 200% AGC target, and 120ms injection time.

## Data analysis

Raw files were analyzed by Proteome Discoverer 4.2. Peak lists were generated using the Sequest HT search engine and ToxoDB GT1 version 49 sequence database. Trypsin was specified as the digestion enzyme, with a maximum of two missed cleavages. Modifications included dynamic oxidation (+15.995 Da; M), dynamic acetylation (+42.011 Da; N-terminus), static TMT6plex (+229.163 Da; N-terminus), static TMTpro (+304.207 Da; K), and static methylthio (+45.988 Da; C). The allowed mass tolerance for precursor and fragment ions was 10 ppm and 0.02 Da, respectively. False discovery was assessed using Percolator with a concatenated target/decoy strategy using a strict FDR of 0.01, relaxed FDR of 0.05, and maximum Delta CN of 0.05. Protein abundance values were calculated using default workflows. Only unique peptide quantification values were used. Co-isolation and signal-to-noise thresholds were set to 50% and 10, respectively. Protein abundances were calculated from summation of peptide abundances. Normalization was performed according to total peptide amount. Sample ratios were calculated based on protein abundances. Hypothesis testing was performed by ANOVA against individual proteins. Protein abundances are reported in *Supplementary file 3*. The mass spectrometry proteomics data have been deposited to the ProteomeXchange Consortium via the PRIDE partner repository (*Perez-Riverol et al., 2022*) with the dataset identifier PXD040598.

## SPARK, SPARKEL, and PKA C3 depletion time course proteomes

### Parasite treatment and harvest

HFF monolayers were infected with parasites expressing SPARK-mNG-mAID, SPARKEL-HaloTag-mAID, or PKA C3-mNG-mAID. Eight h later, uninvaded parasites were washed away. IAA was added to 500 µM for the indicated periods of time before the samples were harvested together 32 h post-infection. The host cell monolayers were scraped, and parasites were mechanically released by passage through syringes with 27-gauge needles. The suspension was passed through 5 µm filters and collected into conical tubes, which were spun for 7 mintes at 1000 x *g*. The supernatant was decanted, and the parasite pellet was resuspended in 1 ml Fluorobrite DMEM lacking serum. The suspension was spun for 1000 x *g* for 5 min. The supernatant was aspirated, and an additional wash in Fluorobrite DMEM followed by a spin at 1000 x *g* for 5 min was performed. The parasite pellet was resupended in 50 µl of 1 x lysis buffer (5% SDS, 2 mM MgCl$_2$, 100 mM TEAB pH 7.55 with 1 X Halt Protease and Phosphatase Inhibitors and 500 U/ml benzonase). The experiment was performed in biological duplicate.

### Proteomics sample preparation and TMT labeling

Samples were reduced, alkylated, and digested as described for the SPARK 24 hr depletion proteome. The lyophilized peptides were quantified using a Pierce fluorometric peptide assay. Samples destined for phosphopeptide enrichment were normalized to 100 µg of peptides per channel and were labeled with 200 µg of TMT10plex reagents according to the manufacturer's protocol, with the following modifications. The eluates were combined with 200 µg of TMT10plex reagent in 15 µl of acetonitrile, for an

estimated 1:2 w/w peptide:tag labeling reaction. The labeling proceeded for 1 hr at room temperature and was quenched for 15 min with 5% hydroxylamine. The samples were then pooled, flash-frozen, and lyophilized to dryness. The SPARKEL-AID proteome, which was not phospho-enriched, contained only 7 µg per channel and was labeled with 15 µg of TMT10plex reagent.

## Phosphopeptide enrichment

Phosphopeptide enrichment was performed as previously described (*Herneisen et al., 2022*) using the SMOAC protocol (*Tsai et al., 2014*). Approximately 95% of the proteomics sample was used for phosphopeptide enrichment. Resuspended TMT10plex-labeled samples were enriched with the High-Select TiO2 Phosphopeptide Enrichment Kit (Thermo Fisher Scientific A32993). The flow-through and the eluate from IMAC enrichment were immediately snap-frozen and lyophilized. The flow-through was resuspended and enriched with the High-Select Fe-NTA Phosphopeptide Enrichment Kit (Thermo Fisher Scientific A32992) according to the manufacturer's instructions. The eluted phosphopeptides were immediately snap-frozen and lyophilized. The remaining 5% of the sample was not subjected to the phosphopeptide enrichment protocol. Unenriched and enriched proteome samples were fractionated with the Pierce High pH Reversed-Phase Peptide Fractionation Kit according to the manufacturer's instructions for TMT-labeled peptides. The phosphopeptides enriched with the TiO2 and Fe-NTA methods were combined prior to fractionation.

## LC-MS data acquisition

The fractions were lyophilized and resuspended in 10–20 µl of 0.1% formic acid for MS analysis and were analyzed on an Exploris 480 Orbitrap mass spectrometer equipped with a FAIMS Pro source (*Bekker-Jensen et al., 2020*) connected to an EASY-nLC chromatography system as described above. Peptides were separated at 300 nl/min on a gradient of 6–21% B for 41 min, 21–36% B for 20 min, 36–50% B for 10 min, and 50 to 100% B over 15 min. The orbitrap and FAIMS were operated in positive ion mode with a positive ion voltage of 1800 V; with an ion transfer tube temperature of 270 °C; using standard FAIMS resolution and compensation voltages of –50 and –65 V (injection 1) or –40 and –60 V (injection 2). Full scan spectra were acquired in profile mode at a resolution of 120,000, with a scan range of 350–1200 m/z, automatically determined maximum fill time, standard AGC target, intensity threshold of $5 \times 10^3$, 2–5 charge state, and dynamic exclusion of 30 s with a cycle time of 2 s between master scans. MS2 spectra were generated with a HCD collision energy of 36 at a resolution of 30,000 using TurboTMT settings with a first mass at 110 m/z, an isolation window of 0.7 m/z, standard AGC target, and auto injection time.

## Data analysis

Raw files were analyzed in Proteome Discoverer 2.4 (Thermo Fisher Scientific) to generate peak lists and protein and peptide IDs using Sequest HT (Thermo Fisher Scientific) and the ToxoDB GT1 version 49 sequence database. Raw files of the phosphopeptide-enriched and unenriched samples were analyzed separately. The unenriched sample search included the following post-translational modifications: dynamic oxidation (+15.995 Da; M), dynamic acetylation (+42.011 Da; N-terminus), static TMT6plex (+229.163 Da; any N-terminus), static TMT6plex (+229.163 Da; K), and static methylthio (+45.988 Da; C). The enriched sample search included the same post-translational modifications, but with the addition of dynamic phosphorylation (+79.966 Da; S, T, Y). The allowed mass tolerance for precursor and fragment ions was 10 ppm and 0.02 Da, respectively. False discovery was assessed using Percolator with a concatenated target/decoy strategy using a strict FDR of 0.01, relaxed FDR of 0.05, and maximum Delta CN of 0.05. Only unique peptide quantification values were used. Co-isolation and signal-to-noise thresholds were set to 50% and 10, respectively. Normalization was performed according to total peptide amount. In the case of the unenriched samples, protein abundances were calculated from summation of non-phosphopeptide abundances. The mass spectrometry proteomics datasets have been deposited to the ProteomeXchange Consortium via the PRIDE partner repository (*Perez-Riverol et al., 2022*) with the following dataset identifiers: SPARK-AID phosphoproteomics, PXD040602; SPARKEL-AID proteomics, PXD040635; PKA C3-AID phosphoproteomics, PXD042989.

Exported phosphopeptide and unenriched protein abundance files from Proteome Discoverer 2.4 were loaded into R (version 4.0.4). The normalized abundance values of all phosphopeptides were used as input for a principal component analysis using the R stats package (version 3.6.2). The strain

basal downregulation score was determined for the SPARK-AID phosphoproteome by calculating the modified z-score of the $\log_2$-ratios of altered abundances in the SPARK-AID strain with no IAA treatment relative to the TIR1 parental strain. The IAA score was calculated by summing the $\log_2$-ratios of altered phosphopeptide abundances in the SPARK-AID strain relative to the SPARK-AID untreated samples, followed by transformation into a modified z-score. Clustering analysis was performed with the mclust package (*Scrucca et al., 2016*) (version 5.4.7) using the $\log_2$ ratios of relative to the vehicle-treated SPARK-AID samples. Cluster assignments are reported in *Supplementary file 3*. The PKA C3-AID phosphoproteome IAA score was calculated as described above; a strain basal downregulation score was not calculated because the experiment did not include a TIR1 parental sample.

## Enrichment analysis

Gene ontology enrichment analysis was performed as described (*Herneisen et al., 2022*). Sets of gene ontology terms from the differentially regulated (SPARK-dependent) and background proteome (all proteins with quantification values in the SPARK-AID mass spectrometry experiment) were downloaded from https://toxodb.org/toxo/app (Molecular Function, Computed evidence, P-Value cutoff set to 1). Gene ontology terms were tested for enrichment across all gene ontology terms identified in the background proteome. A p-value for the likelihood of a given enrichment to have occurred by chance was obtained using a hypergeometric test.

## PKA R depletion phosphoproteome

### Parasite treatment and harvest

HFF monolayers were infected with parasites expressing PKA R-V5-mAID-3HA. The next morning, IAA was added to 500 µM one hour prior to sample harvest. The host cell monolayers were scraped, and parasites were mechanically released by passage through syringes with 27-gauge needles. The suspension was passed through 5 µm filters and collected into conical tubes, which were spun for 7 min at 1000 x *g*. The supernatant was decanted, and the parasite pellet was resuspended in 1 ml Fluorobrite DMEM lacking serum. The suspension was spun for 1000 x *g* for 5 min. The supernatant was aspirated, and an additional wash in Fluorobrite DMEM followed by a spin at 1000 x *g* for 5 min was performed. The parasite pellet was resuspended in 250 µl Fluorobrite DMEM containing 500 µM zaprinast, and 50 µl of this suspension was combined at 0, 10, 30, and 60 s post-treatment with 50 µl of 2 x lysis buffer for a final composition of 5% SDS, 2 mM $MgCl_2$, 100 mM TEAB pH 7.55 with 1 X Halt Protease and Phosphatase Inhibitors and 500 U/ml benzonase. The experiment was performed in biological duplicate.

### Proteomics sample preparation and TMT labeling

Samples were reduced, alkylated, and digested as described previously (*Herneisen et al., 2022*). In brief, the eluates were reduced with 5 mM TCEP for 20 min at 50 °C. Alkylation was performed with 20 mM MMTS at room temperature for 10 min. The samples were then precipitated, washed, and digested with 1 µg trypsin/LysC using the S-trap mini columns (Protifi) according to the manufacturer's protocol. The lyophilized peptides were quantified using a Pierce fluorometric peptide assay. Samples destined for phosphopeptide enrichment were normalized to 50 µg of peptides per channel at a concentration of 1 µg/µl and were labeled with 250 µg of TMTpro reagents according to the manufacturer's protocol, with volumes adjusted accordingly. The labeling proceeded for 1 hr at room temperature and was quenched for 15 min with 5% hydroxylamine. The samples were pooled, acidified to 3% with formic acid, and were loaded onto an EasyPep Maxi Sample Prep column. The samples were washed and eluted according to the manufacturer's instructions. The eluted peptides were snap-frozen and lyophilized until dry.

### Phosphopeptide enrichment

Phosphopeptide enrichment was performed as previously described (*Herneisen et al., 2022*) using the SMOAC protocol (*Tsai et al., 2014*). Approximately 95% of the proteomics sample was used for phosphopeptide enrichment. Resuspended TMTpro-labeled samples were enriched with the High-Select $TiO_2$ Phosphopeptide Enrichment Kit (Thermo Fisher Scientific A32993). The flow-through and the eluate from IMAC enrichment were immediately snap-frozen and lyophilized. The flow-through was resuspended and enriched with the High-Select Fe-NTA Phosphopeptide Enrichment Kit (Thermo

Fisher Scientific A32992) according to the manufacturer's instructions. The eluted phosphopeptides were immediately snap-frozen and lyophilized. Unenriched proteome samples were fractionated with the Pierce High pH Reversed-Phase Peptide Fractionation Kit according to the manufacturer's instructions for TMT-labeled peptides. The phosphopeptides enriched with the TiO2 and Fe-NTA methods were combined prior to fractionation. The remaining 5% of the sample was not subjected to the phosphopeptide enrichment protocol. Enriched and non-phosphopeptide-enriched proteome samples were fractionated with the Pierce High pH Reversed-Phase Peptide Fractionation Kit using the following gradients (% acetonitrile/0.1%triethylamine): 1.5, 3, 5, 7.5, 10, 15, 50. Samples were immediately flash-frozen and lyophilized to dryness.

## LC-MS data acquisition

The fractions were lyophilized and resuspended in 5–15 µl of 0.1% formic acid for MS analysis and were analyzed on an Exploris 480 Orbitrap mass spectrometer equipped with a FAIMS Pro source (*Bekker-Jensen et al., 2020*) connected to an EASY-nLC chromatography system with a 15 cm column with 75 µm diameter (Thermo Fisher ES900). Peptides were separated at 300 nl/min on a gradient of 5–20% B for 110 min, 20–28% B for 10 min, 28–95% B for 10 min, 95% B for 10 min, and a seesaw gradient of 95–2% B for 2 min, 2% B for 2 min, 2–98% B for 2 min, 98% B for 2 min, 98–2% B for 2 min, and 2% B for 2 min. The orbitrap and FAIMS were operated in positive ion mode with a positive ion voltage of 1800 V; with an ion transfer tube temperature of 270 °C; using standard FAIMS resolution and compensation voltages of –50 and –65 V, an inner electrode temperature of 100 °C, and outer electrode temperature of 80 °C with 4.6 ml/min carrier gas. Full scan spectra were acquired in profile mode at a resolution of 60,000, with a scan range of 400–1400 m/z, automatically determined maximum fill time, 300% AGC target, intensity threshold of $5 \times 10^4$, 2–5 charge state, and dynamic exclusion of 30 s with a cycle time of 1.5 s between master scans. MS2 spectra were generated with a HCD collision energy of 32 at a resolution of 45,000 using TurboTMT settings with a first mass at 110 m/z, an isolation window of 0.7 m/z, 200% AGC target, and 120ms injection time.

## Data analysis

Raw files were analyzed in Proteome Discoverer 2.4 (Thermo Fisher Scientific) to generate peak lists and protein and peptide IDs using Sequest HT as described for the SPARK 24 hr depletion proteome. The mass spectrometry proteomics datasets have been deposited to the ProteomeXchange Consortium via the PRIDE partner repository (*Perez-Riverol et al., 2022*) with the dataset identifier PXD044398.

Exported peptide and protein abundance files from Proteome Discoverer 2.4 were loaded into R (version 4.0.4). Ratios relative to the vehicle-treated timepoint were standardized by a modified Z-score. Peptides with the same site phosphorylations were matched across the PKA R and SPARK depletion phosphoproteomes. We defined basally PKA C1-dependent phosphorylation sites as phosphopeptides that were up-regulated by more than two modified Z-scores when PKA R was depleted for one hour. The distributions of all phosphopeptides identified in both the SPARK and PKA R phosphoproteomes were compared with the distributions of PKA C1-dependent sites using a Kolmogorov-Smirnov test.

## PKG depletion phosphoproteome

### Parasite treatment and harvest

HFF monolayers were infected with parasites expressing PKG-mAID-3HA or the parental strain expressing TIR1. The next morning, IAA was added to 500 µM 4 hr prior to sample harvest. The host cell monolayers were scraped, and parasites were mechanically released by passage through syringes with 27-gauge needles. The suspension was passed through 5 µm filters and collected into conical tubes, which were spun for 7 min at 1000 x *g*. The supernatant was decanted, and the parasite pellet was resuspended in 1 ml Fluorobrite DMEM lacking serum. The suspension was spun for 1000 x *g* for 5 min. The supernatant was aspirated, and an additional wash in Fluorobrite DMEM followed by a spin at 1000 x *g* for 5 min was performed. The parasite pellet was resuspended in 250 µl Fluorobrite DMEM containing 500 µM zaprinast, and 50 µl of this suspension was combined at 0, 10, 30, and 60 s post-treatment with 50 µl of 2 x lysis buffer for a final composition of 5% SDS, 2 mM MgCl$_2$, 100 mM TEAB pH 7.55 with 1 X Halt Protease and Phosphatase Inhibitors and 500 U/ml benzonase. The experiment was performed in biological duplicate.

## Proteomics sample preparation and TMT labeling

Samples were reduced, alkylated, and digested as described previously (*Herneisen et al., 2022*). In brief, the eluates were reduced with 5 mM TCEP for 20 min at 50 °C. Alkylation was performed with 20 mM MMTS at room temperature for 10 min. The samples were then precipitated, washed, and digested with 1 µg trypsin/LysC using the S-trap mini columns (Protifi) according to the manufacturer's protocol. The lyophilized peptides were quantified using a Pierce fluorometric peptide assay. Samples destined for phosphopeptide enrichment were normalized to 50 µg of peptides per channel at a concentration of 1 µg/µl and were labeled with 250 µg of TMTpro reagents according to the manufacturer's protocol, with volumes adjusted accordingly. The labeling proceeded for 1 hr at room temperature and was quenched for 15 min with 5% hydroxylamine. The samples were then pooled, flash-frozen, and lyophilized to dryness. The dry pooled sample was resuspended in 1 ml of 5% formic acid and was desalted using the EasyPep Maxi Kit (Thermo Scientific) according to the manufacturer's instructions. The eluted peptides were flash-frozen and lyophilized to dryness.

## Phosphopeptide enrichment

Phosphopeptide enrichment was performed as previously described (*Herneisen et al., 2022*) using the SMOAC protocol (*Tsai et al., 2014*). Approximately 95% of the proteomics sample was used for phosphopeptide enrichment. Resuspended TMTpro-labeled samples were enriched with the High-Select $TiO_2$ Phosphopeptide Enrichment Kit (Thermo Fisher Scientific A32993). The flow-through and the eluate from IMAC enrichment were immediately snap-frozen and lyophilized. The flow-through was resuspended and enriched with the High-Select Fe-NTA Phosphopeptide Enrichment Kit (Thermo Fisher Scientific A32992) according to the manufacturer's instructions. The eluted phosphopeptides were immediately snap-frozen and lyophilized. Unenriched proteome samples were fractionated with the Pierce High pH Reversed-Phase Peptide Fractionation Kit according to the manufacturer's instructions for TMT-labeled peptides. The phosphopeptides enriched with the TiO2 and Fe-NTA methods were combined prior to fractionation. The remaining 5% of the sample was not subjected to the phosphopeptide enrichment protocol. Enriched and non-phosphopeptide-enriched proteome samples were fractionated with the Pierce High pH Reversed-Phase Peptide Fractionation Kit using the following gradients (% acetonitrile/0.1% triethylamine): 1.5, 3, 5, 7.5, 10, 15, 50. Samples were immediately flash-frozen and lyophilized to dryness.

## LC-MS data acquisition

The fractions were lyophilized and resuspended in 5–15 µl of 0.1% formic acid for MS analysis and were analyzed on an Exploris 480 Orbitrap mass spectrometer equipped with a FAIMS Pro source (*Bekker-Jensen et al., 2020*) connected to an EASY-nLC chromatography system with a 25 cm column with 75 µm diameter (Thermo Fisher ES902). Peptides were separated at 300 nl/min on a gradient of 5–20% B for 110 min, 20–28% B for 10 min, 28–95% B for 10 min, 95% B for 10 min, and a seesaw gradient of 95–2% B for 2 min, 2% B for 2 min, 2–98% B for 2 min, 98% B for 2 min, 98–2% B for 2 min, and 2% B for 2 min. The orbitrap and FAIMS were operated in positive ion mode with a positive ion voltage of 1800 V; with an ion transfer tube temperature of 270 °C; using standard FAIMS resolution and compensation voltages of –50 and –65 V, an inner electrode temperature of 100 °C, and outer electrode temperature of 85 °C with 4.6 ml/min carrier gas. Full scan spectra were acquired in profile mode at a resolution of 60,000, with a scan range of 400–1400 m/z, automatically determined maximum fill time, 300% AGC target, intensity threshold of $5×10^4$, 2–5 charge state, and dynamic exclusion of 30 s with a cycle time of 1.5 s between master scans. MS2 spectra were generated with a HCD collision energy of 32 at a resolution of 45,000 using TurboTMT settings with a first mass at 110 m/z, an isolation window of 0.7 m/z, 200% AGC target, and 120ms injection time.

## Data analysis

Raw files were analyzed in Proteome Discoverer 2.4 (Thermo Fisher Scientific) to generate peak lists and protein and peptide IDs using Sequest HT as described for the SPARK 24 hr depletion proteome. The mass spectrometry proteomics datasets have been deposited to the ProteomeXchange Consortium via the PRIDE partner repository (*Perez-Riverol et al., 2022*) with the dataset identifier PXD044361.

Exported peptide and protein abundance files from Proteome Discoverer 2.4 were loaded into R (version 4.0.4). Ratios relative to the vehicle-treated timepoint were standardized by a modified Z-score. Peptides with the same site phosphorylations were matched across the PKG and SPARK depletion phosphoproteomes. We defined basally PKG-dependent phosphorylation sites as phospho-peptides that were down-regulated by more than two modified Z-scores when PKG was depleted for four h. The distributions of all phosphopeptides identified in both the SPARK and PKG phosphoproteomes were compared with the distributions of PKG-dependent sites using a Kolmogorov-Smirnov test.

## PKA C3 depletion transcriptomics

### Parasite Harvest

TIR1/RH and PKA C3-mAID parasites were inoculated onto 15 cm dishes of HFF monolayers. After ~6–8 hr, auxin was added to a final concentration of 500 µM. Twenty-four hr after addition of auxin, dishes were placed on ice and washed twice with PBS containing 100 µg/mL cycloheximide. Lysis buffer (20 mM HEPES pH 7.5, 100 mM KCl, 5 mM MgCl$_2$, 1% Triton X-100, 2 mM DTT, 100 ug/mL cycloheximide, 500 U/mL RNasln Plus, and cOmplete, Mini Protease Inhibitor Cocktail) was added to the dish, the monolayer was scraped, and lysate was transferred to an eppendorf tube to incubate for 10 min on ice. Lysate was passed through a 27-gauge needle ix times to facilitate lysis. Lysate was then centrifuged for 10 min at 1300 x $g$, and the supernatant was flash-frozen and stored at –80°C. Three biological replicates of each strain were collected, and each sample was split into two technical replicates for downstream library preparation.

### RNA-sequencing and analysis

Lysates were thawed on ice and a spike-in of $D.$ $melanogaster$ mRNA was added to each sample before Trizol extraction of RNA. RNA was polyA selected using the NEXTFLEX Poly(A) Beads 2.0 Kit, and sequencing libraries were prepared according to the manufacturer's protocol with the NEXTFLEX Rapid Directional RNA-Seq Kit 2.0 and NEXTFLEX RNA-Seq 2.0 Unique Dual Index barcodes for multiplexing. Libraries were sequenced on one lane of a NovaSeq S4 with 150x150 paired-end reads at the Whitehead Institute Genome Technology Core.

Reads adapters were trimmed with TrimGalore (version 0.6.7). A metagenome of $T.$ $gondii,$ $H.$ $sapiens,$ and $D.$ $melanogaster$ was prepared using STAR (version 2.7.1 a) from genome FASTA files and gff annotation files downloaded from VEuPathDb Release 59. Trimmed reads were aligned to the metagenome using STAR, and mapped reads were quantified using featureCounts (version 1.6.2). Differential expression analysis of $T.$ $gondii$ genes with at least 10 reads across all samples was performed using DESeq2 (version 1.30.1) in R.

## SPARKEL phylogenetic analysis

SPARKEL (TGGT1_291010) homologs were identified from representative apicomplexan and meta-zoan (outgroup) genomes. The Skp1 domain sequences were extracted based on annotation with Interpro domain IPR011333 or PFam domain PF03931. Domain alignment was performed in MEGA using ClustalW with a gap opening penalty of 3, gap extension penalty of 1.8, and remaining default parameters. A Neighbor-Joining Tree was generated and visualized with MEGA using the boot-strap method with 1000 replications as a test of phylogeny and a Jones-Taylor-Thornton model and remaining default parameters.

## SPARK depletion RT-qPCR

### Parasite treatment and harvest

HFF monolayers were infected with parasites expressing SPARK-mNG-mAID or TIR1 parental line. Eight h later, uninvaded parasites were washed away. IAA was added to 500 µM for the 24, 3, or 0 hr before the samples were harvested together 32 hr post-infection. Intracellular parasites were mechanically released by passage through 27-gauge needles. The suspension was passed through 5 µm pore filters, pelleted for 7 min at 1000 x $g$. The supernatant was decanted, and the parasite pellet was resuspended in 5 mL PBS and pelleted again, the resuspended with 1 mL PBS, transferred to micro-centrifuge tubes and pelleted a final time. All PBS was removed and pellets were resuspended in TRIzol Reagent (Life Technologies), vortexed, and snap-frozen.

## RNA extraction and RT–qPCR

Total RNA was extracted using TRIzol Reagent and then quantified using a NanoDrop spectrophotometer before proceeding with first-strand cDNA synthesis with SuperScriptIII Reverse Transcriptase (Thermo Fisher) using random hexamer priming. Total RNA was heated with dNTPs and random hexamer oligo to 65 °C for 5 min and immediately chilled on ice before proceeding with remaining cDNA synthesis steps. First-strand reaction mixtures heated to 25 °C for 5 min, 50 °C for 30 min, and 70 °C for 15 min.

qRT-PCRs were performed using diluted cDNA with pPCR primers against PKG (P9-10), PKA C1 (P11-12), PKA C3 (P13-14), GCN5b (P15-16), and ACT1 (P17-18) and PowerUp SYBR Green Master Mix (Thermo Fisher) according to manufacturer's instructions. Reactions were run on a QuantStudio6 real-time PCR system (Applied Biosystems) with the following cycling conditions: 2 min at 50 °C, 2 min at 95 °C, and 40 cycles of 15 s at 95 °C, followed by 15 s at 56 °C and 1 min at 72 °C. The transcripts of interest were quantified by comparative Ct ($\Delta\Delta$Ct) against GCN5b and ACT1 as reference genes, and the fold changes averaged. The experiment was performed in biological quadruplicate.

## Materials availability statement

All mass spectrometry proteomics data have been deposited to the ProteomeXchange Consortium via the PRIDE partner repository with the accession numbers PXD039896, PXD039919, PXD039922, PXD039979, PXD039983, PXD039985, PXD039986, PXD040598, PXD040602, PXD040635, PXD042989, PXD044361, PXD044398. Sequences of cloning vectors generated for this study are listed in *Supplementary file 5*. Strains and plasmids generated for this study can be obtained by emailing the corresponding author.

## Acknowledgements

We thank Faye Harling for assistance with cell culture, L David Sibley for ALD1 antibody and RH TIR1 strain, Dominique Soldati-Favre and Drew Etheridge for the GAP45 antibodies, Marc-Jan Gubbels for the TUB1 antibody, Kevin Brown for the PKG-AID strain, and members of the Lourido laboratory for helpful discussions. Aditi Shukla developed the BFD1 knockout strategy. The Whitehead Institute Genome Technology Core assisted in nucleic acid sequencing, and the Whitehead Flow Cytometry Core assisted with sorting of fluorescent parasite strains. This work relied extensively on VEuPathDB (https://VEuPathDB.org) and we thank all contributors to this resource. This research was supported by funds from R01AI144369 to SL and a National Science Foundation Graduate Research Fellowship to ALH (174530) and TAS (2018259980).

## Additional information

### Competing interests

Sebastian Lourido: Reviewing editor, *eLife*. The other authors declare that no competing interests exist.

### Funding

| Funder | Grant reference number | Author |
|---|---|---|
| National Institutes of Health | R01AI144369 | Sebastian Lourido |
| National Science Foundation Graduate Research Fellowship Program | 174530 | Alice L Herneisen |
| National Science Foundation Graduate Research Fellowship Program | 2018259980 | Tyler A Smith |

| Funder | Grant reference number | Author |
|---|---|---|

The funders had no role in study design, data collection and interpretation, or the decision to submit the work for publication.

## Author contributions

Alice L Herneisen, Conceptualization, Funding acquisition, Validation, Investigation, Visualization, Methodology, Writing - original draft, Writing - review and editing; Michelle L Peters, Formal analysis, Investigation, Methodology; Tyler A Smith, Conceptualization, Resources, Formal analysis, Investigation; Emily Shortt, Resources, Formal analysis, Investigation; Sebastian Lourido, Conceptualization, Resources, Supervision, Funding acquisition, Project administration, Writing - review and editing

## Author ORCIDs

Alice L Herneisen http://orcid.org/0000-0003-3368-0893
Sebastian Lourido https://orcid.org/0000-0002-5237-1095

Reviewer #1 (Public Review): https://doi.org/10.7554/eLife.93877.3.sa1
Reviewer #2 (Public Review): https://doi.org/10.7554/eLife.93877.3.sa2
Reviewer #3 (Public Review): https://doi.org/10.7554/eLife.93877.3.sa3
Author response https://doi.org/10.7554/eLife.93877.3.sa4

# Additional files

## Supplementary files

• Supplementary file 1. Protein quantification and statistical tests for immunoprecipitation-mass spectrometry experiments (SPARK, SPARKEL, and PKA C3-tagged strains), exported from the Proteome Discoverer 2.4 software.

• Supplementary file 2. Protein quantification and statistical tests for TurboID mass spectrometry experiments (SPARKEL and SPARK TurboID-tagged strains), exported from the Proteome Discoverer 2.4 software.

• Supplementary file 3. Protein and peptide quantification and statistical tests for SPARK, SPARKEL, and PKA C3 depletion proteomes, exported from the Proteome Discoverer 2.4 software.

• Supplementary file 4. Peptide ratios quantified in the PKA R-AID, and PKG-AID, and SPARK-AID depletion phosphoproteomes. Ratios are reported relative to the vehicle-treated sample within each experiment. Modified Z-scores were calculated by standardizing each value with respect to the median and median absolute deviation within each ratio.

• Supplementary file 5. Oligonucleotides and DNA sequences used in this study.

• MDAR checklist

## Data availability

All data generated and analyzed during this study are included in the manuscript and supporting files. The mass spectrometry proteomics data have been deposited to the ProteomeXchange Consortium via the PRIDE partner repository with the following dataset identifiers: PXD039922, PXD039919, PXD039979, PXD039896, PXD039983, PXD039985, PXD039986, PXD040598, PXD040602, PXD040635, PXD042989, PXD044398, PXD044361; numerical data are provided in *Supplementary files 1–4*. Oligonucleotides and DNA sequences used in this study are provided in *Supplementary file 5*. Source data files have been provided for all immunoblots.

The following datasets were generated:

| Author(s) | Year | Dataset title | Dataset URL | Database and Identifier |
|---|---|---|---|---|
| Herneisen AL, Lourido S | 2023 | SPARK-mNG-mAID Immunoprecipitation Mass Spectrometry | https://www.ebi.ac.uk/pride/archive/projects/PXD039922 | PRIDE, PXD039922 |
| Herneisen AL, Lourido S | 2023 | Immunoprecipitation mass spectrometry of SPARK-mNG | https://www.ebi.ac.uk/pride/archive/projects/PXD039919 | PRIDE, PXD039919 |
| Herneisen AL, Lourido S | 2023 | Immunoprecipitation mass spectrometry of SPARKEL | https://www.ebi.ac.uk/pride/archive/projects/PXD039979 | PRIDE, PXD039979 |
| Herneisen AL, Lourido S | 2023 | PKA C3 immunoprecipitation mass spectrometry | https://www.ebi.ac.uk/pride/archive/projects/PXD039896 | PRIDE, PXD039896 |
| Herneisen AL, Lourido S | 2023 | Proximity labeling of SPARK-TurboID | https://www.ebi.ac.uk/pride/archive/projects/PXD039983 | PRIDE, PXD039983 |
| Herneisen AL, Lourido S | 2023 | Proximity labeling of SPARKEL-TurboID | https://www.ebi.ac.uk/pride/archive/projects/PXD039985 | PRIDE, PXD039985 |
| Herneisen AL, Lourido S | 2023 | Proximity labeling of TurboID-SPARKEL | https://www.ebi.ac.uk/pride/archive/projects/PXD039986 | PRIDE, PXD039986 |
| Herneisen AL, Lourido S | 2023 | Protein abundances following 24h of SPARK depletion in *Toxoplasma gondii* | https://www.ebi.ac.uk/pride/archive/projects/PXD040598 | PRIDE, PXD040598 |
| Herneisen AL, Lourido S | 2023 | *T. gondii* SPARK depletion phosphoproteome | https://www.ebi.ac.uk/pride/archive/projects/PXD040602 | PRIDE, PXD040602 |
| Herneisen AL, Lourido S | 2023 | SPARKEL depletion time course proteome | https://www.ebi.ac.uk/pride/archive/projects/PXD040635 | PRIDE, PXD040635 |
| Herneisen AL, Lourido S | 2023 | PKA C3 depletion phosphoproteomics | https://www.ebi.ac.uk/pride/archive/projects/PXD042989 | PRIDE, PXD042989 |
| Herneisen AL, Lourido S | 2023 | PKA R-dependent phosphoproteome of *T. gondii* treated with the phosphodiesterase inhibitor zaprinast | https://www.ebi.ac.uk/pride/archive/projects/PXD044398 | PRIDE, PXD044398 |
| Herneisen AL, Lourido S | 2023 | PKG-dependent phosphoproteome of *T. gondii* treated with the phosphodiesterase inhibitor zaprinast | https://www.ebi.ac.uk/pride/archive/projects/PXD044361 | PRIDE, PXD044361 |

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

# Appendix 1

## Appendix 1—key resources table

| Reagent type (species) or resource | Designation | Source or reference | Identifiers | Additional information |
|---|---|---|---|---|
| Strain, strain background RH (*T. gondii*) | RH/TIR1 | PMID: 28465425 | | RH/TIR1/ΔKU80/ΔHXGPRT. Mycoplasma negative. |
| Strain, strain background RH (*T. gondii*) | ME49/TIR1 | PMID: 37081202 | | ME49/TIR1/ΔKU80/ΔHXGPRT. Mycoplasma negative. |
| Strain, strain background RH (*T. gondii*) | RH/TIR1/GCaMP6f | PMID: 35484233 | | RH/TIR1/pTUB1-GCaMP6f/ΔKU80/ΔHXGPRT. Mycoplasma negative. |
| Strain, strain background RH (*T. gondii*) | RH/TIR1/GCaMP6f | PMID: 35976251 | | RH/TIR1/pMIC2-MIC2-GLuc-myc-P2A-GCaMP6f/ΔKU80/ΔHXGPRT. Mycoplasma negative. |
| Strain, strain background RH (*T. gondii*) | RH/TIR1/mNG-TurboID | PMID: 36712004 | | RH/TIR1/mNG-TurboID-Ty/ΔKU80/ΔHXGPRT. Mycoplasma negative. |
| Strain, strain background RH (*T. gondii*) | RH/TIR1/PKG-AID | PMID: 28465425 | TGGT1_311360 | RH/TIR1/ΔKU80/ΔHXGPRT/PKG-mAID-HA/HXGPRT. Mycoplasma negative. |
| Strain, strain background RH (*T. gondii*) | RH/TIR1/CDPK1-AID | PMID: 37610220 | TGGT1_301440 | RH/TIR1/ΔKU80/ΔHXGPRT/CDPK1-V5-mNeonGreen-mAID-Ty. Mycoplasma negative. |
| Strain, strain background RH (*T. gondii*) | SPARK-mNG-AID | PMID: 35484233 | TGGT1_268210 | RH/TIR1/ΔKU80/ΔHXGPRT/SPARK-V5-mNG-mAID-Ty. Mycoplasma negative. |
| Strain, strain background RH (*T. gondii*) | SPARK-mNG | This paper | TGGT1_268210 | RH/ΔKU80/ΔHXGPRT/SPARK-V5-mNG-Ty. Mycoplasma negative. Strain construction and validation described in the Materials and Methods section. Strain available upon request. |
| Strain, strain background RH (*T. gondii*) | SPARKEL-TurboID | This paper | TGGT1_291010 | RH/TIR1/ΔKU80/ΔHXGPRT/SPARKEL-TurboID-Ty. Mycoplasma negative. Strain construction and validation described in the Materials and Methods section. Strain available upon request. |
| Strain, strain background RH (*T. gondii*) | TurboID-SPARKEL | This paper | TGGT1_291010 | RH/TIR1/ΔKU80/ΔHXGPRT/3xHA-TurboID-SPARKEL. Mycoplasma negative. Strain construction and validation described in the Materials and Methods section. Strain available upon request. |
| Strain, strain background RH (*T. gondii*) | SPARKEL-Ty | This paper | TGGT1_291010 | RH/TIR1/ΔKU80/ΔHXGPRT/SPARKEL-V5-mNG-Ty. Mycoplasma negative. Strain construction and validation described in the Materials and Methods section. Strain available upon request. |
| Strain, strain background RH (*T. gondii*) | SPARKEL-mNG/SPARK-mCherry | This paper | TGGT1_268210, TGGT1_291010 | RH/TIR1/ΔKU80/ΔHXGPRT/SPARKEL-V5-mNG-Ty/SPARK-V5-mCherry-HA/HXGPRT. Mycoplasma negative. Strain construction and validation described in the Materials and Methods section. Strain available upon request. |
| Strain, strain background RH (*T. gondii*) | SPARKEL-AID | This paper | TGGT1_291010 | RH/TIR1/ΔKU80/ΔHXGPRT/SPARKEL-V5-HaloTag-mAID-Ty. Mycoplasma negative. Strain construction and validation described in the Materials and Methods section. Strain available upon request. |
| Strain, strain background RH (*T. gondii*) | SPARKEL-AID/ME49 | This paper | TGGT1_291010 | ME49/TIR1/ΔKU80/ΔHXGPRT/SPARKEL-V5-HaloTag-mAID-Ty. Mycoplasma negative. Strain construction and validation described in the Materials and Methods section. Strain available upon request. |
| Strain, strain background RH (*T. gondii*) | SPARKEL-AID/GCaMP | This paper | TGGT1_291010 | RH/TIR1/pMIC2-MIC2-GLuc-myc-P2A-GCaMP6f/ΔKU80/ΔHXGPRT/SPARKEL-V5-HaloTag-mAID-Ty. Mycoplasma negative. Strain construction and validation described in the Materials and Methods section. Strain available upon request. |

*Appendix 1 Continued on next page*

*Appendix 1 Continued*

| Reagent type (species) or resource | Designation | Source or reference | Identifiers | Additional information |
|---|---|---|---|---|
| Strain, strain background RH (*T. gondii*) | SPARKEL-AID/SPARK-mCherry | This paper | TGGT1_268210, TGGT1_291010 | RH/TIR1/ΔKU80/ΔHXGPRT/SPARKEL-V5-HaloTag-mAID-Ty/SPARK-V5-mCherry-HA/HXGPRT. Mycoplasma negative. Strain construction and validation described in the Materials and Methods section. Strain available upon request. |
| Strain, strain background RH (*T. gondii*) | SPARK-AID | This paper | TGGT1_268210 | RH/TIR1/ΔKU80/ΔHXGPRT/SPARK-V5-mAID-HA/HXGPRT. Mycoplasma negative. Strain construction and validation described in the Materials and Methods section. Strain available upon request. |
| Strain, strain background RH (*T. gondii*) | SPARK-AID/SPARKEL-mNG | This paper | TGGT1_268210, TGGT1_291010 | RH/TIR1/ΔKU80/ΔHXGPRT/SPARKEL-V5-mNG-Ty/SPARK-V5-mAID-HA/HXGPRT. Mycoplasma negative. Strain construction and validation described in the Materials and Methods section. Strain available upon request. |
| Strain, strain background RH (*T. gondii*) | PKA R-AID | This paper | TGGT1_242070 | RH/TIR1/ΔKU80/ΔHXGPRT/PKA R-V5-mAID-HA/HXGPRT. Mycoplasma negative. Strain construction and validation described in the Materials and Methods section. Strain available upon request. |
| Strain, strain background RH (*T. gondii*) | AID-SPARKEL | This paper | TGGT1_291010 | RH/TIR1/ΔKU80/ΔHXGPRT/3xHA-mAID-SPARKEL. Mycoplasma negative. Strain construction and validation described in the Materials and Methods section. Strain available upon request. |
| Strain, strain background RH (*T. gondii*) | SPARK-TurboID | This paper | TGGT1_268210 | RH/TIR1/ΔKU80/ΔHXGPRT/SPARK-TurboID-Ty. Mycoplasma negative. Strain construction and validation described in the Materials and Methods section. Strain available upon request. |
| Strain, strain background RH (*T. gondii*) | PKA C1-mNG/PKA R-mCherry | This paper | TGGT1_226030, TGGT1_242070 | RH/TIR1/ΔKU80/ΔHXGPRT/PKA C1-mNG-3myc/PKA R-mCherry-Ty. Mycoplasma negative. Strain construction and validation described in the Materials and Methods section. Strain available upon request. |
| Strain, strain background RH (*T. gondii*) | SPARKEL-AID/PKA C1-mNG/PKA R-mCherry | This paper | TGGT1_291010, TGGT1_226030, TGGT1_242070 | RH/TIR1/ΔKU80/ΔHXGPRT/PKA C1-mNG-3myc/PKA R-mCherry-Ty/SPARKEL-V5-HaloTag-mAID-Ty. Mycoplasma negative. Strain construction and validation described in the Materials and Methods section. Strain available upon request. |
| Strain, strain background RH (*T. gondii*) | SPARK-AID/PKA C1-mNG | This paper | TGGT1_268210, TGGT1_226030 | RH/TIR1/ΔKU80/ΔHXGPRT/SPARK-V5-mAID-HA/HXGPRT/PKA C1-mNG-3myc. Mycoplasma negative. Strain construction and validation described in the Materials and Methods section. Strain available upon request. |
| Strain, strain background RH (*T. gondii*) | SPARK-AID/PKA C1-mNG/PKA R-mCherry | This paper | TGGT1_268210, TGGT1_226030, TGGT1_242070 | RH/TIR1/ΔKU80/ΔHXGPRT/SPARK-V5-mAID-HA/HXGPRT/PKA C1-mNG-3myc/PKA R-mCherry-P2A-DHFR. Mycoplasma negative. Strain construction and validation described in the Materials and Methods section. Strain available upon request. |
| Strain, strain background RH (*T. gondii*) | PKA C3-AID | This paper | TGGT1_286470 | RH/TIR1/ΔKU80/ΔHXGPRT/PKA C3-V5-mNG-mAID-Ty. Mycoplasma negative. Strain construction and validation described in the Materials and Methods section. Strain available upon request. |
| Strain, strain background RH (*T. gondii*) | PKA C3-AID/ME49 | This paper | TGGT1_286470 | ME49/TIR1/ΔKU80/ΔHXGPRT/PKA C3-V5-mNG-mAID-Ty. Mycoplasma negative. Strain construction and validation described in the Materials and Methods section. Strain available upon request. |
| Strain, strain background RH (*T. gondii*) | SPARK-AID/ΔBFD1 | This paper | TGGT1_268210, TGGT1_200385 | RH/TIR1/ΔKU80/ΔHXGPRT/ΔBFD1::dTomato/SPARK-V5-mAID-HA/HXGPRT. Mycoplasma negative. Strain construction and validation described in the Materials and Methods section. Strain available upon request. |
| Strain, strain background RH (*T. gondii*) | PKA C3-AID/ΔBFD1 | This paper | TGGT1_286470, TGGT1_200385 | RH/TIR1/ΔKU80/ΔHXGPRT/ΔBFD1::dTomato/PKA C3-V5-mNG-mAID-Ty. Mycoplasma negative. Strain construction and validation described in the Materials and Methods section. Strain available upon request. |
| Strain, strain background RH (*T. gondii*) | PKA C3-mNG/SPARK-AID | This paper | TGGT1_286470, TGGT1_268210 | RH/TIR1/ΔKU80/ΔHXGPRT/PKA C3-mNG-P2A-DHFR/SPARK-V5-mAID-HA/HXGPRT. Mycoplasma negative. Strain construction and validation described in the Materials and Methods section. Strain available upon request. |

*Appendix 1 Continued on next page*

*Appendix 1 Continued*

| Reagent type (species) or resource | Designation | Source or reference | Identifiers | Additional information |
|---|---|---|---|---|
| Cell line (*Homo sapiens*) | Human Foreskin Fibroblasts (HFFs) | ATCC | SCRC-1041 | Mycoplasma negative |
| Antibody | Rat monoclonal (16D7) anti-mCherry | Thermo Fisher | | IFA (1/1,000) |
| Antibody | Mouse monoclonal (32F6) anti-mNG | ChromoTek | | IFA (1/1,000) and WB (1/500) |
| Antibody | Guinea pig monoclonal anti-CDPK1 | Covance | Custom antibody | WB (1/50,000) |
| Antibody | Rabbit polyclonal anti-GAP45 | Lampire Biological Laboratory | | Provided by R. Drew Etheridge Lab. IFA (1/1,000). Used for differentiation assays. |
| Antibody | Rabbit polyclonal anti-GAP45 | PMID: 18312842 | | Provided by Dominique Soldati-Favre Lab. IFA (1/10,000) |
| Antibody | Mouse polyclonal anti-SAG1 | PMID: 3183382 | | Provided by L. David Sibley Lab. IFA (1/1,000). Used for invasion assays. |
| Antibody | Mouse monoclonal anti-V5 | Invitrogen | Invitrogen: R960-25 | WB (1/1000) |
| Antibody | Mouse monoclonal anti-TUB1 (clone 12G10) | Developmental Studies Hybridoma Bank at the University of Iowa | RRID: AB_1157911 | WB (1/5000) |
| Antibody | Rat monoclonal (3F10) anti-HA | Roche | RRID: AB_390919 | IFA (1/1000) |
| Antibody | Mouse monoclonal (BB2) anti-Ty1 | PMID: 8813669 | | Protein G crosslinking (60 μg/1 mg beads) |
| Antibody | Rabbit polyclonal clone WU1614 anti-ALD | PMID: 16923803 | | Provided by L. David Sibley Lab. WB (1/10,000) |
| Antibody | biotinylated dolichos | Vector labs B-1035 | | IFA (1/1000) |
| Chemical compound, drug | Hoechst 33342 | Invitrogen | Invitrogen: H3570 | IFA (1/20,000) |
| Chemical compound, drug | DAPI (4',6-Diamidino-2-Phenylindole, Dihydrochloride) | Invitrogen | Invitrogen: D1306 | |
| Chemical compound, drug | Prolong Diamond | Thermo Fisher | Thermo Fisher: P36965 | |
| Chemical compound, drug | Zaprinast | Calbiochem | Calbiochem: 684500 | Egress assay (500 μM) |
| Chemical compound, drug | A23187 | Calbiochem | Calbiochem: 100105 | Egress assay (2 μM) |
| Chemical compound, drug | Compound 1 | PMID: 12455981 | | Egress assay (as indicated) |
| Chemical compound, drug | Biotin | Sigma Aldrich | Sigma Aldrich: B4501-1G | TurboID (500 μM) |
| Chemical compound, drug | TRIzol reagent | Thermo Fisher | Thermo Fisher Scientific: 15596018 | |
| Commercial assay or kit | S-trap micro | Protifi | Protifi: C02-micro-80 | |
| Commercial assay or kit | Pierce Quantitative Fluorometric Peptide Assay | Thermo Fisher Scientific | Thermo Fisher Scientific: 23290 | |
| Commercial assay or kit | TMT10plex Label Reagent Set | Thermo Fisher Scientific | Thermo Fisher Scientific: 90111 | |
| Commercial assay or kit | TMTpro 16plex Label Reagent Set | Thermo Fisher Scientific | Thermo Fisher Scientific: A44522 | |

*Appendix 1 Continued on next page*

*Appendix 1 Continued*

| Reagent type (species) or resource | Designation | Source or reference | Identifiers | Additional information |
|---|---|---|---|---|
| Commercial assay or kit | EasyPep MS Sample Prep Kits - Maxi | Thermo Fisher Scientific | Thermo Fisher Scientific: A45734 | |
| Commercial assay or kit | High-Select TiO2 Phosphopeptide Enrichment Kit | Thermo Fisher Scientific | Thermo Fisher Scientific: A32993 | |
| Commercial assay or kit | High-Select Fe-NTA Phosphopeptide Enrichment Kit | Thermo Fisher Scientific | Thermo Fisher Scientific: A32992 | |
| Commercial assay or kit | Pierce High pH Reversed-Phase Peptide Fractionation Kit | Thermo Fisher Scientific | Thermo Fisher Scientific: 84868 | |
| Commercial assay or kit | Pierce Streptavidin Magnetic Beads | Thermo Scientific | Thermo Scientific: 88817 | |
| Commercial assay or kit | Sep-Pak C18 Plus Short Cartridge, 360 mg Sorbent per Cartridge, 55–105 µm | Waters | Waters: WAT020515 | |
| Commercial assay or kit | Pierce Protein G Magnetic Beads | Thermo Fisher | Thermo-Fisher: 88847 | |
| Commercial assay or kit | mNeonGreen-Trap Magnetic Agarose | ChromoTek | ChromoTek:ntma-20 | |
| Commercial assay or kit | NEXTFLEX Poly(A) Beads 2.0 Kit | Perkin Elmer | NOVA-512993 | |
| Commercial assay or kit | SuperScriptIII Reverse Transcriptase | Invitrogen | Invitrogen: 18080044 | |
| Commercial assay or kit | PowerUp SYBR Green master mix for qPCR | Applied Biosystems | A25779 | |
| Software, algorithm | Proteome Discoverer 4.2 | Thermo Fisher | | |
| Software, algorithm | R version 4.0 | R Foundation for Statistical Computing | | |
| Software, algorithm | Prism 8 | GraphPad | | |
| Software, algorithm | HHPRED | PMID: 29258817 | | |
| Software, algorithm | MEGA | PMID: 33892491 | | |
| Software, algorithm | ClustalW | PMID: 17846036 | | |
| Software, algorithm | TrimGalore version 0.6.7 | doi:10.5281/zenodo.7598955 | | |
| Software, algorithm | STAR version 2.7.1 a | PMID: 23104886 | | |
| Software, algorithm | featureCounts version 1.6.2 | PMID: 24227677 | | |
| Software, algorithm | DESeq2 version 1.30.1 | PMID: 25516281 | | |

