## [Editor Report · eLife assessment]

This **fundamental** study identifies protein kinases in the parasitic protozoan, *Toxoplasma gondii* that are required for parasite invasion of host cells and differentiation to drug-resistant chronic stages. The use of advanced proteomic and functional approaches provides **compelling** evidence for the proposed signalling pathway, although additional analyses are needed to fully validate some findings. The work will be of broad interest to cell biologists and parasitologists with an interest in cell signalling and environmental sensing.

---

## [Referee Report · Reviewer #1 (Public Review)]

Summary:

Herneisen et al characterise the Toxoplasma PDK1 orthologue SPARK and an associated protein SPARKEL (cute name) in controlling important fate decisions in Toxoplasma. Over recent years this group and others have characterised the role of cAMP and cGMP signalling in negatively and positively regulating egress, motility and invasion, respectively. This manuscript furthers this work by showing that SPARK and SPARKEL likely act upstream, or at least control the levels of the cAMP and cGMP-dependent kinases PKA and PKG, respectively, thus controlling the transition of intracellular replicating parasites into extracellular motile forms (and back again).

The authors use quantitative (phospho)proteomic techniques to elegantly demonstrate the upstream role of SPARK in controlling cAMP and cGMP pathways. They use sophisticated analysis techniques (at least for parasitology) to show the functional association between cGMP and cAMP signalling pathways. They therefore begin to unify our understanding of the complicated signalling pathways used by Toxoplasma to control key regulatory processes that control the activation and suppression of motility. The authors then use molecular and cellular assays on a range of generated transgenic lines to back up their observations made by quantitative proteomics that are clear in their design and approach.

The authors then extend their work by showing that SPARK/SPARKEL also control PKAc3 function. PKAc3 has previously been shown to negatively regulate differentiation into bradyzoite forms and this work backs up and extends this finding to show that SPARK also controls this. The authors conclude that SPARK could act as a central node of regulation of the asexual stage, keeping parasites in their lytic cell growth and preventing differentiation. Whether this is true is beyond the scope of this paper and will have to be determined at a later date.

Strengths:

This is an exceptional body of work. It is elegantly performed, with state-of-the-art proteomic methodologies carefully being applied to Toxoplasma. Observations from the proteomic datasets are masterfully backed up with validation using quantitative molecular and cellular biology assays.

The paper is carefully and concisely written and is not overreaching in its conclusions. This work and its analysis set a new benchmark for the use of proteomics and molecular genetics in apicomplexan parasites.

Weaknesses:

There are no weaknesses in this paper.

---

## [Referee Report · Reviewer #2 (Public Review)]

Summary:

The manuscript by Herneisen et al. examines the Toxoplasma SPARK kinase orthologous to mammalian PDK1 kinase. The extracellular signals trigger cascades of the second messengers and play a central role in the apicomplexan parasites' survival. In Toxoplasma, these cascades regulate active replication of the tachyzoites, which manifests as acute toxoplasmosis, or the development into drug-resilient bradyzoites characteristic of the chronic stage of the disease. This study focuses on the poorly understood signaling mechanisms acting upstream of such second messenger kinases as PKA and PKG. The authors showed that similar to PDK1, Toxoplasma SPARK likely regulates several AGC kinases.

Strengths:

The study demonstrated a strong association of the SPARK kinase with the SPARKL factor and an uncharacterized AGC kinase. Using a set of standard assays, the authors determined the SPARK /SPARLS role in parasite egress, invasion, and bradyzoite differentiation.

Weaknesses:

Although the revised manuscript has significantly improved, the primary concern of incomplete data analysis still needs to be addressed.

---

## [Referee Report · Reviewer #3 (Public Review)]

Summary:

This paper focuses on the roles of a toxoplasma protein (SPARKEL) with homology to an elongin C and the kinase SPARK that it interacts with. They demonstrate that the two proteins regulate the abundance of PKA and PKG and that depletion of SPARKEL reduces invasion and egress (previously shown with SPARK), and that their loss also triggers spontaneous bradyzoite differentiation. The data are overall very convincing and will be of high interest to those who study Toxoplasma and related apicomplexan parasites.

Strengths:

The study is very well executed with appropriate controls. The manuscript is also very well and clearly written. Overall, the work clearly demonstrates that SPARK/SPARKEL regulate invasion and egress and that their loss triggers differentiation.

Comments on the revised version:

The authors have addressed my concerns.

---

## [Author Response]

The following is the authors’ response to the original reviews.

**eLife assessment**
This study defines a fundamental aspect of protein kinase signalling in the protist parasite *Toxoplasma gondii* that is required for acute and chronic infections. The authors provide compelling evidence for the role of SPARK/SPARKEL kinases in regulating cAMP/cGMP signalling, although evidence linking the loss of these kinases to changes in the phosphoproteome is incomplete. Overall, this study will be of great interest to those who study Toxoplasma and related apicomplexan parasites.

We thank the reviewers for their thoughtful and positive evaluation of our work. Below, we have addressed all of the public reviews and recommendations for the authors in point-by-point responses. Additionally, we include with this resubmission RT-qPCR data where we observe no significant change in transcript levels for the relevant AGC kinases, supporting the hypothesis that SPARK/SPARKEL–regulation is post-translational.

**Public Reviews:**

**Reviewer #1 (Public Review):**
Summary:Herneisen et al characterise the Toxoplasma PDK1 orthologue SPARK and an associated protein SPARKEL in controlling important fate decisions in Toxoplasma. Over recent years this group and others have characterised the role of cAMP and cGMP signalling in negatively and positively regulating egress, motility, and invasion, respectively. This manuscript furthers this work by showing that SPARK and SPARKEL likely act upstream, or at least control the levels of the cAMP and cGMP-dependent kinases PKA and PKG, respectively, thus controlling the transition of intracellular replicating parasites into extracellular motile forms (and back again).The authors use quantitative (phospho)proteomic techniques to elegantly demonstrate the upstream role of SPARK in controlling cAMP and cGMP pathways. They use sophisticated analysis techniques (at least for parasitology) to show the functional association between cGMP and cAMP signalling pathways. They therefore begin to unify our understanding of the complicated signalling pathways used by Toxoplasma to control key regulatory processes that control the activation and suppression of motility. The authors then use molecular and cellular assays on a range of generated transgenic lines to back up their observations made by quantitative proteomics that are clear in their design and approach.The authors then extend their work by showing that SPARK/SPARKEL also control PKAc3 function. PKAc3 has previously been shown to negatively regulate differentiation into bradyzoite forms and this work backs up and extends this finding to show that SPARK also controls this. The authors conclude that SPARK could act as a central node of regulation of the asexual stage, keeping parasites in their lytic cell growth and preventing differentiation. Whether this is true is beyond the scope of this paper and will have to be determined at a later date.Strengths:This is an exceptional body of work. It is elegantly performed, with state-of-the-art proteomic methodologies carefully being applied to Toxoplasma. Observations from the proteomic datasets are masterfully backed up with validation using quantitative molecular and cellular biology assays.The paper is carefully and concisely written and is not overreaching in its conclusions. This work and its analysis set a new benchmark for the use of proteomics and molecular genetics in apicomplexan parasites.Weaknesses:This reviewer did not identify any weaknesses.
**Reviewer #2 (Public Review):**
Summary:The manuscript by Herneisen et al. examines the Toxoplasma SPARK kinase orthologous to mammalian PDK1 kinase. The extracellular signals trigger cascades of the second messengers and play a central role in the apicomplexan parasites' survival. In Toxoplasma, these cascades regulate active replication of the tachyzoites, which manifests as acute toxoplasmosis, or the development into drug-resilient bradyzoites characteristic of the chronic stage of the disease. This study focuses on the poorly understood signaling mechanisms acting upstream of such second messenger kinases as PKA and PKG. The authors showed that similar to PDK1, Toxoplasma SPARK appears to regulate several AGC kinases.Strengths:The study demonstrated a strong association of the SPARK kinase with an elongin-like SPARKEL factor and an uncharacterized AGC kinase. Using a set of standard assays, the authors determined the SPARK/SPARKEL role in parasite egress and invasion. Finally, the study presented evidence of the SPARK/SPARKEL involvement in the bradyzoite differentiation.Weaknesses:Although the study can potentially uncover essential sensing mechanisms operating in Toxoplasma, the evidence of the SPARK/SPARKEL mechanisms is weak. Specifically, due to incomplete data analysis, the SPARK/SPARKEL-dependent phosphoregulation of AGC kinases cannot be evaluated. The manuscript requires better organization and lacks guidance on the described experiments. Although the study is built on advanced genetics, at times, it is unnecessarily complicated, raising doubts rather than benefiting the study.

The evidence for the SPARK/SPARKEL interaction is demonstrated through diverse experimental approaches that are internally consistent. Five separate mass spectrometry experiments, with replicates and appropriate controls, with tags on either SPARK or SPARKEL, showed that SPARK and SPARKEL form a strong interaction (Figure 1A, 1D, 1E; Figure 1—figure supplement 1). Global mass spectrometry experiments assessing the impact of SPARK or SPARKEL depletion showed similar features (a reduction in PKG and PKA abundance and up-regulation of bradyzoite-associated proteins; Figure 3C–D). The phenotypes associated with SPARK and SPARKEL depletion phenocopy one another in all cell biological assays we tested (Figure 2A, 2D and PMID: 35484233; Figure 2E–J; Figure 4E–F; Figure 6A–B). Measuring the abundance of SPARK and SPARKEL in unenriched samples was challenging, but immunoblotting and proteomics suggest that depletion of one factor leads to down-regulation of the other (Figure 2B, 2C; Figure 3—figure supplement 1), which explains the genetic and cell biological phenocopying described above. We note that “further biochemical studies are required to discern the regulatory interactions between SPARK and SPARKEL” (first submission lines 590-591) and are beyond the scope of this work.

The evidence for SPARK/SPARKEL regulation of AGC kinase activity is demonstrated through diverse experimental approaches that are also internally consistent. PKA C1 and PKG abundance levels decrease in parasites depleted of SPARK/SPARKEL, as measured by mass spectrometry (Figure 3A and 3C) and cell-based assays for PKA C1/R (Figure 4D–F). Comparisons of the global SPARK-, PKA R-, PKG-, and PKA C3-depleted phosphoproteomes suggest that PKA and PKG activity is reduced upon SPARK depletion whereas the activity of an unrelated factor (PP1) is unaffected (Figure 4G–H, Figure 4—figure supplement 1, Figure 5D–E, Figure 7I–J). Parasites depleted of SPARK are hypersensitized to a PKG inhibitor (Figure 5B–C). SPARK, PKA, and PKG are proximal in cellulo (Figure 3I) and SPARK co-purifies with PKA C3 (Figure 7A). The kinetic-phase phenotypes associated with SPARK and SPARKEL depletion (PMID: 32379047, Figure 2A, 2D–2J) are consistent with reduced PKG activity (PMID: 28465425) and only develop after PKG has been depleted as shown by proteomics experiments (Figure 2E-J and Figure 3C). Other studies have shown that the effects of reduced PKG activity are dominant to reduced PKA C1 activity (PMID: 29030485). The replicative-phase phenotypes associated with SPARK and SPARKEL depletion are consistent with reduced PKA C3 activity (PMID: 27247232 and herein). Mechanistically, PKG and PKA C1 activity must be lower in SPARK-depleted parasites because the abundances of these kinases are lower (Figure 3A, 3C). The mechanism of regulation may be more complex in the case of PKA C3, as SPARK depletion did not cause a reduction in PKA C3 abundance as measured by cellular assays (Figure 7B–F), but PKA C3 activity decreased (Figure 7I–K). We concede that multiple mechanisms may lead to the reduction in PKA C1 and PKG abundances, such as decreased activation loop phosphorylation and autophosphorylation at other stabilizing sites or enhanced ubiquitin ligase activity leading to active degradation of the kinases; we have moved speculation regarding such mechanisms to the Discussion.

Although the reviewer commented that the manuscript “requires better organization” in the public review, no specific recommendations were provided to the authors. Therefore, we did not change the organization of the manuscript. We added an additional paragraph to the Discussion to reiterate key findings: “A prior study identified SPARK as a regulator of parasite invasion and egress following 24 hours of kinase depletion (Smith et al., 2022). Unexpectedly, we observed that three hours of SPARK or SPARKEL depletion were insufficient to impact *T. gondii* motility or calcium-dependent signaling, indicating that the phenotypes associated with SPARK and SPARKEL depletion develop over time. Quantitative proteomics revealed that PKA and PKG abundances began to decrease after more than three hours of SPARK depletion. Proximity labeling experiments also suggested that SPARK, PKA, and PKG are spatially associated within the parasite cell. We propose a model in which SPARK down-regulation coincides with reduced PKG and PKA activity due to diminished protein levels.” This work built upon genetic and proteomic approaches recently described by our group, which we cited in the text and extensive methods section. We added additional experimental detail where noted in the reviewer’s recommendations to the authors.

The study utilizes advanced genetics because biochemical tools for eukaryotic parasites are limited. For example, no antibodies for *T. gondii* SPARK, PKA subunits, or PKG exist; to say nothing of phosphosite-specific antibodies, which are common in the mammalian cell signaling field. Therefore, to measure the relationship between SPARK, SPARKEL, and PKA subunits, we had to generate strains in which multiple proteins were tagged with epitopes for downstream analysis. The genetic experiments included appropriate controls and were internally consistent with results obtained using orthogonal approaches, such as mass spectrometry.

**Reviewer #3 (Public Review):**
Summary:This paper focuses on the roles of a toxoplasma protein (SPARKEL) with homology to an elongin C and the kinase SPARK that it interacts with. They demonstrate that the two proteins regulate the abundance of PKA and PKG, and that depletion of SPARKEL reduces invasion and egress (previously shown with SPARK), and that their loss also triggers spontaneous bradyzoite differentiation. The data are overall very convincing and will be of high interest to those who study Toxoplasma and related apicomplexan parasites.Strengths:The study is very well executed with appropriate controls. The manuscript is also very well and clearly written. Overall, the work clearly demonstrates that SPARK/SPARKEL regulate invasion and egress and that their loss triggers differentiation.Weaknesses:(1) The authors fail to discriminate between SPARK/SPARKEL acting as negative regulators of differentiation as a result of an active role in regulating stage-specific transcription/translation or as a consequence of a stress response activated when either is depleted

We demonstrate a novel function for SPARK and SPARKEL as negative regulators of differentiation. The pathways leading to differentiation are being actively studied. Up-regulation of a positive transcriptional regulator of chronic differentiation, BFD1, is sufficient to trigger differentiation in vitro in the absence of other stressful growth conditions (PMID: 31955846). SPARK or SPARKEL depletion results in up-regulation of proteins that are up-regulated upon BFD1 overexpression. Whether BFD1 overexpression or SPARK and SPARKEL depletion triggers cellular stress pathways is beyond the scope of the current work, which focused instead on the immediate effect of these pathways on AGC kinases. Study of the effect of the various kinases on the parasite phosphoproteome shows that the putative targets of PKA C3 are specifically downregulated upon SPARK knockdown, indicating PKA C3 activity is indeed decreased in the latter condition.

(2) The function of SPARKEL has not been addressed. In mammalian cells, Elongin C is part of an E3 ubiquitin ligase complex that regulates transcription and other processes. From what I can tell from the proteomic data, homologs of the Elongin B/C complex were not identified. This is an important issue as the authors find that PKG and PKA protein levels are reduced in the knockdown strains

Our experiments suggest that SPARK and SPARKEL form a complex, and down-regulation of one complex member leads to down-regulation of the other. Thus in all tested assays, knockdown of SPARK and SPARKEL phenocopy one another. Further biochemical and structural work will be required to determine the mechanism by which SPARKEL regulates SPARK.

Nearly all studies of the function of elongin C have been conducted in mammalian cells. Proteins with elongin C domains may serve alternative and unexplored functions in unicellular eukaryotes. We searched for the presence of Elongin A/B and known Elongin C complex members in the *T. gondii* genome and were unable to identify orthologs, explaining why these proteins were not identified in mass spectrometry experiments. Please see our response in Recommendations for the Authors, Reviewer 3 point 2.

Beyond the concerns raised by the review team, we have identified and corrected the following errors or omissions in the first submission of the manuscript:

- Line 176 of the first submission referred to a “peptide sequence match (PSM)”, which we have changed to “peptide-spectrum match”.

- We recolored and relabeled the lines in Figure 5A so that it is easier to match a specific peptide with a specific line; and also corrected a mislabeling.

- Figure 7B SPARK panel was incorrectly centered. The raw files can be viewed in Figure 7—source data 2.

- Figure 7—figure supplement 1D was missing an x-axis label.

- Line 1172 referred to “Supplementary File X”, which we corrected to “Supplementary File 3”.

- We have updated references to preprints that have since been published, including PMID: 38093015, 37933960, 37966241, and 37610220.

**Editors comments:**
The proteomics data reported in this study underpin the major findings and are very comprehensive. As noted in the reviews, it is strongly recommended that the authors normalize the levels of detected phosphopeptides against the levels of the parent protein in the different mutant lines in order to identify changes in protein phosphorylation that are linked to protein kinase activity rather than protein degradation. A focus on changes that occur at early time points following protein knock-down may also help to identify the main targets of each kinase.

Please see our response to Reviewer 2 Recommendations for the Authors, points 1 and 2.

**Reviewer #1 (Recommendations For The Authors):**
During my reading, I only found one small mistake. In Figure 7F, the x-axis is missing the word 'PKA'.

We have updated the x-axis to read “SPARK-AID/PKA C3-mNG (h. + IAA)”.

All information, code, and reagents are clearly explained.

**Reviewer #2 (Recommendations For The Authors):**
How the phosphoproteome was analyzed needs to be clarified. The normalization step, computing the ratio of the phosphopeptide to the protein (peptide) intensity, appears omitted. It is the most critical step of the analysis. The minor shifts between protein and phosphosite intensity seem negligible, as seen in Figure 4 AB. The significant changes can only be deduced by calculating this ratio. In the current state, the presented results are inconclusive. The manuscript contains overreaching and often unsupported statements because the data has not been appropriately filtered. Related to this topic, it is advisable to use well-accepted terminology and complete words when describing proteome and phosphoproteome. The interexchange of a "peptide" and a "phosphopeptide" in the text confuses and misleads.

To clarify the phosphoproteome analysis:

We cite a previous description of the phosphoproteomics sample preparation workflow (lines 1124-1125 of the first submission for example). Our quantitative phosphoproteomics experiments comprise two datasets generated from the same multiplexed samples. The samples were split at the point of phosphopeptide enrichment. Ninety-five percent of the samples were subjected to phosphopeptide enrichment (titanium dioxide followed by nickel affinity chromatography; “enriched samples”). Five percent of the samples were reserved as a reference for the non-enriched proteome (“non-enriched samples”). To clarify this point, we have added the sentences “Approximately 95% of the proteomics sample was used for phosphopeptide enrichment” and “The remaining 5% of the sample was not subjected to the phosphopeptide enrichment protocol” to the Methods sections, after describing the multiplexing steps.

The samples were fractionated separately and run separately on an LC-MS system, which is described in the Methods section, for example lines 1130-1149 of the first submission. Raw files of the phosphopeptide-enriched and unenriched samples were analyzed separately, which is described in the Methods section, for example lines 1151-1158 of the first submission. To clarify this point, we have added the sentence “Raw files of the phosphopeptide-enriched and unenriched samples were analyzed separately” to the Methods sections. Many of the search parameters and descriptions of normalization and protein abundances were described in lines 1085-1093 of the first submission in reference to the 24h SPARK depletion proteome. We added this information to the description of the SPARK depletion time course phosphoproteome data analysis: “The allowed mass tolerance for precursor and fragment ions was 10 ppm and 0.02 Da, respectively. False discovery was assessed using Percolator with a concatenated target/decoy strategy using a strict FDR of 0.01, relaxed FDR of 0.05, and maximum Delta CN of 0.05. Only unique peptide quantification values were used. Co-isolation and signal-to-noise thresholds were set to 50% and 10, respectively. Normalization was performed according to total peptide amount. In the case of the unenriched samples, protein abundances were calculated from summation of non-phosphopeptide abundances.”

We hope that this clarifies how the unenriched sample protein-level abundances were calculated. When we discuss “protein abundance”, we are referencing the unenriched sample summed non-phosphopeptide abundance. Our phosphoproteome analysis was based only on phosphopeptides, as our phosphopeptide enrichment resulted in 99% efficiency, and peptides lacking phosphorylation sites were filtered out before subsequent analyses. We used “peptide” and “phosphopeptide” interchangeably because the only peptide-level analysis performed was based on phosphopeptide abundances. We have changed any mention of “peptide” to “phosphopeptide” in the main text.

“The normalization step, computing the ratio of the phosphopeptide to the protein (peptide) intensity, appears omitted. It is the most critical step of the analysis.”:

Unlike common differential gene expression analysis pipelines, proteomics analysis pipelines are not settled. Many analyses do not perform peptide-to-parent-protein corrections; some normalize phosphopeptide abundances to parent protein abundances calculated from summing non-phosphopeptides or a combination of phosphopeptide and non-phosphopeptides on an ad hoc basis; some calculate global normalization factors based on regressions of protein and phosphopeptide abundances or other pairwise comparisons. A caveat of protein normalization of phosphopeptides is that it over-corrects cases in which protein abundance and phosphorylation are interdependent, as is the case for auto-phosphorylation and some activation loop phosphorylations (PMID: 37394063). We used the approach that retained the greatest complexity of the data, which is to not normalize abundances across different mass spectrometry experiments and discard information that was not in the overlap. We have updated Supplementary File 3.3 to include protein-level quantification values (from Supplementary File 3.2) if measured.

We clarified that the phosphopeptide abundances and protein-level abundances were derived from different datasets that were each internally normalized (globally centered by total peptide amount). Protein-level abundances were summed from non-phosphopeptide abundances. The calculated log2 changes are based on the globally centered data within each dataset. We analyzed the kinetic profiles of changing phosphopeptide abundances relative to a control using approaches similar to those described for several recent temporally resolved *T. gondii* phosphoproteomes (e.g. PMID: 37933960, 35976251, 36265000, 29141230) and as described in the Methods. The approach does not first correct for unenriched-sample parent protein abundance—in some applications, unenriched samples are not collected at all; instead, phosphopeptide ratios are median-normalized to non-phosphopeptide ratios (quantified due to inefficient phosphopeptide enrichment) and are individually tested against the null distribution of non-phosphopeptide ratios (e.g. PMID: 36265000, 29141230). We did not use this approach because our phosphopeptide enrichment was 99% efficient (18518 phosphopeptides of 18758 peptides with quantification values). In several cases using our approach, parent protein abundance is not quantified in the unenriched proteome dataset, but phosphopeptides are reliably quantified in the enriched proteome dataset. We note that phosphopeptide abundance changes can be difficult to interpret in such cases, e.g. in the first submission lines 178-186 and 193-194. We have added similar text to the results noting that in the case of PKA and PKG, both unenriched parent protein and enriched phosphopeptide abundances decreased (see below). We have also moved speculation about whether SPARK phosphorylates the activation loop of PKA and PKG, or whether the down-regulation of PKA and PKG arises from indirect effects, to the Discussion.

We have moved comparisons of protein and phosphopeptide abundances from the Results to the Discussion. We added the following sentences to the result section Clustering of phosphopeptide kinetics identifies seven response signatures: “Because non-phosphopeptide and phosphopeptide abundances were quantified in different mass spectrometry experiments, it is challenging to compare the rates of phosphopeptide and parent protein abundance changes, especially when phosphorylation status and protein stability are interconnected. In general, both PKA C1, PKA R, and PKG protein and phosphosite abundances decreased following SPARK depletion (Figure 3—figure supplement 1), as discussed further below. We also observed down-regulation of phosphosite and protein abundances of a MIF4G domain protein.” Figure 3—figure supplement 1E is a new panel that shows PKA C1, PKA R, and PKG phosphopeptide and parent protein abundances along with global changes in phosphopeptide and parent protein abundances in the cases which both were quantified. We changed lines 278-282 in the first submission to “The SPARK depletion time course phosphoproteome showed a reduction in the abundance of PKA C1 T190 and T341, which are located in the activation loop and C-terminal tail, respectively (Figure 4A). Several phosphosites residing in the N terminus of PKA R (e.g. S17, S27, and S94) also decreased following SPARK depletion (Figure 4B).” We changed lines 313-315 in the first submission to “The SPARK depletion time course phosphoproteome showed a reduction in the abundance of several phosphosites residing in the N terminus of PKG as well as T838, which corresponds to the activation loop (Figure 5A). By contrast, S105 did not greatly decrease, and S40 abundance slightly increased.”

The description of experiments should be more detailed. For example, the 3, 8, and 24 h treatments were used reversely; thus, they should be emphasized as time points before natural egress. Consequently, it seems that 3h treatment should be prioritized, given the SPARK/SPARKEL role in egress/invasion. Unexpectedly, the study draws more attention to a 24-hour treatment. If the AID-SPARK/SPARKEL is eliminated within 1h, parasites undoubtedly accumulate numerous secondary defects during a prolonged 23h deprivation. Since the SPARK pathway activates kinase/phosphatase cascades, the 24h data is likely overwhelmed with the consequences of the long-term complex degradation, making it a poor source of the putative SPARK substrates. Likewise, the downregulation of PKA observed in the 8 hours after SPARK depletion may be an indirect effect of the SPARK degradation. The direct effects and immediate substrates should be detectable within 2-3h of auxin treatment of the nearly egressing cultures.

The first submission described how parasites were harvested at 32 hours post-infection with 0, 3, 8, or 24 hours of IAA treatment (lines 157-160, 1097-1110, and Figure 3B). To reiterate this experimental detail, we have added “harvested 32 hours post-infection” to the sentence “...quantitative proteomics with tandem mass tag multiplexing that included samples with 0, 3, 8, and 24 hours of SPARK or SPARKEL depletion” and similarly in the figure legend. The time points are unrelated to natural egress because the experiment was terminated at 32 hours post-infection, which is earlier than the window typically used to study natural egress under these conditions (40-48 hours post-infection). We chose to terminate the experiment before natural egress to better localize phosphopeptide changes related to SPARK depletion. The phosphoproteome undergoes dramatic reorganization during egress due to the activity of myriad kinases and phosphatases (see PMID: 35976251, 37933960, and 36265000), which would have likely complicated the signal.

A pivotal result motivating time-course experiments and analysis was that SPARK/SPARKEL's role in egress and invasion emerges only after an extended depletion period (Figure 2E–J, first submission lines 126-145). The 24h depletion was used in the experimental system that first identified SPARK as a regulator of egress, which motivated our initial experiments, as stated in the first submission lines 126-144 and 149-151. We draw attention to the observation that SPARK and SPARKEL phenotypes develop over time in the first submission, lines 137-145. The role for SPARK/SPARKEL in egress/invasion does not manifest at 3h depletion; it manifests at 24h depletion. To ensure that this point is not overlooked by the reader, we have created a new heading in the Results section (SPARK and SPARKEL depletion phenotypes develop over time) for the paragraph that was previously lines 137-145. The remainder of the manuscript integrates data from proteomic, genetic, and cell-based assays across temporal dimensions to build a working model of how the phenotypes associated with SPARK depletion develop over time.

Underpinning this comment is an assumption that phosphopeptides that decrease the most rapidly following a kinase’s depletion are direct substrates, whereas phosphopeptides that decrease with slower kinetics are not. This is not always the case. Consider a kinase that phosphorylates sites on substrate A and substrate B. The site on substrate A is also the target of a phosphatase, whereas the site on substrate B is recalcitrant to phosphatase activity. If the kinase were inhibited, then the site on substrate A would be actively dephosphorylated. As measured by a phosphoproteomics experiment, the abundance of the substrate A phosphopeptide would drop rapidly due to the inactivity of the kinase and activity of the phosphatase. In the text, we called such sites “constitutively regulated” or dynamic—they are actively dephosphorylated and phosphorylated within a short timeframe. The phosphosite on substrate B is comparatively static; once it is phosphorylated by the kinase, it is unaffected by subsequent inhibition of the kinase. Only newly synthesized substrate B molecules would be affected by kinase inhibition. As measured by a phosphoproteomics experiment, the abundance of the substrate B phosphopeptide would drop more gradually after kinase inhibition, as the unphosphorylated peptide is found only on newly synthesized proteins that were not previously exposed to kinase activity. An example of the scenario described for substrate A would be that of yeast Cdk1 T14/Y15, which is phosphorylated by Wee1 and dephosphorylated by Cdc25 (e.g. PMID: 7880537). An example of the scenario described for substrate B would be that of the human PKA C activation loop T197, which is phosphorylated by PDK1 and is phosphatase-resistant under physiological conditions (e.g. PMID: 22493239, 15533936).

Both substrate A and B may be “direct” and functionally relevant targets of the kinase. Categorizing substrates as “immediate” is comparatively less informative in this context (although it may be relevant when studying fast, synchronized processes with high temporal resolution, such as induced Plasmodium spp. gametocyte activation or stimulation of *T. gondii* secretion). Furthermore, our earlier experiments had shown that the role for SPARK/SPARKEL in motility manifests after 3h depletion and is complete by 24h depletion. By this logic, we were most interested in the candidates showing differences at these time points. We conducted proximity labeling experiments to identify the overlap of proteins that exhibited SPARK-dependent decreases in the global proteomics and were also proximal to SPARK in space (first submission Figure 3I and lines 260-275), thus revealing a prioritized list of candidates, which included PKG and PKA. When technically feasible, we included a temporal dimension to follow-up experiments, rather than relying on a 24h terminal comparison (e.g. Figure 4E–H, Figure 5D–E, Figure 7D–F, Figure 7I–K; all first submission).

Fig2 (B and C). What antibodies had been used to detect tagged proteins? There is a concern regarding the use of multiple tags attached to the same protein to the point that it doubles the size of the studied protein. The switch of the mobility of the SPARK and SPARKEL on the WB due to a change in MW adds to the confusion. Furthermore, the study did not use all the fused epitopes (e.g., HA). At the same time, the same V5 tag was used to detect two factors in the same parasite. Although the controls are provided, it does not eliminate the possibility that the second band on the WB results from one protein degradation rather than the presence of two individual proteins. Different tags should be used to confirm the co-expression of two proteins. Panel E is missing the X-axis label.

Figure 2B was incorrectly labeled; the labels corresponding to SPARK and SPARKEL were switched. We corrected this error in the revised figures. The antibodies used were mouse monoclonal anti-V5 as described in the key resources table of the first submission. We added “V5” to Figure 2A and 2B. Regarding the effect of the tagging payload attached to the proteins, we have included in all assays a control relative to a parental strain (TIR1) without a tagging payload, and additionally included internal controls within tagged strains to calculate dependency of a phenotype on IAA treatment. The western blots in Figure 2B and 2C are from two different strains and experiments. The strains and experiments are described in the first submission main text (lines 113-124), the figure legend (lines 1847-1850), the key resources table, and the methods (lines 650-664, 872-891). A description of the SPARK-AID/SPARKEL-mNG strain was included in the key resources table but omitted in the methods. We therefore added the following section to the Methods:

“SPARKEL-V5-mNG-Ty/SPARK-V5-mAID-HA/RHΔku80Δhxgprt/TIR1

The HiT vector cutting unit gBlock for SPARKEL (P1) was cloned into the pALH193 HiT empty vector. The vector was linearized with BsaI and co-transfected with the pSS014 Cas9 expression plasmid into SPARK-V5-mAID-HA/RHΔku80Δhxgprt/TIR1 parasites. Clones were selected with 1 µM pyrimethamine and isolated via limiting dilution to generate the SPARKEL-V5-mNG-Ty/SPARK-V5-mAID-HA/RHΔku80Δhxgprt/TIR1 strain. Clones were verified by PCR amplification and sequencing of the junction between the 3′ end of SPARKEL (5’-GGGAGGCCACAACGGCGC-3’) and 5′ end of the protein tag (5’-gggggtcggtcatgttacgt-3’).”

To clarify the expected MW of each species, we have added the following text to the Methods:

“The expected molecular weight of SPARKEL-V5-HaloTag-mAID-Ty is 66 kDa, from the 42.7 kDa tagging payload and 23.3 kDa protein sequence. The expected molecular weight of SPARK-V5-mCherry-HA is 89.7 kDa, from the 31.9 kDa tagging payload and 57.8 kDa protein sequence. The expected molecular weight of SPARK-V5-mAID-HA is 71.3 kDa, from the 13.5 kDa tagging payload and 57.8 kDa protein sequence. The expected molecular weight of SPARKEL-V5-mNG-Ty is 55.2 kDa, from the 31.9 kDa tagging payload and 23.3 kDa protein sequence.”

SPARK and SPARKEL are lowly expressed, which may have been compounded by basal degradation due to the AID tag (see for example Figure 3—figure supplement 1D of the first submission). We attempted several immunoblot conditions and antibodies, and only the V5 antibody proved effective in recognizing these proteins above the limit of detection. For this reason, we included an additional single-tagged control in each immunoblot experiment. Uncropped images of the blots are included in the first submission as Figure 2—figure supplement 1D and E and as Figure 2 source data. We added the following statement to the results section of the text:

“However, SPARK and SPARKEL abundances are low and approach the limit of detection. We could only detect each protein by the V5 epitope. Although our experiments included single-tagged controls, we cannot formally eliminate the possibility that SPARK-AID yields degradation products that run at the expected molecular weight of SPARKEL. More sensitive methods, such as targeted mass spectrometry, may be required to measure the absolute abundance and stoichiometries of SPARK and SPARKEL.”

We added “h +IAA” to the x-axis of panel 2E.

Fig. 3. There is plentiful proteomic data on the factor-depleted parasites. Can it be used to confirm the co-degradation of the SPARK/SPARKEL complex components? This figure mainly includes quality control data that can be moved to Supplement. Did you detect SPARKEL in the TurboID experiment described in panel I? The plot shows only an AGC kinase.

SPARK and SPARKEL are lowly expressed, and we often do not detect SPARK or SPARKEL peptides with quantification values in complex samples (such as global depletion proteomes and phosphoproteomes; IPs and streptavidin pull-downs are comparatively less complex, with IPs being the least complex samples). We discussed this caveat in the first submission lines 178-186. To additionally clarify this point, we have added “We were unable to measure SPARK or SPARKEL abundances in this proteome” earlier in the text.

We consider the figure panels relevant to the discussion in the text.

SPARKEL was not quantified in the SPARK-TurboID experiment (Supplementary File 2). We have added “SPARKEL was not quantified in this experiment” to the text. “Not quantified” is a different outcome from “quantified but not enriched”. The interaction between SPARK and SPARKEL is supported by five other independent interaction experiments in which SPARKEL was quantified (Figure 1A, 1D, 1E; and Figure 1—figure supplement 1). The added insight from the SPARK proximity labeling experiments comes from integration with the global proteomics, which suggests that AGC kinases are in proximity to SPARK and exhibit SPARK-dependent stability and hence activity. The logic of the proximity labeling experiment is described in lines 258-275 of the first submission.

Fig. 6G is missing deltaBDF1 control for unbiased evaluation of the SPARK KD effect.

The logic of this experiment was to evaluate whether excess differentiation caused by SPARK and PKA C3 depletion (Figure 6A and 6B) was dependent on the BFD1 circuit. The ∆bfd1 phenotype is well-established under these experimental conditions: parasites lacking BFD1 do not differentiate under spontaneous or alkaline conditions (e.g. PMID: 31955846, 37081202, 37770433). Parasites lacking BFD1 do not differentiate when SPARK and PKA C3 are depleted, suggesting that differentiation caused by SPARK or PKA C3 depletion occurs through the BFD1 circuit. If differentiation caused by SPARK or PKA C3 depletion did not depend on the BFD1 circuit, we might have observed differentiation in the SPARK- and PKA C3-AID/∆bfd1 mutants.

To clarify this point, we have changed the first sentences of the last paragraph in the results section Depletion of SPARK, SPARKEL, or PKA C3 promotes chronic differentiation: “To assess whether excess differentiation caused by SPARK and PKA C3 depletion is dependent on a previously characterized transcriptional regulator of differentiation, BFD1 (Waldman et al., 2020), we knocked out the BFD1 CDS with a sortable dTomato cassette in the SPARK- and PKA C3-AID strains (Figure 6–figure supplement 1). The resulting SPARK- and PKA C3-AID/∆bfd1 mutants failed to undergo differentiation as measured by cyst wall staining (Figure 6G–H), suggesting that differentiation caused by depletion of these kinases depends on the BFD1 circuit.”

Lines 239-242. The logic behind the categories of "constitutively regulated sites" and "newly synthesized proteins dependent on SPARK activation" is odd. The former (3h treatment) represents the SPARK-specific events (even though it should be shortened to 1-2h), while an 8h treatment is already contaminated with secondary effects. Since Toxoplasma divides asynchronously, the "newly synthesized" proteins will be present at the time. Also, the protein phosphorylation does not always lead to substrate activation; it can be repressive, too.

We describe the logic in response to a comment above (substrate A vs. substrate B). It is correct that *T. gondii* divides asynchronously, with a cell cycle of approximately 8 hours, and 60% of parasites in G1 at a given time (PMID: 11420103). The proteomics experiments measure peptide and protein abundances at a population level. Newly synthesized proteins will be present at all time points; but the proportion of proteins synthesized after SPARK depletion relative to proteins synthesized before SPARK depletion will increase over time.

We moved lines 238-243 from the first submission to the Discussion.

It is accurate that phosphorylation does not always lead to substrate activation; it can also be repressive or not change substrate behavior. However, in the case of protein kinases, activation loop phosphorylation is highly correlated with activation (e.g. PMID: 15350212, 31521607).

Line 250-252: Because the SPARK degradation did not affect intracellular replication, SPARK is unlikely to affect cell cycle-specific phosphorylation.

To parallel the prior sentences describing different SPARK-dependent down-regulated clusters, we truncated this sentence to “The final cluster of depleted phosphopeptides, Cluster 4, only exhibits down-regulation at 8h of IAA treatment.”

SPARKEL depletion did not significantly affect intracellular replication under the time frames investigated here (approximately 25 hours post-invasion; Figure 2D). A prior study reported that SPARK depletion did not affect intracellular replication measured on a similar timescale (PMID: 35484233).

The opening sentence of the Discussion: Typically, we refer to the newly discovered proteins as the orthologs of the previously discovered counterparts and not the vice versa. Thus, calling Toxoplasma SPARK the ortholog of mammalian PDK1 would be more appropriate.

We changed the opening sentence of the Discussion to “SPARK is an ortholog of PDK1, which is considered a key regulator of AGC kinases”.

**Reviewer #3 (Recommendations For The Authors):**
(1) Authors should show alignment of SPARKEL with Elongin C. Are key residues conserved?

We have added an alignment of the SKP1/BTB/POZ domains of Homo sapiens elongin C, *S. cerevisiae* elongin C, and *T. gondii* SPARKEL as Figure 1—figure supplement 1B. This panel highlights elongin B interface, cullin binding sites, and target protein binding sites based on the human elongin C annotation. As discussed below, these interfaces may not be functionally conserved in *T. gondii*. Ultimately, future mechanistic and structural studies beyond the scope of the current work will be required to determine how SPARK and SPARKEL physically interact. The Discussion states, “further biochemical studies are required to discern the regulatory interactions between SPARK and SPARKEL” (lines 590-591).

(2) The failure to identify other Elongin B/C complex members should be addressed by direct IP analysis.

Indeed, elongin C has traditionally been characterized as a component of multisubunit complexes comprising Elongin A/B/C or Elongin BC/cullin/SOCS that regulate transcription or function as ubiquitin ligases, respectively (for a review, PMID: 22649776). We see two major issues when attempting to generalize these results to apicomplexan parasites. First, nearly all studies of the function of elongin C have been conducted in a single eukaryotic supergroup (the opisthokonts, including yeast and metazoans). The majority of eukaryotic diversity exists in other supergroups, including the SAR supergroup to which apicomplexans such as *T. gondii* belong (PMID: 31606140). Proteins with elongin C domains may serve alternative and unexplored functions in non-opisthokont unicellular eukaryotes. Second (in support of the first), we were unable to find orthologs of many of the opisthokont complex members in *T. gondii*, as systematically described below.

By BLAST, the most similar protein to SPARKEL in *S. cerevisiae* is ELC1 (YPL046C), with a BLAST E = 0.003. The next most similar protein was SCF ubiquitin ligase subunit SKP1 (YDR328C) with an E value of 0.62. ELC1 is 99 amino acids. The Elongin C (IPR039948) and SKP1/BTB/POZ superfamily domains (IPR011333) span most of this sequence. SPARKEL is 216 amino acids; the Elongin C and SKP1/BTB/POZ superfamily domains occupy the C-terminal half of the protein. The N-terminal domain of SPARKEL may be important for its function; however, future work is required to address this hypothesis.

Elongin B: Elongin B is not found universally amongst even opisthokonts; fungi and choanoflagellates lack obvious orthologs. The most similar *T. gondii* protein to human Elongin B (Q15370) by BLAST is TGME49_223125 (E = 0.017), an apicoplast ubiquitin-like protein PUBL (PMID: 28655825, 33053376). TGME49_223125 has a C-terminal ubiquitin-like domain (IPR000626) but no ELOB domain (IPR039049); indeed, no T. gondii protein has an ELOB domain that can be identified by sequence searching. Given the lack of similarity between EloB and TGME49_223125, as well as this protein’s possible red algal endosymbiont origin, we consider it an unlikely ortholog of EloB and topologically unlikely to interact with the SPARK/SPARKEL complex. We did not detect TGME49_223125 in SPARK or SPARKEL IPs (Supplementary File 1).

Elongin A: *T. gondii* appears to lack a human elongin A ortholog (Q14241) on the basis of sequence similarity. The most similar T. gondii protein to yeast Elongin A (O59671) by BLAST is TGME49_299230 (E = 0.022). Yeast EloA is 263 amino acids. TGME49_299230 is 1101 amino acids and does not have an EloA domain (IPR010684), suggesting it is not a true EloA ortholog.

Suppressor of cytokine signaling (SOCS): T. gondii appears to lack human SOCS1 or SOCS2 orthologs (O15524 and O14508) on the basis of sequence similarity. We were unable to identify *T. gondii* proteins with SOCS domains (PF07525, SM00253, SM00969, and SSF158235).

Von Hippel-Lindau tumor suppressor (VHL): T. gondii appears to lack a human VHL ortholog (P40337) on the basis of sequence similarity. We were unable to identify *T. gondii* proteins with VHL domains (IPR024048, IPR024053, PF01847, and SSF49468).

Cul-2/5: Cullins appeared early in the eukaryotic radiation (PMID: 21554755), and thus *T. gondii* possesses several. Since the ELC complex has been best characterized with human cullin-2 (Q13617) and cullin-5 (Q93034), we searched for orthologs of these proteins and identified TGME49_289310, TGME49_289310, and TGME49_316660. TGME49_289310 functionally resembles cullin-1 of the SCF complex (PMID: 31348812). None of these proteins were enriched in the SPARK or SPARKEL IPs (Supplementary Table 1).

Rbx1: We searched for human Rbx1 orthologs (P62877) and identified TGME49_213690, which functionally resembles Rbx1 of the SCF complex (PMID: 31348812); as well as several other RING proteins (TGME49_267520, TGME49_277740, TGME49_261990, and TGME49_232160) that were not found in the SPARK or SPARKEL IPs (Supplementary File 1).

Rbx2: We searched for human Rbx2 orthologs (Q9UBF6) and identified several RING proteins (TGME49_285190, TGME49_254700, TGME49_292340, TGME49_226740, TGME49_244610, and TGME49_304460) that were not found in the SPARK or SPARKEL IPs (Supplementary File 1). No *T. gondii* protein has an Rbx2 domain (cd16466) that can be identified by sequence searching.

In conclusion, we conducted “direct IP analysis” (Figure 1A, 1D; Figure 1-supplement 1A) of the SPARK and SPARKEL complex in the first submission of the manuscript. The observation that SPARK and SPARKEL form strong interactions was validated in cellulo via proximity labeling (Figure 1E; Figure 1-supplement 1B) in the first submission of the manuscript. These results are described together in the results section SPARK complexes with an elongin-like protein, SPARKEL (lines 75-110, first submission of manuscript). The failure to identify an interaction between SPARKEL and Elongin B/C complex members in T. gondii may be due to the observation that Elongin B and several ELC complex members do not exist in most eukaryotes, including *T. gondii*. We added the sentences “The function of proteins with Elongin C-like domains has not been widely investigated in unicellular eukaryotes” to the Results and “However, the SPARK and SPARKEL IPs and proximity experiments failed to identify obvious components of ubiquitin ligase complexes” to the Discussion.

(3) PKA and PKG half-lives should be measured as well as their transcript abundances.

The finding that PKA C1 and PKG protein abundances decreased upon SPARK/SPARKEL depletion was internally consistent across experiments. This down-regulation may be due to transcriptional, translational, or post-translational mechanisms. We measured PKG and PKA C1 transcript abundances in SPARK-AID and TIR1 parasites after 24 hours of IAA treatment using RT-qPCR. We did not detect significant differences in transcript levels of the queried kinases. These findings suggest that SPARK depletion leads to PKG and PKA down-regulation through post-transcriptional mechanisms. Translational control is normally enacted globally, for example through regulation of eukaryotic translation factors (PMID: 15459663). The rapid and specific down-regulation of PKG and PKA C1 would suggest that the kinase abundance levels are regulated by non-global translational mechanisms (e.g. mRNA-specific) or rather post-translational mechanisms.

Substantial additional work is required to determine protein half-lives in eukaryotic parasites. In our discussion of possible mechanisms and models, we were agnostic as to the cause of reduced PKG and PKA abundances upon SPARK depletion. We note in the discussion, “The cause for reduction of PKA C1 and PKG levels requires further study” (lines 541-542).